# The 2023 Foundation Model Transparency Index

**Rishi Bommasani\***                                                     *nlprishi@stanford.edu*
*Stanford University*

**Kevin Klyman\***                                                        *kklyman@stanford.edu*
*Stanford University*

**Shayne Longpre**                                                        *slongpre@media.mit.edu*
*Massachusetts Institute of Technology*

**Sayash Kapoor**                                                         *sayashk@princeton.edu*
*Princeton University*

**Nestor Maslej**                                                         *nmaslej@stanford.edu*
*Stanford University*

**Betty Xiong**                                                           *xiongb@stanford.edu*
*Stanford University*

**Daniel Zhang**                                                          *dzhang105@stanford.edu*
*Stanford University*

**Percy Liang**                                                           *pliang@cs.stanford.edu*
*Stanford University*

**Reviewed on OpenReview:** *https://openreview.net/forum?id=x6fXnsM9Ez*

## Abstract

Foundation models have rapidly permeated society, catalyzing a wave of generative AI applications spanning enterprise and consumer-facing contexts. While the societal impact of foundation models is growing, transparency is on the decline, mirroring the opacity that has plagued past digital technologies (e.g. social media). Reversing this trend is essential: transparency is a vital precondition for public accountability, scientific innovation, and effective governance. To assess the transparency of the foundation model ecosystem and help improve transparency over time, we introduce the **Foundation Model Transparency Index**. The 2023 Foundation Model Transparency Index specifies 100 fine-grained indicators that comprehensively codify transparency for foundation models, spanning the *upstream* resources used to build a foundation model (e.g. data, labor, compute), details about the *model* itself (e.g. size, capabilities, risks), and the *downstream* use (e.g. distribution channels, usage policies, affected geographies). We score 10 major foundation model developers (e.g. OpenAI, Google, Meta) against the 100 indicators to assess their transparency. To facilitate and standardize assessment, we score developers in relation to their practices for their flagship foundation model (e.g. GPT-4 for OpenAI, PaLM 2 for Google, Llama 2 for Meta). We present 10 top-level findings about the foundation model ecosystem: for example, no developer currently discloses significant information about the downstream impact of its flagship model, such as the number of users, affected market sectors, or how users can seek redress for harm. Overall, the Foundation Model Transparency Index establishes the level of transparency today to drive progress on foundation model governance via industry standards and regulatory intervention.

# 1 Introduction

Foundation models (FMs) like LLaMA and DALL-E 3 are an emerging class of digital technology that has transformed artificial intelligence (Bommasani et al., 2021). These resource-intensive models are often built by processing trillions of bytes of data, with some of the most capable systems, like OpenAI's GPT-4, costing hundreds of millions of dollars to build.[1] Foundation models power some of the fastest-growing consumer technologies in history,[2] including myriad generative AI applications,[3] bringing immense commercial investment and public awareness to AI. Simultaneously, these models have captured the interest of policymakers around the world: the United States,[4] China,[5] Canada,[6] the European Union,[7] the United Kingdom,[8] India,[9] Japan,[10] the G7,[11] and a wide range of other governments have already taken action on foundation models and generative AI. Foundation models are positioned to be the defining digital technology of the decade ahead.

Transparency is an essential precondition for public accountability, scientific innovation, and effective governance of digital technologies. Without adequate transparency, stakeholders cannot understand foundation models, who they affect, and the impact they have on society. Historically, digital technologies often follow a familiar pattern: a new technology provides opportunities and benefits, but companies are not transparent in how they develop and deploy the technology, and this opacity eventually leads to harm. In the case of social media, companies have not been transparent about the ways in which they moderate content and share user data, contributing to massacres like the Rohingya genocide in Myanmar[12] and gross violations of privacy like the Cambridge Analytica scandal.[13] Consequently, a chorus of academics, civil society organizations, firms, and governments have called for foundation model developers to improve transparency.[14] Groups such as the Partnership on AI, Mozilla, and Freedom House have noted that increased transparency is a crucial intervention.[15] UN Secretary-General António Guterres has proposed that the international community should "make transparency, fairness and accountability the core of AI governance ... [and] Consider the adoption of a declaration on data rights that enshrines transparency."[16]

Foundation models appear to be on track to replicate the opacity of social media. Consider OpenAI's GPT-4, one of the most influential foundation models today. OpenAI states plainly its intention to be nontransparent in the GPT-4 technical report, which "contains no further details about the architecture (including model size), hardware, training compute, dataset construction, training method, or similar" (OpenAI, 2023). Companies often claim that such information is proprietary or that sharing it would undermine their market position and pose a danger to society as a whole, but this does not negate the enormous risks stemming from foundation models these same companies openly acknowledge, as well as the value of greater transparency.

While the downsides of opacity are clear, transparency in the foundation model ecosystem today remains minimal. Little to no evidence exists about which foundation model developers are transparent about which

---

[1]https://www.wired.com/story/openai-ceo-sam-altman-the-age-of-giant-ai-models-is-already-over/

[2]https://www.reuters.com/technology/chatgpt-sets-record-fastest-growing-user-base-analyst-note-2023-02-01/

[3]https://www.mckinsey.com/capabilities/mckinsey-digital/our-insights/the-economic-potential-of-generative-ai-the-next-productivity-frontier

[4]https://www.whitehouse.gov/wp-content/uploads/2023/07/Ensuring-Safe-Secure-and-Trustworthy-AI.pdf

[5]http://www.cac.gov.cn/2023-07/13/c_1690898327029107.htm

[6]https://ised-isde.canada.ca/site/ised/en/voluntary-code-conduct-responsible-development-and-management-advanced-generative-ai-systems

[7]https://www.europarl.europa.eu/news/en/press-room/20230609IPR96212/meps-ready-to-negotiate-first-ever-rules-for-safe-and-transparent-ai

[8]https://www.gov.uk/cma-cases/ai-foundation-models-initial-review

[9]https://indiaai.s3.ap-south-1.amazonaws.com/docs/generative-ai-report.pdf

[10]https://english.kyodonews.net/news/2023/10/3b83adf1e28d-japans-ai-draft-guidelines-ask-for-measures-to-address-overreliance.html

[11]https://www.politico.eu/wp-content/uploads/2023/09/07/3e39b82d-464d-403a-b6cb-dc0e1bdec642-230906_Ministerial-clean-Draft-Hiroshima-Ministers-Statement68.pdf

[12]/https://about.fb.com/wp-content/uploads/2018/11/bsr-facebook-myanmar-hria_final.pdf

[13]https://www.nytimes.com/2018/04/04/us/politics/cambridge-analytica-scandal-fallout.html

[14]See Appendix D for further discussion.

[15]http://partnershiponai.org/wp-content/uploads/2021/08/PAI-Responsible-Sourcing-of-Data-Enrichment-Services.pdf

[16]https://indonesia.un.org/sites/default/files/2023-07/our-common-agenda-policy-brief-gobal-digi-compact-en.pdf

matters, and where there are blind spots in the industry. How best to improve transparency remains an open question despite rising concerns.

To determine the status quo and track how it evolves over time, we introduce the Foundation Model Transparency Index (FMTI).[17] A composite *index* measures a complex construct (e.g. transparency) as the basis for scoring/ranking entities (e.g. foundation model developers) by aggregating many low-level quantifiable *indicators* of transparency. Indexes are not common in AI[18] but are a standard methodology in the social sciences: iconic examples include the United Nations Development Programme's Human Development Index (UNDP, 2022), which ranks countries, and Ranking Digital Rights' Corporate Accountability Index, which ranks companies (RDR, 2020). We score the transparency of foundation model developers in an effort to promote responsible business practices and greater public accountability. We deconstruct the concept of transparency into 3 high-level domains: the *upstream* (e.g. the data, labor, and compute resources used to build a foundation model), *model*-level (e.g. the capabilities, risks, and evaluations of the foundation model), and *downstream* (e.g. the distribution channels, usage policies, and affected geographies) practices of the foundation model developer.

**The 2023 Foundation Model Transparency Index.** For the 2023 index, each domain is further broken down into 32–35 indicators: these are concrete, specific, and decidable aspects of transparency (e.g. does the foundation model developer disclose the size of its model?). Ultimately, the index consists of 100 indicators (see Appendix B) that comprehensively codify what it means for a foundation model developer to be transparent, building upon formative works on transparency for AI and other digital technologies (Gebru et al., 2021; Bender & Friedman, 2018b; Mitchell et al., 2018; Raji & Buolamwini, 2019; Gray & Suri, 2019a; Crawford, 2021; Vogus & Llansó, 2021; Keller, 2022).[19]

We score 10 major foundation model developers on each of the 100 indicators to determine how transparent each company is in the development and deployment of its models. In particular, we score developers based on their practices in relation to their flagship foundation models: we assess OpenAI (GPT-4), Anthropic (Claude 2), Google (PaLM 2), Meta (Llama 2), Inflection (Inflection-1), Amazon (Titan Text), Cohere (Command), AI21 Labs (Jurassic-2), Hugging Face (BLOOMZ; as host of BigScience),[20] and Stability AI (Stable Diffusion 2). In addition, for downstream indicators we consider the flagship or in-house distribution channel: OpenAI (OpenAI API), Anthropic (Claude API), Google (PaLM API), Meta (Microsoft Azure),[21] Inflection (Pi), Amazon (Bedrock), Cohere (Cohere API), AI21 Labs (AI21 Studio), Hugging Face (Hugging Face Model Hub), and Stability AI (Stability API). We assess developers on the basis of publicly-available information to make our findings reproducible and encourage transparency vis-à-vis the public on the whole. To ensure our scoring is consistent, we identify information using a rigorous search protocol (see Appendix C). To ensure our scoring is accurate, we notified developers and provided them the opportunity to contest any scores prior to the release of this work (all 10 responded and 8 of the 10 explicitly contested some scores). We summarize our core findings, recommendations, and contributions below and make all core materials (e.g. indicators, scores, justifications, visuals) publicly available.[22]

## 1.1 Findings

On the basis of conducting the index, we extensively catalogue 35 empirical findings in §7: RESULTS spanning overarching trends, domain-level analyses, breakdowns for open vs. closed developers, and similarities in developer practices. We summarize the 10 most critical findings.

---

[17]See `https://crfm.stanford.edu/fmti`.

[18]We note that the AI Index from the Stanford Institute for Human-Centered AI (Zhang et al., 2022a; Maslej et al., 2023) is a related effort, but the AI Index tracks broader trends in AI, rather than scoring specific entities or aggregating to a single value.

[19]See `https://transparency.dsa.ec.europa.eu/` and `https://www.tspa.org/curriculum/ts-fundamentals/transparency-report/`

[20]Our objective is to assess Hugging Face as a company that can be tracked over time. BLOOMZ however was not built unilaterally by Hugging Face, but instead through the BigScience open collaboration. As a result, we refer to Hugging Face in the prose but include the BigScience logo in visuals; we provide further discussion in §5.2: MODEL-SELECTION.

[21]Meta announced Microsoft as the "preferred partner" for Llama 2 via Azure: `https://about.fb.com/news/2023/07/llama-2/`

[22]`https://www.github.com/stanford-crfm/fmti`

**Significant but obtainable headroom in overall transparency scores (Figure 7).** Given that the highest overall score is 54 out of 100 and the mean overall score is 37, all developers have significant room for improvement. In many cases, such improvement is already feasible: 82 of the indicators are achieved by some developer, and 71 are achieved by multiple developers.

**Significant unevenness in overall scores with three major clusters (Figure 7).** Overall scores vary significantly given a range of 42 between the highest-scoring developer, Meta, at 54 and the lowest-scoring developer, Amazon, at 12. Relative to the mean overall score of 37, organizations group into three clusters: four well-above the mean (Meta, Hugging Face, OpenAI, Stability AI), three around the mean (Google, Anthropic, Cohere), and three well-below the mean (AI21 Labs, Inflection, Amazon).

**Upstream resource transparency scores the worst (Figure 8).** Breaking down the trends by domain, scores are consistently the worst for the upstream domain, particularly the Data, Data Labor, and Compute subdomains. Several developers (AI21 Labs, Inflection, Amazon) receive 0 points across the entire set of 32 indicators for the upstream domain.

**Several upstream matters surrounding data creation are fully opaque (Figure 10).** Within the upstream domain, no company scores points for indicators about data creators, the copyright and license status of data, and mitigations related to copyright. The industry-wide lack of transparency on these issues relates directly to pressing societal concerns related to copyright and intellectual property, which are the subject of ongoing litigation.

**Transparency is highest, but still imperfect, for very basic model information and downstream distribution (Figure 9).** Breaking down the trends by major dimensions of transparency, the highest-scoring dimensions are Methods, Model Basics, Capabilities, and Distribution. However, even when considering indicators for these high-scoring dimensions of transparency, most companies do not reveal basic information like model size nor do they explain how or why they made certain release decisions.

**Developer transparency on Capabilities does not translate to transparency on Limitations, Risks, and Model Mitigations (Figure 11).** Within the model domain, we consider Capabilities, Limitations, Risks, and Model Mitigations as four tightly-related subdomains that characterize a model's potential societal impact. While many developers score well on Capabilities by describing, demonstrating, and evaluating capabilities, which reflect their models' strengths, the same cannot be said for the other three subdomains. Instead, transparency is significantly worse: just two developers demonstrate limitations, none evaluate multiple intentional harms their models could facilitate, and none provide either externally reproducible or third-party assessments of mitigation efficacy.

**There is virtually no transparency about the downstream impact of foundation models (Figure 12).** Within the downstream domain, no developer provides any transparency into the affected market sectors, affected individuals, affected geographies, or any form of usage reporting. Overall, the average score on the Impact subdomain is the worst in the entire index at 11%; only three developers provide even the most minimal characterization of the number of downstream applications, and no developer provides a mechanism for users to seek redress.

**Open developers are consistently more transparent than closed developers (Figure 13).** Breaking down trends by how developers release their models, open developers (i.e. those that release model weights and, potentially, data) show a clear edge in transparency over closed counterparts (e.g. API providers). Two of the three open developers (Meta and Hugging Face) score better than all other developers, while the third (Stability AI) scores one point below the highest-performing closed developer (OpenAI). Open developers have higher average scores on 17 of the 23 subdomains.

**Open developers are much more transparent on upstream resources and comparably transparent on downstream use when compared to closed developers (Figure 13).** The average score for open developers on upstream indicators is 53% compared to a paltry 9% of closed developers. However, while

closed developers have greater control over the downstream use of their foundation models, this does not translate to greater downstream transparency as the average for open developers on downstream indicators is 49% compared to 43% from closed developers.

**Some companies have highly correlated scores (Figure 14).** Considering pairs of companies, we analyze the extent to which they agree on the indicators where they do and do not score points. In particular, the three members of the Frontier Model Forum (Anthropic, Google, OpenAI) exhibit high indicator-level similarity, as do the two companies that release both model weights and data (Hugging Face, Stability AI) and the four lowest-scoring companies (Cohere, AI21 Labs, Inflection, Amazon). This leaves Meta as the sole outlier in terms of developer-developer indicator-level similarity.

### 1.2 Recommendations

On the basis of our findings, we make specific recommendations aimed at foundation model developers, foundation model deployers, and policymakers in §8: RECOMMENDATIONS. We highlight our top-most recommendation for each stakeholder group.

**Foundation model developers should improve transparency by drawing on the practices of their competitors.** By assessing developers directly, we clarify for each developer the indicators where they lack transparency. In itself, this provides a clear diagnostic on where they stand relative to their competitors and, given our justifications for why transparency on these matters is valuable, why improving transparency would be beneficial for society. Given that 82 indicators are all satisfied by some developer, developers can directly consult the practices of their competitors to provide a clear example of how they might improve their transparency. There is a tremendous gap between the 82 already-feasible indicators and the current top score of 54 and the mean score of 37, meaning there are many areas of low-hanging fruit where developers can readily improve transparency today.

**Foundation model deployers should push for greater transparency from developers.** Foundation models intermediate a growing supply chain: deployers of foundation models (e.g. cloud service providers and companies that license developers' models) as well as other downstream actors are influenced by, and can influence, the transparency of foundation model developers. In particular, deployers should push developers for greater transparency when making the decision, and potentially negotiating the contract, to deploy a developer's model. Deployers and other downstream actors wield leverage collectively: it is their downstream use that generates users and revenue for foundation model developers, meaning they should use this leverage to acquire the necessary transparency from foundation model developers.

**Policymakers should prioritize transparency with sufficient precision.** Given the importance of transparency, policymakers should make transparency a top priority in legislative proposals and regulatory enforcement related to foundation models. While transparency is already broadly recognized in most regulatory frameworks for AI, policymakers should be more precise about what they mean by transparency and the areas in which they hope to reduce opacity via transparency requirements or other measures. In particular, policymakers should understand the status quo for transparency (e.g. via the scores we provide) and use this evidence to inform interventions in the areas where transparency is most urgently needed (e.g. on Data Labor and Impact, given these are lowest-scoring dimensions of transparency across the entire supply chain).

### 1.3 Contributions

To summarize, our contributions are:

1. **Taxonomy.** We taxonomize the vast conceptual space of transparency in the context of foundation models, following on widespread calls for transparency (see Appendix D). In particular, we structure the space hierarchically into 3 domains (i.e. upstream, model, downstream), 23 subdomains (e.g. data, compute, capabilities, risks, distribution, feedback), and 100 decidable and actionable indicators.

2. **Scoring of major foundation model developers.** We score 10 major foundation model developers and their flagship foundation models with a standardized protocol. These developers vary in their company status (e.g. startups, Big Tech), release strategy (e.g. open weights, restricted API), modalities (e.g. text-to-text, text-to-image), and involvement in global policy efforts (e.g. White House voluntary commitments, Frontier Model Forum). We allow developers to directly contest scores: all 10 developers engaged in correspondence and 8 contested specific scores.

3. **Empirical findings.** Our extensive evaluation yields 35 findings, which ground existing discourse and sharpen our understanding of the lack of transparency in the foundation model ecosystem. In many cases, these findings directly bear on critical global AI policy efforts (e.g. the EU AI Act) and provide the basis for clear recommendations on how developers may improve their practices (e.g. by creating centralized documentation artifacts). Our scores offer ample opportunities for further analysis.

4. **Legibility and reproducibility.** We provide a public website that presents our findings and recommendations broadly legible to the general audience.[23] To facilitate further research, and reproduce our scoring and analyses, we make all core materials (e.g. indicators, scores, justifications, visuals) publicly available.[24]

5. **Theory of change and future versions.** Our objective is to simultaneously articulate the status quo and increase transparency over time. To this end, we make very explicit our theory of change: we view our work as compiling the transparency practices across es across companies as an instrument for driving change (see §9.1: CHANGE) and the limitations/risks of our work (see §9.2: LIMITATIONS). Critically, we will conduct additional iterations of the index to track progress over time to work towards a more transparent foundation model ecosystem.

---

[23]https://crfm.stanford.edu/fmti
[24]https://www.github.com/stanford-crfm/fmti

## 2 Background

To begin, we provide a brief primer on the three core concepts underlying this work: foundation models, transparency, and indexes.

### 2.1 Foundation models

Foundation models are the defining paradigm of modern AI, reflecting a broad shift in the field from bespoke models for individual tasks to more general models that can be adapted for a wide range of use cases Bommasani et al. (2021). In this sense, foundation models belong to the broader class of general-purpose technologies that have restructured society such as electricity, the Internet, and smartphones (Bresnahan & Trajtenberg, 1995; Brynjolfsson et al., 2021; Bommasani et al., 2021; Eloundou et al., 2023). Building foundation models requires significant resources: immense volumes of data are processed using immense amounts of computation to yield the foundation model. Using foundation models often requires substantially fewer resources in comparison: models can be adapted, often in lightweight fashion (e.g. through a simple textual interface), for an increasingly wide range of use cases. The disparity in resource requirements between development and deployment has yielded a market where a small set of companies build the most prominent foundation models that are then adopted by thousands of companies and millions of consumers (Bommasani et al., 2023b; Vipra & Korinek, 2023; Widder et al., 2023).

The structure of the foundation model paradigm implicates a broader ecosystem and supply chain Bommasani et al. (2023b); Cen et al. (2023); Jones (2023). We depict a conceptualized view of this supply chain in Figure 1. The supply chain begins with the *upstream* resources that are used to build a foundation model: data, computational hardware, energy, labor, and code. For each of these resources, a further supply chain exists: for example, data to build foundation models is often sourced from the Internet, but this data can only come to be on the Internet as a result of human data-generating process (e.g. publishing news article, authoring personal blogs, uploading videos to YouTube, creating music) along with Internet infrastructure (e.g. networking protocols). Alongside these upstream resources and supply chains, foundation models are then used as the foundation for supply chains that derive from the model. In particular, foundation models are made available for downstream use through *distribution channels* (e.g. an API to access the model or a host that facilitates inference using the model). By way of these distribution channels, foundation models power *downstream* applications (e.g. commercial products and services) across a range of market sectors and geographies. For instance, OpenAI's GPT-4 powers applications in education (e.g. Khan Academy's Khanmigo tutor), finance (e.g. Stripe's fraud detection tool), banking (e.g. Morgan Stanley's internal chatbot), and government (e.g. Iceland's language preservation system).[25] Overall, a comprehensive account of the societal impact of foundation models, and their transparency in particular, requires consideration of the different parts of the foundation model ecosystem (Bommasani et al., 2021, §1.2).

Foundation models have fueled the recent wave of generative AI technologies: these models can be used to generate fluent text, useful code, photorealistic images, and compelling audio. New research efforts built foundation models in an even broader array of domains: biology (Lin et al., 2023), climate change (Lacoste et al., 2023), weather,[26] astronomy (Nguyen et al., 2023), radiology (Chambon et al., 2022), and robotics (Open X-Embodiment Collaboration et al., 2023). Nevertheless, much of the present public and commercial interest centers on language models (e.g. Anthropic's Claude 2, Meta's Llama 2) and multimodal models with language interfaces (e.g. Stability AI's Stable Diffusion 2, OpenAI's GPT-4). Alongside their significant capabilities, researchers have highlighted a large number of potential risks posed by these foundation models spanning malicious uses like generating disinformation to unintended harms like generating text that reinforces societal biases (Bender et al., 2021; Bommasani et al., 2021; Abid et al., 2021; Weidinger et al., 2022). There have also been recent demonstrations of many concrete harms from language models.[27]

---

[25]See `https://openai.com/gpt-4` for a list of several applications built upon OpenAI's GPT-4.

[26]`https://www.earthdata.nasa.gov/news/weather-ai-fm-workshop`

[27]Partnership on AI's AI Incident database (`https://incidentdatabase.ai/`) and the AI, Algorithmic, and Automation Incidents and Controversies database (`https://www.aiaaic.org/aiaaic-repository`) collect incidents of harm caused by AI. For a concrete example, see `https://www.404media.co/inside-the-ai-porn-marketplace-where-everything-and-everyone-is-for-sale/`.

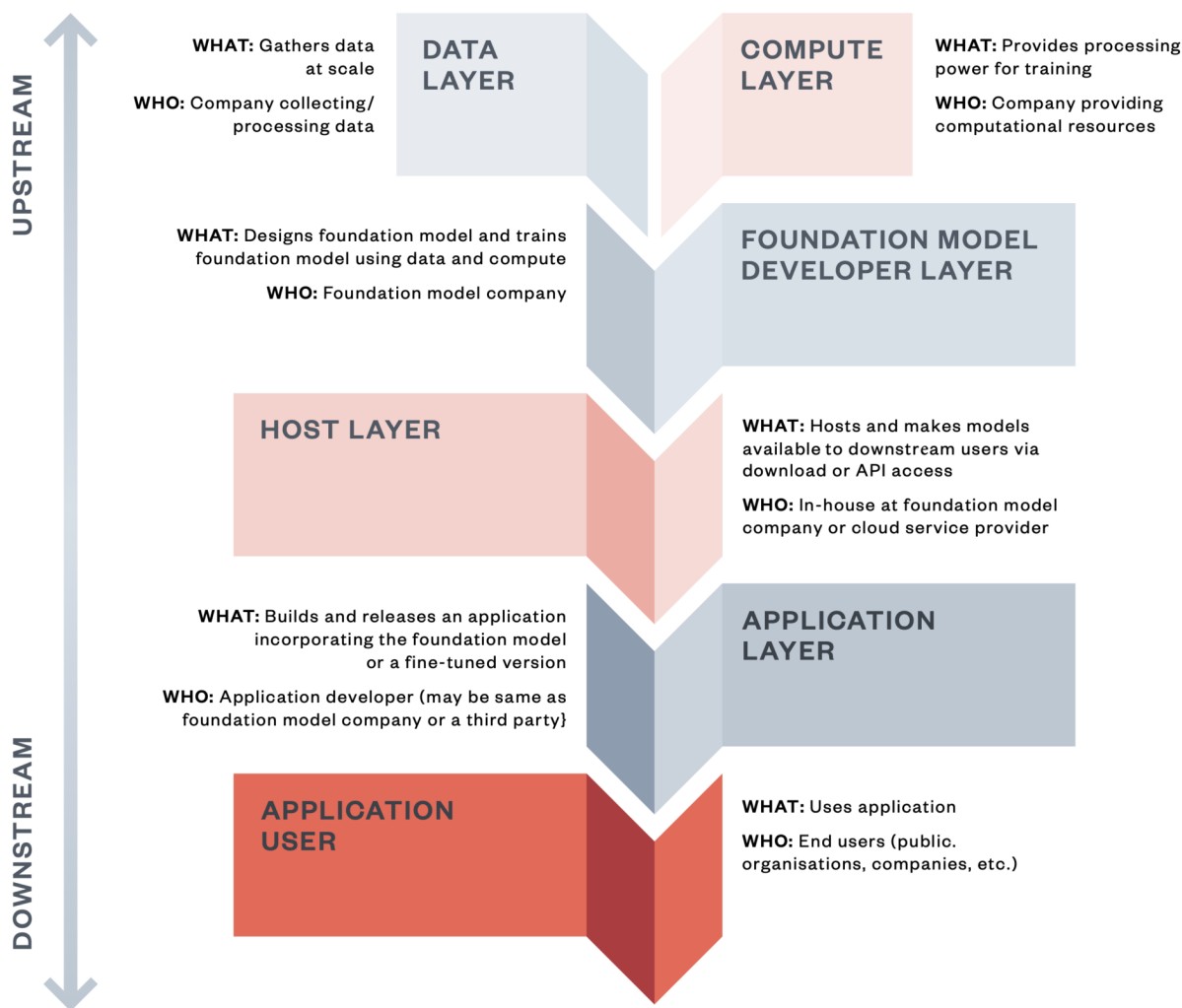

Figure 1: **Foundation Model Supply Chain.** A conceptual depiction of the foundation model supply chain, beginning with the primary *upstream* resources (i.e. data, compute) and transitioning to the foundation model, subsequent hosts (or *distribution channels*), and ending with *downstream* applications. Image taken with permission from Jones (2023).

## 2.2 Transparency

Transparency is broadly understood as the property of being visible and easily understood (Aristotle, 350 B.C.E; Kalderon, 2015), and is often a fundamental prerequisite of social responsibility and accountability (Florini, 2007; Robinson & Acemoglu, 2012).

Transparency is desirable from a variety of standpoints. For example, transparently disclosing information makes that information available, shareable, legible, and verifiable. Transparency when conducting a complex process can make clear the processes' scope, stakes, and pitfalls (Lathrop & Ruma, 2010). Similarly, transparency in decision-making can help those who are not involved in the decision assess the motivations behind the decision, the evidence used to justify it, as well as its costs and benefits. Various philosophers, political theorists, scientists, and journalists have emphasized the importance of transparency across these and other domains (Johnston, 2006; Florini, 2007; Benkler, 2013; Schudson, 2015). Civil society, grassroots organizations, and consumers also regularly call for transparency as a mechanism for fact finding, accountability, and holding organizations responsible for harm (Heikkilä, 2023; DiResta et al., 2022).[28] For our purposes, we consider transparency as it relates to the development and use of digital technologies, with a specific focus on the transparency of the practices of foundation model developers as measured by the information they share regarding their models.[29]

**Why transparent matters for digital technologies.** Transparency in digital technologies is particularly relevant for three reasons. First, new digital technologies, such as AI, are not well understood by society, often appearing as a black box (Castelvecchi, 2016). Second, digital technologies are easily rendered invisible, meaning it is difficult for nonexperts to understand when processes like algorithmic decision-making are taking place (Ng et al., 2021). Third, these technologies can have a profound influence on billions of users across society. And yet these technologies are built by a small cadre of industry actors who do not represent society as a whole. Under these conditions, transparency functions as a prerequisite for public accountability and responsible innovation (Klyman, 2023). Shared visibility engenders public trust and facilitates interventions in the public interest (Hardin, 2002). Without sufficient understanding of industry practices, researchers cannot characterize the societal impact of digital technologies, let alone propose concrete actions to improve business practices (Pasquale, 2015). While the effects of transparency are often difficult to measure as they are diffuse and indirect, transparency helps to expose malpractice and enables the public to respond to such malpractice.

**Limitations of transparency.** Transparency is far from sufficient on its own and it may not always bring about the desired change (Corbett & Denton, 2023). Salient critiques of transparency include:

- Transparency does not equate to responsibility. Without broad based grassroots movements to exert public pressure or concerted government scrutiny, organizations often do not change bad practices (Boyd, 2016; Ananny & Crawford, 2018).

- Transparency-washing provides the illusion of progress. Some organizations may misappropriate transparency as a means for subverting further scrutiny. For instance, major technology companies that vocally support transparency have been accused of *transparency-washing*, whereby "a focus on transparency acts as an obfuscation and redirection from more substantive and fundamental questions about the concentration of power, substantial policies and actions of technology behemoths" (Zalnieriute, 2021).

- Transparency can be gamified. Digital platforms have been accused of performative transparency, offering less insightful information in the place of useful and actionable visibility (Ghosh & Faxon, 2023; Mittelstadt, 2019). As with other metrics, improving transparency can be turned into a game, the object of which is not necessarily to share valuable information.[30]

---

[28]See Appendix D for additional details on calls for transparency.

[29]Note that the term "transparency" is at times also used to describe efforts to make AI more explainable or interpretable at the level of specific AI-based predictions or decisions (Liao & Vaughan, 2023; Zou et al., 2023). Such transparency is not the subject of our work.

[30]According to Goodhart's Law, "when a measure becomes a target, it ceases to be a good measure" (Goodhart, 1984).

- Transparency can inhibit privacy and promote surveillance. Transparency is not an apolitical concept and is often instrumentalized to increase surveillance and diminish privacy (Han, 2015; Mohamed et al., 2020; Birchall, 2021). For foundation models, this critique underscores a potential tension between adequate transparency with respect to the data used to build foundation models and robust data privacy.

- Transparency may compromise competitive advantage or intellectual property rights. Protections of competitive advantage plays a central role in providing companies to the incentives to innovate, thereby yielding competition in the marketplace that benefits consumers. Consequently, work in economics and management studies have studied the interplay and potential trade-off between competitive advantage and transparency (Bloomfield & O'Hara, 1999; Granados & Gupta, 2013; Liu et al., 2023), especially in the discourse on corporate social responsibility.

Transparency is not a panacea. In isolation, more information about foundation models will not necessarily produce a more just or equitable digital world. But if transparency is implemented through engagement with third-party experts, independent auditors, and communities who are directly affected by digital technologies, it can help ensure that foundation models benefit society.

**Transparency in practice for prior digital technologies**  Digital technologies are marked by a long track record of poor transparency. While each major new technology has dramatically restructured society, the powerful corporations that build these technologies have wielded outsized influence and maintained opacity to advance their commercial interests. Consider the following examples of digital technologies that suffer from a lack of transparency as well as associated interventions/studies to reduce opacity: the fight for net neutrality for internet service providers like Comcast (Service, 2021), web cookies for online advertising like Google Ads (Englehardt et al., 2015; Englehardt & Narayanan, 2016; Narayanan & Reisman, 2017), labor practices for crowd-sourcing platforms like Amazon Mechnical Turk (Gray & Suri, 2019a; Crawford, 2021), wage schemes for ride sharing platforms like Uber (Rosenblat & Stark, 2016), and dark patterns for game companies like Epic Games (Commission, 2023).

Stepping through these examples, efforts like the Princeton Web Transparency Project (Englehardt et al., 2015; Englehardt & Narayanan, 2016; Narayanan & Reisman, 2017) have unveiled the ecosystem of online third-party tracking using cookies, which "led to greater public awareness, the cessation of some privacy-infringing practices, and the creation of new consumer privacy tools." Similarly, Rosenblat & Stark (2016) empirically demonstrated that Uber drivers were the subject of a severely asymmetric power dynamic given the control exerted by Uber over their drivers, to the detriment of the ride sharing market. In the context of crowd-sourcing, Gray & Suri (2019a) and Crawford (2021) demonstrated exploitation of the "ghost" workers powering AI, such as on Amazon Mechanical Turk, that was made invisible on these platforms. More recently, these efforts have prompted the scrutiny of lawmakers as to improve transparency and, thereby, labor conditions. As a final example, dark patterns have a pervasive practice for myriad technologies, leading to mismanaged consumer expectations and overall opacity. To this end, the FTC's recent inquiry into Epic Games for dark patterns used to deceive gamers, and particularly children, amounted to a $245M fine on Epic Games (Commission, 2023).

Building on these prior examples, we consider social media more specifically. Social media platforms provide a vivid example of transparency challenges in recent years, and the increasing level of acknowledgement among some technology companies that a baseline level of transparency is a necessity. Given the profound impact of social media in mediating how humans form relationships, communicate with each other, buy goods and services, and access information, a broad body of work argues for greater transparency (see Keller, 2022). Social media platforms have slowly begun to adopt transparency reporting practices. For example, Facebook now hosts its own Ad Library[31], Content Library[32], and a transparency center[33] that reports on content enforcement, widely viewed content, regulatory transparency, government data requests, and intellectual property, among other pieces of mostly voluntary transparency. In parallel, transparency requirements have

---

[31]https://www.facebook.com/ads/library/
[32]https://transparency.fb.com/researchtools/meta-content-library
[33]https://transparency.fb.com/

been enshrined in laws like the EU Digital Services Act (Commission, 2022) and legislative proposals like the U.S. Platform Accountability and Transparency Act (Coons et al., 2021).

**Transparency for AI.** With the rise of AI in the past 10 years, its societal impact has received much greater attention (Barocas & Selbst, 2016; Abebe et al., 2020; Hutchinson et al., 2021; Bender et al., 2021). Transparency is often referenced as a core ethical principle undergirding responsible AI (Fjeld et al., 2020; Hagendorff, 2020).[34] Jobin et al. (2019) find that transparency is the most frequently cited principle in AI ethics guidelines, appearing in 85% of the assessed 84 guidelines.

Given that the standard machine learning pipeline is divided into several stages, transparency efforts often target different stages.[35] Documentation efforts are most common at the level of data (Gebru et al., 2021; Bender & Friedman, 2018b; Pushkarna et al., 2022) and models (Mitchell et al., 2018; Crisan et al., 2022), with evaluations providing further insight into models (Deng et al., 2009; Ribeiro et al., 2020; Perez et al., 2022; Liang et al., 2022c; Bommasani et al., 2023c). More recently, several efforts have studied the broader ecosystem-wide transparency of AI and its supply chains (Bommasani et al., 2023b; Cen et al., 2023), though transparency on the downstream impacts of AI is comparatively understudied (Narayanan & Kapoor, 2023). The Foundation Model Transparency Index advances this view, assessing transparency of foundation models with a comprehensive ecosystem-level approach that spans the data and broader upstream resources, the foundation models themselves, and the downstream use and impact.

## 2.3 Indexes

A (composite) index is a standard methodology (OECD et al., 2008; Greco et al., 2019) for assessing entities (e.g. companies, countries) in relation to a specific construct (e.g. transparency, responsibility). Methodologically, the score on an index for a specific entity is the aggregate of multiple low-level indicators that can be more directly quantified. Composite indexes as a methodology has seen broad adoption across the social sciences, including to directly address major political, economic, and societal concerns such as public corruption (e.g. Transparency International's Corruption Perceptions Index; Transparency International, 2023), environmental welfare (e.g. the World Economic Forum's Environmental Sustainability Index; Whitford & Wong, 2009) and living standards (e.g. the United Nations Development Programme's Human Development Index; Hopkins, 1991). However, indexes have not played a major role in mainstream AI discourse.[36]

Indexes are designed to further several objectives and have certain characteristic strengths (Commission et al., 2008; Saisana & Tarantola, 2002). Most fundamentally, indexes can transform complex and amorphous constructs into straightforward and concrete scores. Indexes and the aggregate quantitative metrics they provide can therefore allow for broad engagement on certain topics, furthering public understanding as well as providing a strong basis for various forms of decision-making such as regulatory intervention. In addition, when indexes are maintained over time, they encourage a long-term focus and can be vital in fostering improvement over time (Kogen, 2022). In this way, while operating at a different level of abstraction and involving a different set of design decisions, indexes are analogous to model benchmarks that are commonplace in AI (Deng et al., 2009; Wang et al., 2019; Liang et al., 2023) and appeal to a similar theory of change (Donoho, 2017; Ethayarajh & Jurafsky, 2020; Raji et al., 2021; Bommasani, 2023). Indexes also have shortcomings: namely, they can be reductive and overly subjective (Saisana & Tarantola, 2002; OECD

---

[34]See UNESCO's Recommendation on the Ethics of Artificial Intelligence, which was adopted by its 193 member states and constitutes the first global normative instrument on AI ethics. Our conceptualization of transparency covers several of UNESCO's 10 principles, namely Transparency and Explainability. See `https://www.unesco.org/en/artificial-intelligence/recommendation-ethics`

[35]As mentioned previously, the term "transparency" is also sometimes used in AI to refer to explainability/interpretability, referring to understanding how a specific model makes predictions (Zou et al., 2023). In part, the emphasis on this topic is due to the inscrutability of the deep neural networks that have powered AI's rise. However, we focus on structural forms of transparency, taking a more macroscopic perspective.

[36]We highlight the AI Index from the Stanford Institute for Human-Centered AI (Maslej et al., 2023; Zhang et al., 2022a), which tracks global progress of AI across a variety of quantitative indicators. In contrast to the composite indexes here, the AI Index neither directly scores specific entities nor does it aggregate individual indicators into a singular aggregate. We also highlight the Generative AI Accountability Scorecard from Ranking Digital Rights as a forthcoming effort that targets the generative AI services downstream of foundation models: `https://rankingdigitalrights.org/mini-report/introducing-rdrs-preliminary-standards-for-generative-ai/`.

et al., 2008; Greco et al., 2019). To design and score an index, researchers must make simplifying decisions about which indicators to include, how to weigh those indicators, and how to grade indicators. Beyond these methodological issues, indexes are subject to a broader conceptual critique that they may oversimplify concepts that are intrinsically complex, discarding valuable nuances.[37] Indexes may also be subject to gaming, which we discuss more extensively in §9.2: LIMITATIONS.

---

[37]The literature and theory on composite indexes is much too extensive to be easily summarized in this brief primer. We recommend the Handbook on Constructing Composite Indicators: Methodology and User Guide (OECD et al., 2008) as a proper introduction to the subject: `https://doi.org/10.1787/9789264043466-en`.

## 3   The Foundation Model Transparency Index

The Foundation Model Transparency Index scores foundation model developers for their comprehensive transparency. We discuss specifics on the developers, indicators, and scoring in subsequent sections. Strategically, our aim is for the index to clarify discourse on foundation models and AI that is muddled and lacks grounding in empirical data. We aim to improve the overall transparency of the AI ecosystem by encouraging foundation model developers to share more information about the development and deployment of their models. We also provide a clear taxonomization of the key issues related to transparency and demonstrate where greater transparency would be especially valuable. Therefore, the Foundation Model Transparency Index provides a frame of reference for assessing whether the ecosystem as a whole—and which developers in particular—become more or less transparency over time. Simultaneously, given the limitations of indexes, we are fully transparent about our methodology, including the core decisions on indicator inclusion, indicator weighting, and indicator scoring. We also discuss methodological shortcomings relating to each of these decisions in §9.2: LIMITATIONS. To guard against unnecessary simplification, we provide discussion and analysis at several levels of abstraction in §7: RESULTS.

Overall, the Foundation Model Transparency Index captures the key dimensions of transparency that are relevant to foundation models at present. As the foundation model ecosystem and AI policy evolves over time, the central questions regarding the transparency of foundation models will evolve as well. Consequently, we will conduct future versions of the index that adjust the indicators to reflect these changes. We more expansively discuss our intended impact (including our theory of change and associated limitations and risks) in §9: IMPACT.

## 4 Indicators

We define 100 indicators that comprehensively characterize transparency for foundation model developers. To select these indicators, we compiled relevant concepts raised across past scientific literature as well as concerns animated by public discourse on foundation models and other digital technologies. In Appendix B we provide specific references for each indicator, and these references advocate for increased transparency and information sharing related to the indicator in question. We derived a concrete set of indicators from this literature, engaging external researchers to converge on the final list of 100 (see Figure 2). These indicators cover each dimension of the foundation model supply chain, from the data, compute, and labor required to build foundation models to model evaluations and developers' policies to restrict their use. We divide our indicators into three broad domains as described in Figure 1: indicators that are *upstream* of the model, indicators that relate to the *model* itself, and indicators that are *downstream* of the model.

### 4.1 Upstream indicators

The upstream indicators identify the *ingredients and processes* involved in building a foundation model. There are 32 upstream indicators, which we further taxonomize into the following 6 subdomains:

- **Data (10 indicators).** Assesses transparency regarding the size and composition of the data used to build the model; the creators whose content is present in the data; and any steps to curate or augment the data. These indicators also address transparency regarding the inclusion of personal, copyrighted, or licensed data.

- **Data Labor (7 indicators).** Assesses transparency regarding the use of human labor in producing the data used to build the model, including the wages, labor protections, employer, and geographic distribution of workers who contributed to data annotation and curation. These indicators also address transparency regarding the third parties that foundation model developers partnered with to construct their models.

- **Data Access (2 indicators).** Assesses the scope of data access given to external parties.

- **Compute (7 indicators).** Assesses transparency regarding the hardware and computation used to build the model, as well as the resulting energy use and environmental impacts.

- **Methods (4 indicators).** Assesses basic technical specifications for the model's training stages and objectives, as well as the software frameworks and dependencies used.

- **Data Mitigations (2 indicators).** Assesses transparency regarding steps taken to mitigate data privacy and copyright concerns.

We depict the upstream indicators in Figure 3. Researchers have widely advocated for greater transparency in relation to Data and Data Access (Bender & Friedman, 2018a; Gebru et al., 2018; Hutchinson et al., 2021; Dodge et al., 2021; Bandy & Vincent, 2021) as a means for contextualizing model capabilities (Sambasivan et al., 2021; Longpre et al., 2023) and risks related to privacy, bias, and copyright (Buolamwini & Gebru, 2018; Bender et al., 2021; Kandpal et al., 2022; Sobel, 2017). Data Labor indicators uplift concerns related to labor practices, include irresponsible or exploitative use of human labor (Gray & Suri, 2019a; Crawford, 2021; Hao & Seetharaman, 2023; Kittur et al., 2013; Dzieza, 2023; West, 2019). Compute indicators relate to concerns around the high computational cost and energy expenditure associated with building foundation models, which can result in environmental harm (Lacoste et al., 2019; Strubell et al., 2019; Schwartz et al., 2020; Patterson et al., 2021; Bender et al., 2021; Henderson et al., 2020; Luccioni & Hernández-García, 2023; Vipra & West, 2023). Data Mitigations indicators also relate to the growing legal and sociotechnical concerns over data privacy, copyright, and licensing (Henderson et al., 2023; Brown et al., 2022; Lee et al., 2023a; Cooper et al., 2023; Saveri et al., 2023).

**2023 Foundation Model Transparency Index Indicators**

| Upstream | Model | Downstream |
|---|---|---|
| Data size | Input modality | Release decision-making |
| Data sources | Output modality | Release process |
| Data creators | Model components | Distribution channels |
| Data source selection | Model size | Products and services |
| Data curation | Model architecture | Detection of machine-generated content |
| Data augmentation | Centralized model documentation | Model License |
| Harmful data filtration | External model access protocol | Terms of service |
| Copyrighted data | Blackbox external model access | Permitted and prohibited users |
| Data license | Full external model access | Permitted, restricted, and prohibited uses |
| Personal information in data | Capabilities description | Usage policy enforcement |
| Use of human labor | Capabilities demonstration | Justification for enforcement action |
| Employment of data laborers | Evaluation of capabilities | Usage policy violation appeals mechanism |
| Geographic distribution of data laborers | External reproducibility of capabilities evaluation | Permitted, restricted, and prohibited model behaviors |
| Wages | Third party capabilities evaluation | Model behavior policy enforcement |
| Instructions for creating data | Limitations description | Interoperability of usage and model behavior policies |
| Labor protections | Limitations demonstration | User interaction with AI system |
| Third party partners | Third party evaluation of limitations | Usage disclaimers |
| Queryable external data access | Risks description | User data protection policy |
| Direct external data access | Risks demonstration | Permitted and prohibited use of user data |
| Compute usage | Unintentional harm evaluation | Usage data access protocol |
| Development duration | External reproducibility of unintentional harm evaluation | Versioning protocol |
| Compute hardware | Intentional harm evaluation | Change log |
| Hardware owner | External reproducibility of intentional harm evaluation | Deprecation policy |
| Energy usage | Third party risks evaluation | Feedback mechanism |
| Carbon emissions | Mitigations description | Feedback summary |
| Broader environmental impact | Mitigations demonstration | Government inquiries |
| Model stages | Mitigations evaluation | Monitoring mechanism |
| Model objectives | External reproducibility of mitigations evaluation | Downstream applications |
| Core frameworks | Third party mitigations evaluation | Affected market sectors |
| Additional dependencies | Trustworthiness evaluation | Affected individuals |
| Mitigations for privacy | External reproducibility of trustworthiness evaluation | Usage reports |
| Mitigations for copyright | Inference duration evaluation | Geographic statistics |
| | Inference compute evaluation | Redress mechanism |
| | | Centralized documentation for downstream use |
| | | Documentation for responsible downstream use |

Figure 2: **Indicators.** The 100 indicators of the Foundation Model Transparency Index spanning the 3 domains: upstream, model, and downstream.

.

**Upstream Indicators for the 2023 Foundation Model Transparency Index**

| Upstream |
| --- |
| **Data size:** For the data used in building the model, is the data size disclosed? |
| **Data sources:** For all data used in building the model, are the data sources disclosed? |
| **Data creators:** For all data used in building the model, is there some characterization of the people who created the data? |
| **Data source selection:** Are the selection protocols for including and excluding data sources disclosed? |
| **Data curation:** For all data sources, are the curation protocols for those data sources disclosed? |
| **Data augmentation:** Are any steps the developer takes to augment its data sources disclosed? |
| **Harmful data filtration:** If data is filtered to remove harmful content, is there a description of the associated filter? |
| **Copyrighted data:** For all data used in building the model, is the associated copyright status disclosed? |
| **Data license:** For all data used in building the model, is the associated license status disclosed? |
| **Personal information in data:** For all data used in building the model, is the inclusion or exclusion of personal information in that data disclosed? |
| **Use of human labor:** Are the phases of the data pipeline where human labor is involved disclosed? |
| **Employment of data laborers:** Is the organization that directly employs the people involved in data labor disclosed for each phase of the data pipeline? |
| **Geographic distribution of data laborers:** Is geographic information regarding the people involved in data labor disclosed for each phase of the data pipeline? |
| **Wages:** Are the wages for people who perform data labor disclosed? |
| **Instructions for creating data:** Are the instructions given to people who perform data labor disclosed? |
| **Labor protections:** Are the labor protections for people who perform data labor disclosed? |
| **Third party partners:** Are the third parties who were or are involved in the development of the model disclosed? |
| **Queryable external data access:** Are external entities provided with queryable access to the data used to build the model? |
| **Direct external data access:** Are external entities provided with direct access to the data used to build the model? |
| **Compute usage:** Is the compute required for building the model disclosed? |
| **Development duration:** Is the amount of time required to build the model disclosed? |
| **Compute hardware:** For the primary hardware used to build the model, is the amount and type of hardware disclosed? |
| **Hardware owner:** For the primary hardware used in building the model, is the owner of the hardware disclosed? |
| **Energy usage:** Is the amount of energy expended in building the model disclosed? |
| **Carbon emissions:** Is the amount of carbon emitted (associated with the energy used) in building the model disclosed? |
| **Broader environmental impact:** Are any broader environmental impacts from building the model besides carbon emissions disclosed? |
| **Model stages:** Are all stages in the model development process disclosed? |
| **Model objectives:** For all stages that are described, is there a clear description of the associated learning objectives or a clear characterization of the nature of this update to the model? |
| **Core frameworks:** Are the core frameworks used for model development disclosed? |
| **Additional dependencies:** Are any dependencies required to build the model disclosed besides data, compute, and code? |
| **Mitigations for privacy:** Are any steps the developer takes to mitigate the presence of PII in the data disclosed? |
| **Mitigations for copyright:** Are any steps the developer takes to mitigate the presence of copyrighted information in the data disclosed? |

Figure 3: **Upstream Indicators.** The 32 upstream indicators that span Data, Data Labor, Data Access, Compute, Methods, and Data Mitigations.

## 4.2 Model indicators

The model indicators identify the *properties and function* of the foundation model. There are 33 model indicators, which we further taxonomize into the following 8 subdomains:

- **Model Basics (6 indicators).** Assesses transparency regarding fundamental information about the model such as modalities, size, and architecture as well as the presence of centralized model documentation.

- **Model Access (3 indicators).** Assesses the scope of model access given to external entities.

- **Capabilities (5 indicators).** Assesses transparency regarding the capabilities of the model, including evaluations.

- **Limitations (3 indicators).** Assesses transparency regarding the limitations of the model, including evaluations.

- **Risks (7 indicators).** Assesses transparency regarding the risks of the model, including evaluations, with specific focus on both unintentional harm (e.g. bias) and intentional harm (e.g. fraud).

- **Model Mitigations (5 indicators).** Assesses transparency regarding model-level mitigations, including evaluations of their efficacy.

- **Trustworthiness (2 indicators).** Assesses transparency regarding the trustworthiness of the model, including evaluations.

- **Inference (2 indicators).** Assesses transparency regarding standardized inference with the model.

We depict the model indicators in Figure 4. Model Basics indicators refer to fundamental information that is expected by model documentation standards (Mitchell et al., 2019; Crisan et al., 2022; Bommasani et al., 2023b) and, historically, have been reliably reported in the release of machine learning models. Model Access indicators reflect literature tied to the spectrum of model release and the associated differences in external access (Solaiman et al., 2019; Sastry, 2021; Shevlane, 2022; Liang et al., 2022b; Solaiman, 2023). The indicators on Capabilities, Limitations, Risks and Model Mitigations are motivated by a common understanding that these factors jointly influence the societal impact of machine learning models and AI systems (Tabassi, 2023b; Weidinger et al., 2023). For these subdomains, the description and demonstration indicators gauge whether there is some non-technical articulation and legibility of these concepts, primed by concerns surrounding public understanding of foundation models.[38] To make these assessments more rigorous, the evaluation indicators build on the extensive tradition of evaluation in AI spanning iconic benchmarks like ImageNet (Deng et al., 2009), broader benchmarks like SuperGLUE Wang et al. (2019), and extensive meta-benchmarks like LM-Harness, BIG-bench, HELM and BEHAVIOR (Gao et al., 2021b; Srivastava et al., 2022; Liang et al., 2023; Srivastava et al., 2021). Indicators assessing evaluations also highlight the importance of reproducibility (Lipton & Steinhardt, 2019; Kapoor et al., 2023; Kapoor & Narayanan, 2023)[39] and independent assessment (Sandvig et al., 2014; Raji & Buolamwini, 2019; Metaxa et al., 2021; Costanza-Chock et al., 2022; Raji et al., 2022b; Raji, 2022; Lam et al., 2022; Weidinger et al., 2023), which enable open science and external verification of developers' claims about their models. In the case of risks, finer distinctions between unintentional harms (e.g. biases, toxicity) and intentional harms (e.g. disinformation, fraud) build on harm taxonomies (Bender et al., 2021; Bommasani et al., 2021; Weidinger et al., 2021; Tabassi, 2023a; Weidinger et al., 2023). Indicators on trustworthiness and inference are especially motivated by the Trustworthy ML Initiative[40] and MLPerf (Reddi et al., 2020) respectively, among other works (Brundage et al., 2020; Cammarota et al., 2020; Kumar et al., 2020; Liu et al., 2022; Shneiderman, 2020; Patterson et al., 2021; Narayanan et al., 2023).

---

[38]See `https://www.gov.uk/government/publications/public-perceptions-towards-the-use-of-foundation-models-in-the-public-sector`.

[39]See the ML Reproducibility challenge: `https://paperswithcode.com/rc2022`, CodaLab worksheets for reproducible ML: `https://worksheets.codalab.org/`, and Joelle Pineau's reproducibility checklist: `https://www.cs.mcgill.ca/~jpineau/ReproducibilityChecklist.pdf`.

[40]`https://www.trustworthyml.org/`

**Model Indicators for the 2023 Foundation Model Transparency Index**

| Model |
|---|
| **Input modality:** Are the input modalities for the model disclosed? |
| **Output modality:** Are the output modalities for the model disclosed? |
| **Model components:** Are all components of the model disclosed? |
| **Model size:** For all components of the model, is the associated model size disclosed? |
| **Model architecture:** Is the model architecture disclosed? |
| **Centralized model documentation:** Is key information about the model included in a centralized artifact such as a model card? |
| **External model access protocol:** Is a protocol for granting external entities access to the model disclosed? |
| **Blackbox external model access:** Is black box model access provided to external entities? |
| **Full external model access:** Is full model access provided to external entities? |
| **Capabilities description:** Are the model's capabilities described? |
| **Capabilities demonstration:** Are the model's capabilities demonstrated? |
| **Evaluation of capabilities:** Are the model's capabilities rigorously evaluated, with the results of these evaluations reported prior to or concurrent with the initial release of the model? |
| **External reproducibility of capabilities evaluation:** Are the evaluations of the model's capabilities reproducible by external entities? |
| **Third party capabilities evaluation:** Are the model's capabilities evaluated by third parties? |
| **Limitations description:** Are the model's limitations disclosed? |
| **Limitations demonstration:** Are the model's limitations demonstrated? |
| **Third party evaluation of limitations:** Can the model's limitations be evaluated by third parties? |
| **Risks description:** Are the model's risks disclosed? |
| **Risks demonstration:** Are the model's risks demonstrated? |
| **Unintentional harm evaluation:** Are the model's risks related to unintentional harm rigorously evaluated, with the results of these evaluations reported prior to or concurrent with the initial release of the model? |
| **External reproducibility of unintentional harm evaluation:** Are the evaluations of the model's risks related to unintentional harm reproducible by external entities? |
| **Intentional harm evaluation:** Are the model's risks related to intentional harm rigorously evaluated, with the results of these evaluations reported prior to or concurrent with the initial release of the model?. |
| **External reproducibility of intentional harm evaluation:** Are the evaluations of the model's risks related to intentional harm reproducible by external entities? |
| **Third party risks evaluation:** Are the model's risks evaluated by third parties? |
| **Mitigations description:** Are the model mitigations disclosed? |
| **Mitigations demonstration:** Are the model mitigations demonstrated? |
| **Mitigations evaluation:** Are the model mitigations rigorously evaluated, with the results of these evaluations reported? |
| **External reproducibility of mitigations evaluation:** Are the model mitigation evaluations reproducible by external entities? |
| **Third party mitigations evaluation:** Can the model mitigations be evaluated by third parties? |
| **Trustworthiness evaluation:** Is the trustworthiness of the model rigorously evaluated, with the results of these evaluations disclosed? |
| **External reproducibility of trustworthiness evaluation:** Are the trustworthiness evaluations reproducible by external entities? |
| **Inference duration evaluation:** Is the time required for model inference disclosed for a clearly-specified task on a clearly-specified set of hardware? |
| **Inference compute evaluation:** Is the compute usage for model inference disclosed for a clearly-specified task on a clearly-specified set of hardware? |

Figure 4: **Model Indicators.** The 33 model indicators that span Model Basics, Model Access, Capabilities, Limitations, Risks, Model Mitigations, Trustworthiness, and Inference.

### 4.3 Downstream indicators

The downstream indicators identify the *use* of the foundation model, including details about its *release.* There are 35 downstream indicators, which we further taxonomize into the following 9 subdomains:

- **Distribution (7 indicators).** Assesses transparency regarding the release process, the distribution channels for the model, and the products and services that arise through internal use. Additionally, this subdomain assesses the presence of model licenses, terms of service, and mechanisms for detecting model-generated content.

- **Usage Policy (5 indicators).** Assesses transparency regarding the developer's acceptable use policy such as restrictions on specific uses or users, as well as transparency regarding how it enforces such policies.

- **Model Behavior Policy (3 indicators).** Assesses transparency regarding the developer's policy on acceptable and unacceptable model behavior as well as transparency regarding enforcement of this policy and expectations in the event of usage policy violations.

- **User Interface (2 indicators).** Assesses transparency in the user interface for the developer's flagship distribution channel, if the channel includes a user interface.

- **User Data Protection (3 indicators).** Assesses transparency regarding the developer's policies with respect to user data protection, such as how data is stored, shared, and accessed.

- **Model Updates (3 indicators).** Assesses transparency regarding the developer's versioning protocol, change log, and deprecation policy.

- **Feedback (3 indicators).** Assesses transparency regarding mechanisms for reporting feedback on the model, summaries of feedback received, and related government inquiries.

- **Impact (7 indicators).** Assesses transparency regarding the downstream impact of the model on society, such as affected market sectors, individuals, and geographies. Additionally, this subdomain assesses transparency regarding downstream applications, usage statistics, and mechanisms for monitoring usage as well as providing redress in the event of harm to users.

- **Downstream Documentation (2 indicators).** Assesses the presence of centralized documentation for downstream use and documentation for responsible downstream use.

We depict the downstream indicators in Figure 5. Given that foundation models are the basis for a downstream supply chain (Bommasani et al., 2021), the distribution indicators are informed by the literature on AI supply chains (Bommasani et al., 2023b; Vipra & Korinek, 2023; Cen et al., 2023; Cobbe et al., 2023; Widder & Wong, 2023; Brown, 2023) and release practices (Liang, 2022; Solaiman, 2023; Henderson et al., 2023; Kirchenbauer et al., 2023; Kuditipudi et al., 2023; Liesenfeld et al., 2023). Usage policy indicators draw from company publications on responsible model deployment Cohere (2022) as well precedents from social media. Model behavior policy indicators are rooted in literature that discusses AI behavior and trustworthiness, risks, mitigation and refusal Kumar et al. (2022); Weidinger et al. (2021); Brundage et al. (2020); Cammarota et al. (2020); Kumar et al. (2020); Liu et al. (2022); Reuter & Schulze (2023). User interface indicators are derived from research on safety by design and human-centered user interfaces Wang et al. (2023b); Nakao et al. (2022). User data protection indicators are inspired by policy recommendations on user data minimization, privacy, preservation, protection and contextual integrity EU (2016); Brown et al. (2022); Vipra & Myers West (2023); Winograd (2023); Nissenbaum (2024); King (2020); Mulligan et al. (2016). Model updates indicators stem from work focused on adequately updating systems and version control of AI systems (Sathyavageesran et al., 2022; Hashesh, 2023; Chen et al., 2023b). For feedback, impact and downstream documentation, the indicators were motivated by the literature on algorithmic auditing Liang (2022); Solaiman (2023); Raji et al. (2022b) as well as transparency reporting practices for social media.[41]

---

[41]See `https://www.tspa.org/curriculum/ts-fundamentals/transparency-report/`, `https://transparencyreport.google.com/` and `https://transparency.fb.com/reports/`.

**Downstream Indicators for the 2023 Foundation Model Transparency Index**

| Downstream |
| --- |
| **Release decision-making:** Is the developer's protocol for deciding whether or not to release a model disclosed? |
| **Release process:** Is a description of the process of how the model was released disclosed? |
| **Distribution channels:** Are all distribution channels disclosed? |
| **Products and services:** Does the developer disclose whether any products and services offered by the developer are dependent on the model? |
| **Detection of machine-generated content:** Are any mechanisms for detecting content generated by this model disclosed? |
| **Model License:** Is a license for the model disclosed? |
| **Terms of service:** Are terms of service disclosed for each distribution channel? |
| **Permitted and prohibited users:** Is a description of who can and cannot use the model disclosed? |
| **Permitted, restricted, and prohibited uses:** Are permitted, restricted, and prohibited uses of the model disclosed? |
| **Usage policy enforcement:** Is the enforcement protocol for the usage policy disclosed? |
| **Justification for enforcement action:** Do users receive a justification when they are subject to an enforcement action for violating the usage policy? |
| **Usage policy violation appeals mechanism:** Is a mechanism for appealing potential usage policy violations disclosed? |
| **Permitted, restricted, and prohibited model behaviors:** Are model behaviors that are permitted, restricted, and prohibited disclosed? |
| **Model behavior policy enforcement:** Is the enforcement protocol for the model behavior policy disclosed? |
| **Interoperability of usage and model behavior policies:** Is the way that the usage policy and the model behavior policy interoperate disclosed? |
| **User interaction with AI system:** For distribution channels with user-facing interfaces, are users notified (i) that they are interacting with an AI system, (ii) of the specific foundation model they are interacting with, and (iii) that outputs are machine-generated? |
| **Usage disclaimers:** For distribution channels with user-facing interfaces, are users provided with disclaimers involving model use? |
| **User data protection policy:** Are the protocols for how the developer stores, accesses, and shares user data disclosed? |
| **Permitted and prohibited use of user data:** Are permitted and prohibited uses of user data disclosed? |
| **Usage data access protocol:** Is a protocol for granting external entities access to usage data disclosed? |
| **Versioning protocol:** Is there a disclosed version and versioning protocol for the model? |
| **Change log:** Is there a disclosed change log for the model? |
| **Deprecation policy:** Is there a disclosed deprecation policy for the developer? |
| **Feedback mechanism:** Is a feedback mechanism disclosed? |
| **Feedback summary:** Is a report or summary disclosed regarding the feedback the developer received or, alternatively, the way the developer responded to that feedback? |
| **Government inquiries:** Is a summary of government inquiries related to the model received by the developer disclosed? |
| **Monitoring mechanism:** For each distribution channel, is a monitoring mechanism for tracking model use disclosed? |
| **Downstream applications:** Across all forms of downstream use, is the number of applications dependent on the foundation model disclosed? |
| **Affected market sectors:** Across all downstream applications, is the fraction of applications corresponding to each market sector disclosed? |
| **Affected individuals:** Across all forms of downstream use, is the number of individuals affected by the foundation model disclosed? |
| **Usage reports:** Is a usage report that gives usage statistics describing the impact of the model on users disclosed? |
| **Geographic statistics:** Across all forms of downstream use, are statistics of model usage across geographies disclosed? |
| **Redress mechanism:** Is any mechanism to provide redress to users for harm disclosed? |
| **Centralized documentation for downstream use:** Is documentation for downstream use centralized in a centralized artifact? |
| **Documentation for responsible downstream use:** Is documentation for responsible downstream use disclosed? |

Figure 5: **Downstream Indicators.** The 35 downstream indicators that span Distribution, Usage Policy, Model Behavior Policy, User Interface, User Data Protection, Model Updates, Feedback, Impact, and Downstream Documentation.

**Note on assessment of indicators.** We assess each indicator based on the information that developers share publicly about their flagship foundation models and their practices that apply to these models. Our standard for awarding points on an indicator is that the developer must explicitly state the information related to the indicator in its documentation, or it must explicitly point to the information in its documentation. This implies that if developers are overly vague or do not link to a key external document for a particular indicator then they do not receive a point. In addition, if developers explicitly state in their documentation that they *do not* carry out a specific action related to an indicator (e.g. they do not have a mechanism for users to provide feedback) then we generally award a point for that indicator. We note that this is exceedingly rare and that, in general, developers share little information about the actions they do or do not take in the process of developing and deploying foundation models.

**Note on inclusion of deployment.** Our view of transparency is expansive, considering the broader supply chain beyond just foundation models. As we discuss in §2.2: BACKGROUND-TRANSPARENCY, existing conceptualizations of transparency in AI often consider upstream resources (especially data) in addition to machine learning models. But these works and broader public discourse usually do not foreground the downstream use and impact of AI, even though this is the most direct way in which AI affects society. To this end, we include the entire downstream domain to bring greater attention to this vital topic.

In particular, while we are assessing foundation model developers, we assess them in relation to distribution channels and other factors that determine their downstream impact. At present, we recognize that characterizing the downstream impact of foundation models may be challenging, especially for open model developers. By releasing a model openly, developers may cede the ability to easily monitor the model's downstream use and impact. Open model developers can be fully transparent by being clear about the ways in which they do or do not monitor downstream use and impact. In addition, we believe in the potential for greater coordination between foundation model developers and distribution channels to increase transparency; for example, distribution channels could supply information about how the model is used to the foundation model developer. Partnerships with distribution channels that promote transparency provide a promising means for all foundation model developers to share more information about the impact their models have on society.

# 5 Foundation model developers

| Name | Flagship | Release | Input | Output | Status | Headquarters | WH1 | WH2 | WH3 | FMF |
|------|----------|---------|-------|--------|--------|--------------|-----|-----|-----|-----|
| AI21 Labs | Jurassic-2 | API | Text | Text | Startup | Tel Aviv, Israel | ✗ | ✗ | ✗ | ✗ |
| Amazon | Titan Text | API | Text | Text | Big Tech | Seattle, USA | ✗ | ✓ | ✗ | ✗ |
| Anthropic | Claude 2 | API | Text | Text | Startup | San Francisco, USA | ✓ | ✓ | ✗ | ✓ |
| Cohere | Command | API | Text | Text | Startup | Toronto, Canada | ✓ | ✗ | ✓ | ✗ |
| Google | PaLM 2 | API | Text | Text | Big Tech | Mountain View, USA | ✓ | ✓ | ✗ | ✓ |
| Hugging Face | BLOOMZ | Open weights, open data | Text | Text | Startup | Brooklyn, USA | ✓ | ✗ | ✗ | ✗ |
| Inflection | Inflection-1 | No access (API forthcoming) | Text | Text | Startup | Palo Alto, USA | ✗ | ✓ | ✗ | ✗ |
| Meta | Llama 2 | Open weights | Text | Text | Big Tech | Menlo Park, USA | ✗ | ✓ | ✗ | ✗ |
| OpenAI | GPT-4 | API | Text, Images | Text | Startup | San Francisco, USA | ✓ | ✓ | ✗ | ✓ |
| Stability AI | Stable Diffusion 2 | Open weights, open data | Text | Images | Startup | London, UK | ✓ | ✗ | ✓ | ✗ |

Table 1: **Selected foundation model developers.** Information on the 10 selected foundation model developers: the developer name, their flagship model, the release strategy for the model (see Figure 6), the input and output modalities for the model, the developer's status as either Big Tech or Startup, and the developer's headquarters. We note which of the developers were involved in the White House's initiative for public evaluation of AI systems announced in May 2023 (WH1), voluntary commitments for the management of risks posed by AI announced in July 2023 (WH2), and commitments by additional organizations on the same matters of risks by AI announced in September 2023 (WH3). Additionally, we note which of the developers are founding members of the Frontier Model Forum, announced in July 2023.

Transparency initiatives in AI (e.g. datasheets and model cards) often introduce frameworks that support machine learning developers in achieving greater transparency in their own work. In contrast, we proactively assess foundation model developers for their transparency using the 100 indicators we specify. By conducting the assessment ourselves, we sidestep concerns of uneven uptake that have arisen with past transparency initiatives (e.g. Gebru et al., 2018; Mitchell et al., 2018) and provide greater consistency in the scoring of each indicator across developers. Most importantly, scoring many developers allows for the comparison of their scores, which provides a rich context for how to improve transparency in the foundation model ecosystem.

Efforts like Ecosystem Graphs (Bommasani et al., 2023b) and the UK Competition and Markets Authority (CMA) report on the foundation model market[42] track the organizations that develop foundation models. At the time of writing in September 2023, the CMA report documented 160 foundation models (based on data drawn from Ecosystem Graphs) built by more than 50 organizations.[43] However, as the CMA report states, a small number of developers control the majority of the market at present (Vipra & Korinek, 2023). Due to this intense level of market concentration, we decided to assess 10 major foundation model developers.

---

[42]https://www.gov.uk/government/publications/ai-foundation-models-initial-report
[43]https://assets.publishing.service.gov.uk/government/uploads/system/uploads/attachment_data/file/1185508/Full_report_.pdf#page=22

### 5.1 Selecting developers

We considered a variety of selection criteria in choosing the 10 developers to assess, arriving at the following three principles:

1. **Impact.** We selected developers that have built the most influential foundation models.

2. **Diversity.** We selected developers that, when considered collectively, represent many axes of variation in the foundation model ecosystem. For example, developers that release models along different points on the release gradient (e.g. open vs. closed, Solaiman, 2023), build models with different modalities (e.g. text-to-text vs. text-to-image), and occupy different positions in the market (e.g. startups vs. Big Tech).

3. **Companies.** We selected developers that are established companies as enduring targets for longitudinal improvement. This to some extent parallels current regulatory initiatives that explicitly focus on companies as the target of policy for foundation models.[44]

On this basis, we chose 10 companies that all are influential foundation model developers: AI21 Labs, Amazon, Anthropic, Cohere, Google, Hugging Face, Inflection, Meta, OpenAI, and Stability AI. These 10 provide significant diversity in terms of release strategy (e.g. Anthropic, Meta, and Hugging Face all release flagship models with different levels of openness; see Figure 6), modality (e.g. Cohere, OpenAI, and Stability AI all provide different input-output modalities), and market position (e.g. Google, Inflection, and OpenAI occupy different market positions).

Additionally, in parallel to our research, the White House made three announcements involving companies that develop foundation models: a red-teaming exercise announced in May 2023,[45] a set of voluntary commitments announced in July 2023,[46] and another set of voluntary commitments announced in September 2023.[47] Separately, three of the companies we assess jointly announced the formation of the Frontier Model Forum in July 2023.[48] When taken together, these announcements name 16 companies: Adobe, Amazon, Anthropic, Cohere, Google, Hugging Face, IBM, Inflection, Meta, Microsoft, NVIDIA, OpenAI, Palantir, Salesforce, Scale AI, and Stability AI. We note that 9 of the 10 companies we selected are within this set of 16 (all but AI21 Labs).

**The gradient of release strategies.** The strategies for releasing foundation models differ widely (see Figure 6). Some developers release the weights of the model as well as the data used, which allows independent researchers and developers to use the models on their own and investigate the data. For example, EleutherAI released the weights of its Neo-X model (Black et al., 2022) along with The Pile, which Neo-X was trained on (Gao et al., 2021a). Meta released the weights to its OPT model (Zhang et al., 2022b), but did not release the associated training data. For our purposes, we will often refer to any release where model weights are made broadly available as "open," which includes the flagship models of Hugging Face, Meta, and Stability AI.

In contrast, other developers do not release the weights of their flagship model, retaining greater control over who has access to the model and the extent to which it may be used externally (if at all). The majority of the developers we assess provide a programmatic API to query their flagship model as a black box. Other developers in the ecosystem do not provide a programmatic API but do allow for some forms of black box

---

[44]See `https://www.blumenthal.senate.gov/imo/media/doc/09072023bipartisanaiframework.pdf`.

[45]`https://www.whitehouse.gov/briefing-room/statements-releases/2023/05/04/fact-sheet-biden-harris-administration-announces-new-actions-to-promote-responsible-ai-innovation-that-protects-americans-rights-and-safety/`

[46]`https://www.whitehouse.gov/briefing-room/statements-releases/2023/07/21/fact-sheet-biden-harris-administration-secures-voluntary-commitments-from-leading-artificial-intelligence-companies-to-manage-the-risks-posed-by-ai/`

[47]`https://www.whitehouse.gov/briefing-room/statements-releases/2023/09/12/fact-sheet-biden-harris-administration-secures-voluntary-commitments-from-eight-additional-artificial-intelligence-companies-to-manage-the-risks-posed-by-ai/`

[48]`https://blogs.microsoft.com/on-the-issues/2023/07/26/anthropic-google-microsoft-openai-launch-frontier-model-forum/`

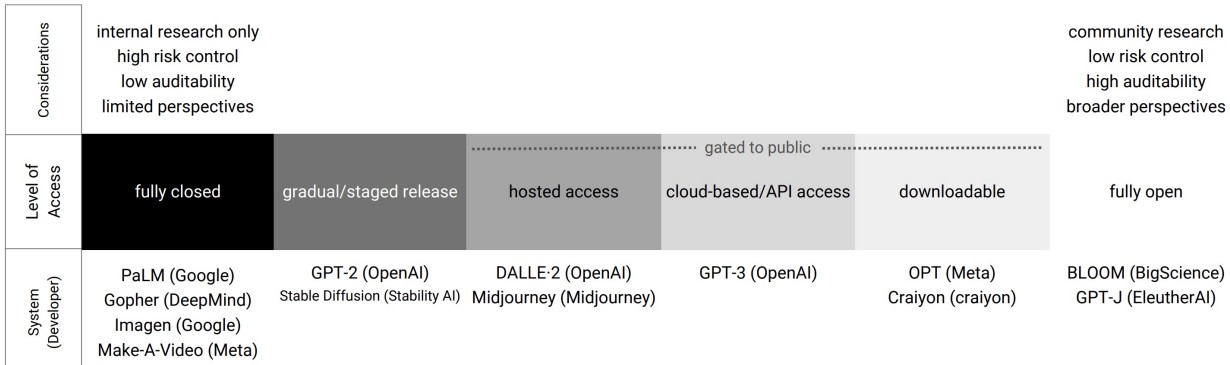

Figure 6: **The gradient of release of foundation models.** Foundation models can be fully closed (e.g. only used internally within the company, without public release), released gradually as their risks and benefits are better understood (e.g. via a staged rollout involving initial testers), released via a web or app interface (e.g. users need to visit a website or join a Discord server to access the model's outputs), released via a programmatic API (e.g. users can query the model and receive outputs programmatically), released via downloadable model weights (e.g. users can access and adapt the model), or released with the training data alongside downloadable model weights (i.e. ostensibly maximal openness). For the ten models we consider, one falls under the fully closed category at the time of writing (Inflection-1), though Inflection plans to make it available via an API; six are available via an API (GPT-4, Claude 2, PaLM 2, Jurassic-2, Command, Titan Text); one is downloadable (Llama 2), and two are released with their model weights as well as underlying training data downloadable (Stable Diffusion 2 and BLOOMZ). For simplicity, we at times binarize these distinctions into models with downloadable weights ("open") and models without downloadable weights ("closed"). Image taken with permission from Solaiman (2023).

access, as Midjourney does for its text-to-image models that it makes available via a Discord server.[49] Still other developers provide no external access to their models as is the case for Google's Chinchilla model (Hoffmann et al., 2022a) and Meta's Make-A-Video model (Singer et al., 2022). For our purposes, we will often refer to any release where model weights are not made externally available as "closed," which includes the flagship models of AI21 Labs, Amazon, Anthropic, Cohere, Google, Inflection, and OpenAI.

The overall approach to release is informed by a developer's business strategy and perspective on its model's utility and risks. In particular, many organizations may adopt different release approaches for different foundation models. For example, when releasing GPT-4, OpenAI did not disclose many details about the modeling architecture and training data, citing competition and safety as the two main reasons.[50] On the other hand, when releasing the text-to-speech Whisper model Radford et al. (2022), OpenAI disclosed many details and released the model weights openly. For other developers, the release decision may directly relate to their purpose for building a foundation model in the first place. For example, the BigScience collaboration led by Hugging Face that led to the BLOOM model (Le Scao et al., 2022) was explicitly designed to democratize access to multilingual large language models with capabilities in traditionally underrepresented languages. As a result, the initiative released model weights and data.

---

[49]See `https://docs.midjourney.com/docs/midjourney-discord`.

[50]Interview with OpenAI's chief scientist and co-founder: `https://www.theverge.com/2023/3/15/23640180/openai-gpt-4-launch-closed-research-ilya-sutskever-interview`

## 5.2 Selecting flagship models

Almost all major foundation model developers release multiple foundation models over time and, even at the time of writing, many have multiple salient foundation models (often across different modalities). For example, OpenAI has developed GPT, GPT-2, GPT-3, GPT-4, InstructGPT, WebGPT, Codex, CLIP, DALL-E, DALL-E 2, DALL-E 3, Jukebox, and Whisper among other models. Given that developers are not guaranteed to provide uniform transparency for each foundation model (e.g. OpenAI releases the weights openly for some of these models but not others), we decide to assess developers in relation to their *flagship* foundation model. By flagship foundation model, we mean the foundation model that is most salient and/or capable from the developer based on our judgment, which is directly informed by the company's public description of the model. We provide basic information about each of the developers and their flagship model in Table 1.[51]

**Note on Hugging Face.** In the case of Hugging Face, we are assessing the company in general as an enduring target over time. However, for this version of the index, we assess BLOOMZ (Muennighoff et al., 2022), which was collaboratively developed through the year-long BigScience initiative that was initiated and led by Hugging Face from May 2021 to May 2022. As a result, we refer to Hugging Face throughout the prose, but include the BigScience logo in visuals (which may also be distributed absent the context we provide in this paper) to highlight this nuance.

---

[51]For OpenAI, we evaluate GPT-4, which was released in March 2023, not GPT-4V, a model OpenAI released in September 2023 after we completed our analysis. With respect to input and output modality, OpenAI (2023) states that GPT-4 is "a large multimodal model capable of processing image and text inputs and producing text outputs."

# 6    Scoring

By selecting the indicators and companies, we abstractly specify the form of the index. By defining each indicator and designating the flagship foundation model to be assessed for each developer, we move to a more precise operationalization. To make the index fully precise, we describe how we sourced the information that was used to assess each developer on each indicator, resulting in the final scores.

**Search protocol.**    To source information that we use to score developers, we exclusively use publicly available information provided by developers themselves. We recognize that this information may be incomplete (e.g. clients or governments may have greater access to information from the developer), but given that our focus includes public accountability, and we are academic researchers, we choose to consider only publicly available information. Given that public information may change, we use information available as of September 15, 2023.

For each developer, we initially compile a basic set of resources disclosed by the developer about their model development practices and their flagship foundation model. To gather information for a specific indicator, we perform a structured search to identify all relevant information that is public. The exact details of how we execute this search are provided in Appendix C.

**Initial scoring.**    Having identified the information basis for scoring an indicator, 2 researchers on the team independently scored the developer on the indicator. This entails specifying a *score* (i.e. 0 or 1), *source* used in arriving at that score (e.g. one or more webpages), and a textual *justification* for how the evidence from sources is weighed against the criteria for the indicator in determining the score. Given these initial score assignments, the researchers reviewed their scores to identify any errors.

Binary scoring provided several advantages. First, it simplified the scoring process by allowing researchers to focus on the sharp distinction between 0 and 1 point for each indicator. Second, a narrow criterion for making a binary scoring decision for each indicator reduced subjectivity in the initial scoring. Third, by reducing the level of complexity of each indicator we were able to reduce overlap between indicators, ensuring that we assess distinct dimensions of transparency. At the same time, binary scoring limits the level of complexity of each indicator, potentially leaving out valuable information that can be captured by more complex scoring schemes (cf. Bommasani et al., 2023a).[52]

In some instances, the researchers responsible for the same (indicator, developer) pair arrived at different scores, indicating disagreement. Given the systematic information gathering process, the iterative refinement of indicator definitions, and the binary scoring scheme, we found that disagreements were fairly infrequent. Disagreements generally related to relevant information being erroneously neglected by one researcher or differences in the fine-grained interpretation of how to score an indicator. Overall, across all $100 \times 10$ (indicator, developer) pairs, the agreement rate was 85.2% (Cohen's $\kappa = 0.67$, indicating substantial agreement; Landis & Koch, 1977). To resolve disagreements, the researchers discussed and jointly came to a resolution. Following the disagreement resolution, the scores were finalized and sources and justifications were merged to yield an initial set of 1000 (score, source, justification) triples for all 1000 (indicator, developer) pairs.

---

[52]See §9.2: LIMITATIONS for further discussion.

**Company feedback.** Given that these scores constitute a direct assessment of specific companies, we engaged these companies to provide them with the opportunity to review, respond, and potentially rebut or contest the scores we assigned. Concretely, we contacted leaders at each of the companies with (i) a description of the Foundation Model Transparency Index, (ii) the 100 indicators and their definitions, and (iii) their 100 (score, source, justification) triples. We encouraged each company to review our scores, provide any general feedback and, especially, to directly contest any scores the company viewed as incorrect (by referencing public information available as of September 15, 2023). Companies were provided two business weeks to respond with clear assurance that all correspondence would be strictly private.

Of the 10 companies, all 10 responded. Of these, 8 companies (Amazon, Anthropic, Cohere, Hugging Face, Inflection, Meta, OpenAI, Stability AI) provided rebuttals for specific scores, which we extensively reviewed. In most cases, we did not change scores, though some rebuttals led to improvements in the scores (an average increase of 1.25 points across the 8 developers that contested on average 8.75 scores). Rather than improving developers' scores, these rebuttals often revealed misunderstandings regarding definitions of indicators or our justifications for scores, leading to more robust definitions and justifications. Beyond the scores, several companies scheduled calls with us or provided broader forms of feedback, which provided insight regarding how they conceptualize best practices for transparency and responsible AI. Following company feedback, we again verified all scores, sources, and justifications that constitute the finalized materials used throughout this paper and made publicly available.

We also notified the companies prior to the release of this paper, responding to their feedback. In addition, we encouraged companies to provide a public written response regarding their perspective on this initiative, their specific scores, and their broader approach as an organization to transparency and responsible AI as it relates to foundation models. Moving forward, we hope these organizations implement more transparent practices and we provide specific recommendations to that effect in §8.1: RECOMMENDATIONS-DEVELOPERS.

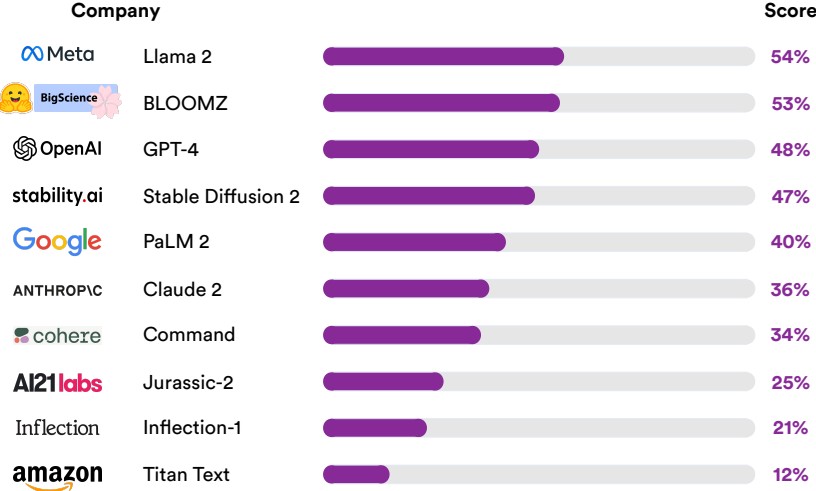

Figure 7: **Overall Scores.** The overall Foundation Model Transparency Index score and ranking across all 100 indicators.

## 7 Analysis

The finalized results of the Foundation Model Transparency Index are the scores for each of the 100 indicators across all 10 companies. These result are accessible at `https://www.github.com/stanford-crfm/fmti` to facilitate subsequent analyses. Here, we specifically consider overarching trends in the results, along with more specific trends based on the structure of the index. Namely, we analyze along the rows/indicators (e.g. domains), the columns/companies (e.g. release strategy), as well as data-driven trends (e.g. correlations).

### 7.1 Overarching results

We begin our analysis by first establishing the broad trends when viewing the index as a whole. We consider results aggregated at the level of a single overall score per company (Figure 7) as well as the scores broken down into the 3 domains (upstream, model, downstream; Figure 8). We supplement our findings on these overarching trends with a more granular consideration of the *major dimensions of transparency* in the index in Figure 9.[53]

**All developers have significant room for improvement. But most transparency indicators are very obtainable, having been implemented by at least one developer.** Based on Figure 7, the highest-scoring developer scores points for 54 of the 100 indicators, and the average score across all developers is 37. This establishes a pervasive lack of transparency across major foundation model developers. With that said, for 82 of the 100 indicators, there exists some developer that scores points, and of these there are 71 where multiple developers score points. Consequently, there is clear reason to believe that across all developers, the necessary change to become more transparent is feasible. That companies' competitors are more transparent in certain issue areas suggests that such transparency, even if not fully costless, is unlikely to cause serious damage to their business. Companies can emulate the higher level of transparency their competitors exhibit on certain indicators, providing a precedent and a starting point for improving transparency in the foundation model ecosystem.

---

[53]The major dimensions of transparency we highlight are 13 large subdomains among the 23 subdomains.

**Foundation Model Transparency Index Scores by Domain, 2023**

Source: 2023 Foundation Model Transparency Index

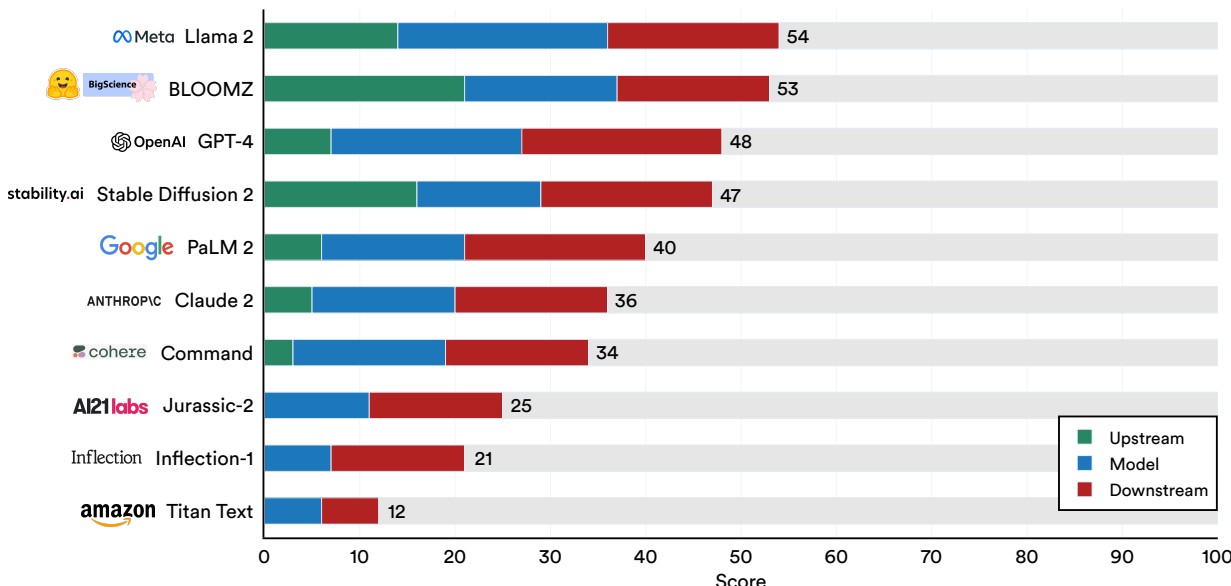

Figure 8: **Scores by Domain.** The aggregate score of each developer broken down by the three domains: upstream, model, and downstream.

**Foundation Model Transparency Index Scores by Major Dimensions of Transparency, 2023**

Source: 2023 Foundation Model Transparency Index

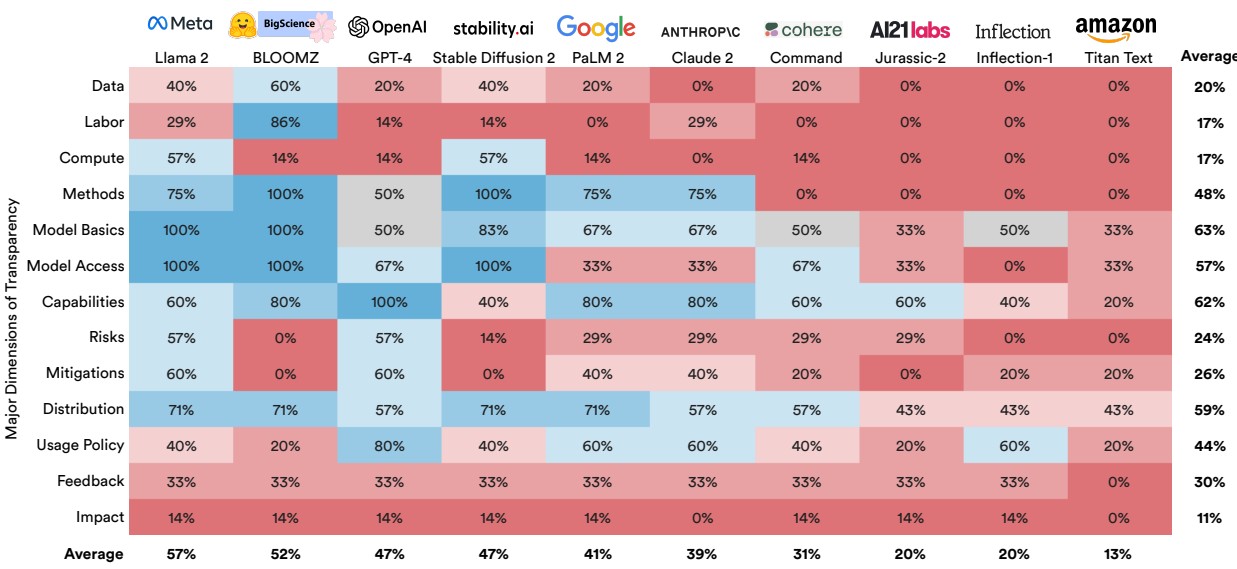

Figure 9: **Scores by Major Dimensions of Transparency.** The fraction of achieved indicators in each of the 13 major dimension of transparency. Major dimension of transparency are large subdomains within the 23 subdomains.

**Developers show significant variance in overall transparency scores.** While all developers have significant room for improvement, the current transparency of developers is strikingly uneven. Namely, the range in overall scores is 42 between the highest-scoring Meta at 54 and the lowest-scoring Amazon at 12. Even excluding Amazon's score as especially low, we still see an effective range of 30 points between Meta and the next lowest Inflection. Overall, with respect to the mean of 37, the standard deviation is 14.2, which is quite substantial. The four top-scoring developers (Meta, Hugging Face, OpenAI, Stability AI) all cluster well above the mean, the next three are very close to the mean (Google, Anthropic, Cohere), and the three lowest-scoring developers (AI21 Labs, Inflection, Amazon) are well below the mean. In many cases, the lowest-scoring developers have clear opportunities for improvement through straightforward changes related to some of the least challenging indicators. Examples include improved documentation (e.g. change logs, versioning protocols, model cards, centralized documentation for downstream use), clearer language in corporate policies (e.g. usage policies, model behavior policies, deprecation policies), and disclosing additional information that is unlikely to have implications for business competitiveness or safety (e.g. basic details on methods, dependencies, feedback).

**The Upstream domain sees the worst transparency scores.** To gain additional insight beyond developers' basic overall scores, we consider scores broken down by the 3 top-level domains in Figure 8. On this basis, we see clear evidence that developers are, on average, least transparent with respect to the upstream resources required to build their models, such as data, labor, and compute. Concretely, the mean score on upstream indicators is 7.2 out of 32 (22.5%), compared to 14.1 out of 33 (42.7%) for model indicators and 15.7 out of 35 (44.9%) for downstream indicators. To confirm this is not overly biased by outliers, we note that the medians show the same trend: the median score on upstream indicators is 3.5, compared to 12.5 for model indicators and 16 for downstream indicators. We specifically highlight that the four lowest-scoring developers overall (Figure 7) also fare the worst on the upstream domain (Figure 8), with Cohere receiving 3 points and all of AI21 Labs, Inflection, and Amazon receiving 0 points. In contrast, for both the model and downstream domains, all 10 companies receive at least 6 points.

**Domain-level discrepancies explain some of the differences between companies with similar overall scores.** We partition the 10 companies into three groups based on whether their overall score (Figure 7) is well-above (Meta, Hugging Face, OpenAI, Stability AI), around (Google, Anthropic, Cohere), or well-below (AI21 Labs, Inflection, Amazon) the mean. Within these groups, while companies receive somewhat similar scores, we find that their domain-level scores clarify discrepancies between them. Among the highest scorers, OpenAI is considerably less transparent on upstream matters (7) as compared to the other three high-scoring companies (Meta with 14, Hugging Face with 21, Stability AI with 16). In particular, OpenAI and Stability AI receive the nearly the same overall score, with OpenAI making up the deficit to Stability AI on upstream transparency mostly through better model-level transparency (and, specifically, many of the indicators on evaluations and risks). For the middle category of Google, Anthropic, and Cohere, the discrepancies are less stark, but we do see that Cohere is at 3 in the upstream category compared to Google with 6 and Anthropic with 5. Given the broadly similar scores for these three developers across all of the domains, we revisit the extent to which they are correlated at a finer-grained level in §7.6: CORRELATIONS. Among the three lowest-scoring developers, we see that AI21 Labs and Inflection are differentiated by the model domain, with both scoring a zero on the upstream domain and similarly on the downstream domain.

**Data, Data Labor, and Compute are pervasive blind spots across developers.** While the overall and domain-level results provide a basic lay of the land, we find that the major dimensions of transparency provide the Goldilocks region for clear and incisive analysis as shown in Figure 9. In particular, these dimensions of transparency are subdomains with several indicators (so the subdomain scores are more reliable) that are tied to broadly-understandable concepts like labor and capabilities. We hone in on the following major dimensions of transparency: Data, Data Labor, Compute, Methods, Model Basics, Model Access, Capabilities, Risks, Model Mitigations, Distribution, Usage Policy, Model Behavior Policy, Model Updates, User Data Protection, Feedback, and Impact. Analysis at this level reveals actionable insight into what types of transparency or opacity lead to many of our top findings. For example, we find that the poor upstream transparency stems from low performance on the Data, Data Labor, and Compute subdomains;

developers average just 20%, 17%, and 17% for Data, Data Labor, and Compute respectively. In terms of smaller subdomains, developers on average score 25% of the available points on Data Mitigations.

**Model Basics, Capabilities, Limitations, and User Data Protection are the most transparent subdomains at present, but still short of the ideal.** Developers score the highest proportion of points on indicators related to the following subdomains: User Interface (85%), Downstream Documentation (70%), User Data Protection (67%), Model Basics (63%), and Model Updates (63%). This reflects some baseline level of transparency across developers with respect to notifying users they are interacting with AI systems, providing centralized documentation for downstream use, publishing data protection policies, and disclosing the modalities associated with their model. Still, there are gaps in even for these subdomains. No developer provides a protocol for accessing usage data. Most developers (8 of 10) do not disclose the size of their model. And only half of the developers provide any form of deprecation policy.

## 7.2   Upstream results

Upstream indicators assess transparency regarding the ingredients that go into the foundation model including data, labor, compute, methods, and code. These ingredients are important predictors of the capabilities and risks of the foundation model they produce, as well as externalities of the model development process (e.g. impacts on human laborers and the environment). As we show in Figure 8, the upstream indicators are the most sparsely awarded (22.5% coverage on average). Here, we analyze at the level of subdomains and indicators based on Figure 10.

**The Upstream domain shows the greatest spread.** Building on the fact that developers score worst on the upstream domain–with several developers scoring exactly or nearly 0 points–we find the range in scores is the greatest for this domain. Namely, only one developer (Hugging Face) scores more than half of the indicators (21 of the available 32 indicators; 65.6%), yielding a range of 21 when compared to the lowest-scoring developers: AI21 Labs, Inflection, and Amazon (0 of the available 32 indicators; 0%). We emphasize this striking disparity given that many of the fundamental societal issues in connection with foundation models relate to upstream resources: bias, copyright, and privacy in relation to data, worker protections and fair compensation in relation to labor, environmental impact and energy expenditure in relation to compute, reproducibility in relation to methods, and cybersecurity in relation to code.

**The Methods subdomain is the most transparent in aggregate, while Data Labor is the least transparent.** Among the upstream subdomains, only Methods shows some degree of coverage, with six of the developers giving some description of training stages, training objectives, and dependencies. On the other end of the spectrum, Data Labor sees little to no coverage with the exception of BLOOMZ, which involved volunteers providing data. Developers generally share no information about the use of human labor in their data pipeline, the employer, wages, and geographical distribution of these workers, instructions they give to data annotators, or any labor protections they implement. This industry norm of being nontransparent with respect to data labor is in tension with the fact that such information is critical to reinforcement learning with human feedback (Ziegler et al., 2019; Ouyang et al., 2022; Casper et al., 2023). That data labor is one of the two least transparent subdomains is consistent with prior work documenting widespread ethical challenges with data labor (Gray & Suri, 2019a; Crawford, 2021; Hao & Seetharaman, 2023).

**The Compute subdomain shows major discrepancies among developers.** Meta and Stability AI document some aspects of compute, energy, and hardware usage, as well as the carbon footprint of model development, whereas many developers do not. Given the significant compute expenditure required to build many foundation models, the practice of documenting energy use and environmental impact is well-established along with associated tooling to measure these quantities (Lacoste et al., 2019; Strubell et al., 2019; Schwartz et al., 2020; Luccioni & Hernández-García, 2023). In spite of this, most developers do not disclose minimal, or sometimes any, details related to compute usage, particularly with respect to energy usage, carbon footprint, and environmental impact.

The broader environmental impact of building foundation models is also essential to consider; although there has been significant public attention concerning energy expenditure, other matters such as water

**Foundation Model Transparency Index Indicator-Level Scores for Upstream, 2023**
Source: 2023 Foundation Model Transparency Index

| Subdomain | Indicator | Llama 2 | BLOOMZ | GPT-4 | Stable Diffusion 2 | PaLM 2 | Claude 2 | Command | Jurassic-2 | Inflection-1 | Titan Text |
|---|---|---|---|---|---|---|---|---|---|---|---|
| Data | Data size | 1 | 1 | 0 | 1 | 0 | 0 | 1 | 0 | 0 | 0 |
| | Data sources | 0 | 1 | 0 | 0 | 0 | 0 | 0 | 0 | 0 | 0 |
| | Data creators | 0 | 0 | 0 | 0 | 0 | 0 | 0 | 0 | 0 | 0 |
| | Data source selection | 0 | 1 | 0 | 0 | 0 | 0 | 0 | 0 | 0 | 0 |
| | Data curation | 1 | 1 | 1 | 1 | 1 | 0 | 1 | 0 | 0 | 0 |
| | Data augmentation | 1 | 1 | 0 | 1 | 1 | 0 | 0 | 0 | 0 | 0 |
| | Harmful data filtration | 1 | 1 | 1 | 1 | 0 | 0 | 0 | 0 | 0 | 0 |
| | Copyrighted data | 0 | 0 | 0 | 0 | 0 | 0 | 0 | 0 | 0 | 0 |
| | Data license | 0 | 0 | 0 | 0 | 0 | 0 | 0 | 0 | 0 | 0 |
| | Personal information in data | 0 | 0 | 0 | 0 | 0 | 0 | 0 | 0 | 0 | 0 |
| Data Labor | Use of human labor | 1 | 1 | 0 | 0 | 0 | 0 | 0 | 0 | 0 | 0 |
| | Employment of data laborers | 0 | 1 | 0 | 0 | 0 | 0 | 0 | 0 | 0 | 0 |
| | Geographic distribution of data laborers | 0 | 1 | 0 | 0 | 0 | 0 | 0 | 0 | 0 | 0 |
| | Wages | 0 | 1 | 0 | 0 | 0 | 1 | 0 | 0 | 0 | 0 |
| | Instructions for creating data | 1 | 1 | 0 | 0 | 0 | 1 | 0 | 0 | 0 | 0 |
| | Labor protections | 0 | 0 | 1 | 0 | 0 | 0 | 0 | 0 | 0 | 0 |
| | Third party partners | 0 | 1 | 0 | 1 | 0 | 0 | 0 | 0 | 0 | 0 |
| Data Access | Queryable external data access | 0 | 1 | 0 | 1 | 0 | 0 | 0 | 0 | 0 | 0 |
| | Direct external data access | 0 | 1 | 0 | 1 | 0 | 0 | 0 | 0 | 0 | 0 |
| Compute | Compute usage | 0 | 0 | 0 | 0 | 0 | 0 | 0 | 0 | 0 | 0 |
| | Development duration | 1 | 0 | 0 | 1 | 0 | 0 | 0 | 0 | 0 | 0 |
| | Compute hardware | 0 | 1 | 0 | 1 | 0 | 0 | 0 | 0 | 0 | 0 |
| | Hardware owner | 1 | 0 | 1 | 1 | 1 | 0 | 1 | 0 | 0 | 0 |
| | Energy usage | 1 | 0 | 0 | 0 | 0 | 0 | 0 | 0 | 0 | 0 |
| | Carbon emissions | 1 | 0 | 0 | 1 | 0 | 0 | 0 | 0 | 0 | 0 |
| | Broader environmental impact | 0 | 0 | 0 | 0 | 0 | 0 | 0 | 0 | 0 | 0 |
| Methods | Model stages | 1 | 1 | 1 | 1 | 1 | 1 | 0 | 0 | 0 | 0 |
| | Model objectives | 1 | 1 | 1 | 1 | 1 | 1 | 0 | 0 | 0 | 0 |
| | Core frameworks | 0 | 1 | 0 | 1 | 1 | 0 | 0 | 0 | 0 | 0 |
| | Additional dependencies | 1 | 1 | 0 | 1 | 1 | 1 | 0 | 0 | 0 | 0 |
| Data Mitigations | Mitigations for privacy | 1 | 1 | 1 | 1 | 0 | 0 | 0 | 0 | 0 | 0 |
| | Mitigations for copyright | 0 | 1 | 0 | 0 | 0 | 0 | 0 | 0 | 0 | 0 |
| | **Upstream Subtotal** | **44%** | **66%** | **22%** | **50%** | **19%** | **16%** | **9%** | **0%** | **0%** | **0%** |

Figure 10: **Upstream Scores by Indicator.** The scores for each of the 32 upstream indicators.

usage may be of similar consequence environmentally (Luccioni & Hernández-García, 2023). Luccioni et al. (2022) provides an excellent example, documenting the embodied emissions, dynamic consumption, and idle consumption associated with BLOOM (Le Scao et al., 2022). Given that BLOOMZ is derived from BLOOM, we note the potential for *documentation transience*, where prior documentation is not updated to reflect substantial changes and, therefore, does not correctly persist to the new asset. In particular, the additional broader environmental impact of deriving BLOOMZ from BLOOM is not disclosed.

**Widespread lack of upstream transparency on data creators, data license, copyrighted data and associated mitigations, and broader environmental impact.** Of the 32 indicators, no company scores points on six of them. These are the indicators for data creators, data license status, copyrighted data, copyright mitigations, compute usage and broader environmental impact. For data creators, in part we believe this reflects the nascent status of methods for providing web-scale understanding of who created the data (e.g. text, images) scraped from the Internet. However, we recognize that Hugging Face in particular has taken important steps to characterize aspects of who created the data, along with associated metadata for copyright, license, and personal information, for the ROOTS corpus used to build BLOOM (though not the additional data involved in building BLOOMZ). With respect to the copyrighted data and data license status indicators, we emphasize that information related to these indicators is at issue in ongoing litigation. In particular, Stability AI has explicitly argued that training foundation models on copyrighted data is protected by fair use doctrine in the U.S.[54] Closed developers may also view information related to their data as a key competitive advantage, or be disincentivized to share this information due to a perception of legal risk. Additionally, we note that we are surprised no developer directly discloses the compute usage in FLOPs to sufficient precision, though several disclose information that could be used to compute an estimate or upper bound.

---

[54]See https://www.judiciary.senate.gov/imo/media/doc/2023-07-12_pm_-_testimony_-_brooks.pdf and https://www.documentcloud.org/documents/23589439-openai-motion-to-dismiss as well as Lemley & Casey (2020).

**Foundation Model Transparency Index Indicator-Level Scores for Model, 2023**
Source: 2023 Foundation Model Transparency Index

| Subdomain | Indicator | Meta Llama 2 | BigScience BLOOMZ | OpenAI GPT-4 | stability.ai Stable Diffusion 2 | Google PaLM 2 | ANTHROPIC Claude 2 | cohere Command | AI21labs Jurassic-2 | Inflection Inflection-1 | amazon Titan Text |
|---|---|---|---|---|---|---|---|---|---|---|---|
| Model Basics | Input modality | 1 | 1 | 1 | 1 | 1 | 1 | 1 | 1 | 1 | 1 |
| | Output modality | 1 | 1 | 1 | 1 | 1 | 1 | 1 | 1 | 1 | 1 |
| | Model components | 1 | 1 | 0 | 1 | 0 | 0 | 0 | 0 | 0 | 0 |
| | Model size | 1 | 1 | 0 | 0 | 0 | 0 | 0 | 0 | 0 | 0 |
| | Model architecture | 1 | 1 | 0 | 1 | 1 | 1 | 0 | 0 | 0 | 0 |
| | Centralized model documentation | 1 | 1 | 1 | 1 | 1 | 1 | 1 | 0 | 1 | 0 |
| Model Access | External model access protocol | 1 | 1 | 1 | 1 | 0 | 0 | 1 | 0 | 0 | 0 |
| | Blackbox external model access | 1 | 1 | 1 | 1 | 1 | 1 | 1 | 1 | 0 | 1 |
| | Full external model access | 1 | 1 | 0 | 1 | 0 | 0 | 0 | 0 | 0 | 0 |
| Capabilities | Capabilities description | 0 | 1 | 1 | 1 | 1 | 1 | 1 | 1 | 1 | 1 |
| | Capabilities demonstration | 1 | 1 | 1 | 1 | 1 | 1 | 1 | 1 | 0 | 0 |
| | Evaluation of capabilities | 1 | 1 | 1 | 0 | 1 | 1 | 0 | 0 | 1 | 0 |
| | External reproducibility of capabilities evaluation | 1 | 1 | 1 | 0 | 1 | 1 | 0 | 0 | 0 | 0 |
| | Third party capabilities evaluation | 0 | 0 | 1 | 0 | 0 | 0 | 1 | 1 | 0 | 0 |
| Limitations | Limitations description | 1 | 1 | 1 | 1 | 1 | 1 | 1 | 0 | 1 | 0 |
| | Limitations demonstration | 0 | 0 | 0 | 0 | 0 | 0 | 1 | 0 | 0 | 0 |
| | Third party evaluation of limitations | 1 | 1 | 1 | 1 | 1 | 1 | 1 | 1 | 0 | 1 |
| Risks | Risks description | 1 | 0 | 1 | 1 | 0 | 1 | 1 | 1 | 0 | 0 |
| | Risks demonstration | 1 | 0 | 1 | 0 | 0 | 0 | 0 | 0 | 0 | 0 |
| | Unintentional harm evaluation | 1 | 0 | 0 | 0 | 1 | 0 | 0 | 0 | 0 | 0 |
| | External reproducibility of unintentional harm evaluation | 1 | 0 | 0 | 0 | 1 | 0 | 0 | 0 | 0 | 0 |
| | Intentional harm evaluation | 0 | 0 | 0 | 0 | 0 | 0 | 0 | 0 | 0 | 0 |
| | External reproducibility of intentional harm evaluation | 0 | 0 | 1 | 0 | 0 | 0 | 0 | 0 | 0 | 0 |
| | Third party risks evaluation | 0 | 0 | 1 | 0 | 0 | 1 | 1 | 1 | 0 | 0 |
| Model Mitigations | Mitigations description | 1 | 0 | 1 | 0 | 1 | 1 | 1 | 0 | 1 | 1 |
| | Mitigations demonstration | 1 | 0 | 1 | 0 | 0 | 0 | 0 | 0 | 0 | 0 |
| Mitigations | Mitigations evaluation | 1 | 0 | 1 | 0 | 1 | 1 | 0 | 0 | 0 | 0 |
| | External reproducibility of mitigations evaluation | 0 | 0 | 0 | 0 | 0 | 0 | 0 | 0 | 0 | 0 |
| | Third party mitigations evaluation | 0 | 0 | 0 | 0 | 0 | 0 | 0 | 0 | 0 | 0 |
| Trustworthiness | Trustworthiness evaluation | 0 | 0 | 1 | 0 | 0 | 0 | 1 | 1 | 0 | 0 |
| | External reproducibility of trustworthiness evaluation | 0 | 0 | 0 | 0 | 0 | 0 | 1 | 1 | 0 | 0 |
| Inference | Inference duration evaluation | 1 | 1 | 0 | 0 | 0 | 0 | 0 | 0 | 0 | 0 |
| | Inference compute evaluation | 0 | 0 | 0 | 0 | 0 | 0 | 0 | 0 | 0 | 0 |
| | **Model Subtotal** | **67%** | **48%** | **61%** | **39%** | **45%** | **45%** | **48%** | **33%** | **21%** | **18%** |

Figure 11: **Model Scores by Indicator.** The scores for each of the 33 model indicators.

**No upstream indicators are satisfied by all developers.** At the indicator level, there is no upstream indicator for which every developer receives points. Of course, this is guaranteed by the presence of (multiple) developers that score 0 points on the entire upstream domain. Even putting these 3 developers aside, there is no indicator that is satisfied by all of the remaining 7. The indicators where the greatest number of developers score points are data curation (all but Anthropic) and model stages (all but Cohere), which both suggest that developers are generally willing to describe the basics of the overall pipeline of model development. With that said, we take the absence of any upstream indicator where all companies score points, and the fact that 5 or more developers score no points on 30 of 32 upstream indicators, as strong evidence that upstream transparency is the domain with the broadest room for improvement.

## 7.3 Model results

Model indicators assess transparency regarding the function of foundation models, spanning model access, capabilities, risks, limitations, mitigations, trustworthiness and inference efficiency, as well as basic information about the model. The indicators in this domain comprehensively characterize the foundation model as a standalone artifact: what tasks the model can and cannot perform, what is the model's basic structure, who has access to the model, and more. Here, we analyze developers at the level of subdomains and indicators based on Figure 11.

**Model subdomains are some of the highest-scoring across the index.** Overall, the mean score on model indicators is 14.1 out of 33 (42.7%) and the median developer receives 12.5 points (37.9%). With this in mind, several of the highest-scoring subdomains belong to the model domain. Developers score best on Model Basics (63%), Capabilities (62%), Limitations (60%), and Model Access (57%) within the domain. These scores arise partially because of very generous indicators within these subdomains (e.g. input modality, output modality, description of capabilities, description of limitations).

**Transparency on capabilities does not translate to transparency on limitations, risks, or mitigations.** Of the 33 model indicators, 20 are in the Capabilities, Limitations, Risks, and Model Mitigations subdomains. Within these subdomains, Capabilities is clearly the most transparent subdomain: nearly all developers provide descriptions (9 of 10) and demonstrations (8 of 10) of multiple model capabilities, with the majority reporting evaluations (6 of 10), half reporting reproducible evaluations (5 of 10), and few providing third party evaluations (3 of 10). In general, we see a decline in the number of developers who score the point from the most rudimentary (i.e. description) to the most substantive (i.e. third party evaluations) across these four subdomains. With respect to Capabilities, while we assume most or all developers conduct internal evaluations, they may not score points on evaluations indicators because (i) they do not disclose sufficient details about internal evaluations for these evaluations to be externally reproducible, (ii) they do not assess multiple capabilities, or (iii) they do not report the results of the evaluations, perhaps due to a concern that a model may underperform competitors' models.

With this in mind, developers consistently score worse on Limitations, Risks, and Model Mitigations indicators than on Capabilities. For example, only Cohere receive points for demonstrating limitations, while 8 developers score points for demonstrating capabilities. These asymmetries where companies are more willing to share information about capabilities than limitations, risks, and mitigations are concerning, as they may lead to an inflated sense of trust in companies' foundation models. In fact, these asymmetries are especially pronounced for Risks (average score of 24%) and Model Mitigations (average score of 26%), given that these scores are considerably worse than the average scores for Capabilities (62%) and Limitations (60%).

**Developers score poorly on Trustworthiness, largely in line with Risks and Model Mitigations.** With respect to the Trustworthiness subdomain, only OpenAI, Cohere, and AI21 Labs provide information about rigorous evaluations of their flagship model related to robustness, reliability, hallucinations, calibration, or explainability. Of those developers, only Cohere and AI21 Labs provide sufficient detail for their evaluations to be deemed externally reproducible due to their use of the HELM benchmark (Liang et al., 2023), compared to OpenAI's unclear description of their evaluations of model calibration. Given the previous asymmetry we establish around greater disclosure of capabilities as compared to limitations, risks, and mitigations, the absence of trustworthiness evaluations exacerbates these concerns. Put together, the lack of sufficient public information on limitations, risks, mitigations, and trustworthiness makes it more likely that consumers will not have well-calibrated expectations. In turn, this could lead to undesirable overreliance on foundation models because not enough is done to calibrate consumers on the appropriate levels of trust (Parasuraman & Manzey, 2010).[55] With this said, we do acknowledge that developers may take other routes towards improving trustworthiness including methods like reinforcement learning from human feedback (Ziegler et al., 2019; Ouyang et al., 2022) and constitutional AI (Bai et al., 2022), though transparency is lacking on these approaches (Casper et al., 2023).

**Model Access reveals slight differences beyond just release strategy.** In aggregate, companies score 17 of the 30 points (57%) in the Model Access subdomain across the 3 indicators and 10 companies. On the external model access protocol indicator, Meta, Hugging Face, OpenAI, and Stability AI are the only developers to score points. We find this particularly interesting given Meta, Hugging Face and Stability AI release their models openly in terms of both model weights and data, whereas OpenAI is considerably more closed, providing only API access. However, in particular, OpenAI has a clear researcher access program with a form to request access, criteria it discloses for granting access, and a period of 4–6 weeks disclosed as the expected turnaround for a decision. This demonstrates that developers across the release spectrum (Solaiman, 2023) may achieve transparency on some indicators while taking substantively different approaches. In practice, we find that several closed developers have access forms that allow external entities greater access to the model, but these forms often lack key components of transparency that clarify the specific steps the developer will take to assess and grant applications (e.g. in comparison to OpenAI's process). With that said, the indicator for full external model access is exclusively achieved by the three open developers, though every developer other than Inflection provides black box access access to its model.

---

[55]See https://www.theverge.com/2023/5/30/23741996/openai-chatgpt-false-information-misinformation-responsibility as an example.

**Foundation Model Transparency Index Indicator-Level Scores for Downstream, 2023**

Source: 2023 Foundation Model Transparency Index

| Subdomain | Indicator | Llama 2 | BLOOMZ | GPT-4 | Stable Diffusion 2 | PaLM 2 | Claude 2 | Command | Jurassic-2 | Inflection-1 | Titan Text |
|---|---|---|---|---|---|---|---|---|---|---|---|
| Distribution | Release decision-making protocol | 0 | 1 | 0 | 0 | 0 | 0 | 0 | 0 | 0 | 0 |
| | Release process | 1 | 1 | 1 | 1 | 1 | 1 | 0 | 1 | 0 | 1 |
| | Distribution channels | 1 | 1 | 1 | 1 | 1 | 1 | 1 | 1 | 1 | 1 |
| | Products and services | 1 | 0 | 1 | 0 | 1 | 1 | 1 | 0 | 1 | 0 |
| | Machine-generated content | 0 | 0 | 0 | 1 | 0 | 0 | 0 | 0 | 0 | 0 |
| | Model License | 1 | 1 | 0 | 1 | 1 | 0 | 1 | 0 | 0 | 0 |
| | Terms of service | 1 | 1 | 1 | 1 | 1 | 1 | 1 | 1 | 1 | 1 |
| Usage Policy | Permitted and prohibited users | 1 | 0 | 1 | 0 | 1 | 1 | 1 | 0 | 0 | 0 |
| | Permitted, restricted, and prohibited uses | 1 | 1 | 1 | 1 | 1 | 1 | 1 | 1 | 1 | 1 |
| | Usage policy enforcement | 0 | 0 | 1 | 1 | 0 | 1 | 0 | 0 | 0 | 0 |
| | Justification for enforcement action | 0 | 0 | 0 | 0 | 1 | 0 | 0 | 0 | 1 | 0 |
| | Usage policy violation appeals mechanism | 0 | 0 | 1 | 0 | 0 | 0 | 0 | 0 | 1 | 0 |
| Model Behavior Policy | Permitted, restricted, and prohibited model behaviors | 0 | 0 | 1 | 0 | 0 | 1 | 0 | 0 | 1 | 0 |
| | Model behavior policy enforcement | 0 | 0 | 0 | 1 | 0 | 0 | 0 | 0 | 1 | 0 |
| | Interoperability of usage and model behavior policies | 0 | 0 | 1 | 0 | 0 | 1 | 0 | 0 | 0 | 0 |
| User Interface | User interaction with AI system | 1 | 1 | 1 | 1 | 1 | 0 | 1 | 1 | 1 | 0 |
| | Usage disclaimers | 1 | 1 | 1 | 1 | 1 | 1 | 1 | 1 | 1 | 0 |
| User Data Protection | User data protection policy | 1 | 1 | 1 | 1 | 1 | 1 | 1 | 1 | 1 | 1 |
| | Permitted and prohibited use of user data | 1 | 1 | 1 | 1 | 1 | 1 | 1 | 1 | 1 | 1 |
| | Usage data access protocol | 0 | 0 | 0 | 0 | 0 | 0 | 0 | 0 | 0 | 0 |
| Model Updates | Versioning protocol | 1 | 1 | 1 | 1 | 1 | 1 | 0 | 1 | 0 | 0 |
| | Change log | 1 | 1 | 1 | 1 | 1 | 0 | 1 | 1 | 0 | 0 |
| | Deprecation policy | 1 | 1 | 1 | 1 | 1 | 0 | 0 | 0 | 0 | 0 |
| Feedback | Feedback mechanism | 1 | 1 | 1 | 1 | 1 | 1 | 1 | 1 | 1 | 0 |
| | Feedback summary | 0 | 0 | 0 | 0 | 0 | 0 | 0 | 0 | 0 | 0 |
| | Government inquiries | 0 | 0 | 0 | 0 | 0 | 0 | 0 | 0 | 0 | 0 |
| Impact | Monitoring mechanism | 0 | 0 | 1 | 0 | 1 | 0 | 1 | 1 | 1 | 0 |
| | Downstream applications | 1 | 1 | 0 | 1 | 0 | 0 | 0 | 0 | 0 | 0 |
| | Affected market sectors | 0 | 0 | 0 | 0 | 0 | 0 | 0 | 0 | 0 | 0 |
| | Affected individuals | 0 | 0 | 0 | 0 | 0 | 0 | 0 | 0 | 0 | 0 |
| | Usage reports | 0 | 0 | 0 | 0 | 0 | 0 | 0 | 0 | 0 | 0 |
| | Geographic statistics | 0 | 0 | 0 | 0 | 0 | 0 | 0 | 0 | 0 | 0 |
| | Redress mechanism | 0 | 0 | 0 | 0 | 0 | 0 | 0 | 0 | 0 | 0 |
| Downstream Documentation | Centralized documentation for downstream use | 1 | 1 | 1 | 1 | 1 | 1 | 1 | 1 | 0 | 0 |
| | Documentation for responsible downstream use | 1 | 0 | 1 | 0 | 1 | 1 | 1 | 1 | 0 | 0 |
| | **Downstream Subtotal** | **51%** | **46%** | **60%** | **51%** | **54%** | **46%** | **43%** | **40%** | **40%** | **17%** |

Figure 12: **Downstream Scores by Indicator.** The scores for each of the 35 downstream indicators.

**Model Mitigations are a weak point for most developers.** Developers on average scored just 26% of the total available points on the five Model Mitigations indicators. Hugging Face, Stability AI, and AI21 Labs score 0 points, while Cohere, Inflection, and Amazon score only the point on mitigations description, which is the most lightweight of these indicators. In general, we highlight an important mismatch between the many risks that are enumerated and the relatively few mitigations that are described, implemented, and/or evaluated. Even when mitigations are described, in scoring we find the mapping between stated risks and stated mitigations is often vague or nonexistent. Moving forward, we hope developers will directly aim mitigations at addressing specific risks, with appropriate evaluations to confirm the efficacy of mitigations in achieving the stated goals.

**Most model indicators are scored by some developer, though most developers score poorly on indicators related to evaluating intentional harms, mitigations, and inference efficiency.** Of the 33 indicators in the model domain, at least one developer scores a point on 29 of them. Further, multiple developers score points on 27 model indicators. The 4 indicators for which no developer scores points are (i) intentional harm evaluation, (ii) external reproducibility of mitigations evaluations, (iii) third party mitigations evaluations, and (iv) inference compute evaluation. The 2 additional indicators for which only one developer scores points are limitations demonstration (Cohere) and external reproducibility of internal harm evaluation (OpenAI). While many companies describe risks (including the risk of intentional harms), they do not share sufficient information related to evaluations of intentional harm or the reproducibility of evaluations of mitigations. In the case of inference, we believe standards are needed akin to MLPerf (Reddi et al., 2020) to rigorously benchmark the inference of foundation models (Narayanan et al., 2023) given the key role of efficient inference and low latency in the usability of models (Lee et al., 2023b). We see that BLOOMZ in particular provides a potential benchmark for language models by tracking the time spent for a fixed task (generating 100 tokens given a 7 token prefix) on fixed hardware (a NVIDIA A100-80GB GPU), though compute is not measured.[56]

---

[56]See https://huggingface.co/blog/habana-gaudi-2-bloom.

### 7.4 Downstream results

Downstream indicators assess transparency regarding the use of foundation models, spanning subdomains related to distribution, policies constraining the use and behavior of the model, user interfaces, user data protection, model updates, feedback, impact, and documentation. Indicators in these subdomains characterize transparency related to how the foundation model is deployed and its downstream effects on the ecosystem and society. Our analysis is based on publicly available information about how the foundation model is distributed, how it can and cannot be used, how users can give feedback and seek redress, broader societal impacts, and the how the model affects actors downstream of the developer in the supply chain. Here, we conduct a fine-grained analysis at the level of subdomains and indicators based on Figure 12.

**Downstream scores show less spread across developers.** Total scores on downstream indicators are tightly clustered around the mean of 15.7 out of 35, which corresponds to 44.9% of the 35 downstream indicators. With the exception of Amazon (6 out of 35; 17.1%), the other nine developers all score between 14 and 21 points. The highest-scoring on the downstream domain is OpenAI at 21 points and the lowest-scoring (barring Amazon) are AI21 Labs and Inflection at 14 points. In §7.6: CORRELATIONS, we clarify the extent to which these smaller margins in scoring discrepancies in the downstream domain are due to high agreement in indicator-level scores across companies.

**Impact is the least transparent subdomain in the entire index.** To clarify the downstream impact of a given foundation model, the Impact subdomain includes indicators on monitoring mechanisms, affected market sectors, affected individuals, usage reports, geographic statistics, and redress mechanisms. Strikingly, the mean score across all developers on this subdomain is just 11%, with 8 developers scoring points on just 1 of the possible 7 indicators and the remaining 2 scoring none of the indicators. No developer scores points on affected market sectors, affected individuals, usage reports, geographic statistics, or redress mechanism. This means that there is essentially no information about how many people, sectors, and regions foundation models are impacting. OpenAI, Google, Cohere, AI21 Labs, and Inflection are the only developers to disclose a potential monitoring mechanism for tracking model use. And only open foundation model developers share limited information about downstream applications, whereas the rest provide no information.[57]

**Developers are significantly more transparent about Distribution than other major dimensions of (downstream) transparency.** Across the four major dimensions of transparency in the downstream domain (Distribution, Usage Policy, Feedback, Impact), mean scores are on the higher end only for Distribution at 59%, with the other three all below 50%. Every developer shares information about distribution channels, or the pathways by which the model is made available to entities beyond the model developer organization. Every developer provides terms of service that cover the distribution of its foundation model.[58] Most developers share information about their process for releasing their flagship model (8 of 10) as well as the developer's products and services that use the foundation model (6 of 10). Half of developers share information about the license under which the model is distributed.

**In spite of broad transparency on the Distribution subdomain, developers are highly opaque around release decisions.** Within the Distribution subdomain, developers score poorly on the release decision-making protocol indicator; Hugging Face is the only developer that shares information about its decision-making protocol for release. Although there has been an extensive focus on release strategies in the literature on foundation models (Solaiman et al., 2019; Sastry, 2021; Shevlane, 2022; Liang et al., 2022a; Liang & Reich, 2022; Solaiman, 2023; Widder et al., 2023; Seger et al., 2023), developers across the release spectrum share very little information about how and why they release their flagship models. In particular,

---

[57]We score the downstream applications indicator quite generously: all of the open developers score points because they discloses which Hugging Face "Spaces" are also using the model via Hugging Face's platform. However, we emphasize that this is still a poor proxy for the number of applications dependent on the foundation model.

[58]As with several downstream indicators, we assessed the terms of service of the primary distribution channel. For example, this meant that we assessed Microsoft Azure's terms of service for Meta.

we highlight that many of companies we assess have written about the broader topic of release, but not in a way that is precise to their specific decisions for their flagship models.[59]

**Usage Policy and Model Behavior Policy subdomain scores are uneven across developers.** Scores on the Usage Policy subdomain are uneven, with all developers scoring points on the indicator for permitted, restricted, and prohibited uses, but only two (OpenAI and Inflection) scoring points on the usage policy violation appeals indicator. This reflects the lack of industry standards regarding precisely how foundation model developers should restrict the use of their models. We found that different developers provide this information in different types of documentation, ranging from standalone Acceptable Use Policies to Content Policies to terms in the model license, and that many developers share some of this information in several different documents.

While developers did provide some transparency on usage policies related to a user's obligations, they did not provide a similar level of transparency on the restrictions they place on their model's behavior. Scores on indicators in the Model Behavior Policy subdomain were relatively weaker, with a mean across the 3 indicators of 23% compared to 44% for the 5 usage policy indicators. OpenAI, Anthropic, and Inflection are the only developers who provide information about permitted, restricted, and prohibited model behaviors, while only Inflection and Stability AI provide information about how they might enforce such restrictions. OpenAI and Anthropic are the only developers who make clear how their models are expected to behave in the event that a user violates the usage policy. In part, we believe the norms and standards around model behavior are rather immature, meaning that developers do not provide a clear conceptualization of if/how they impose a model behavior policy. For example, the role of modeling decisions (e.g. the use of reinforcement learning from human feedback or constitutional AI) on behaviors (e.g. model refusals to specific requests) are not made clear.

**Identical scores on the User Data Protection subdomain across all developers.** For the User Data Protection subdomain, scores are uniform across developers, with every developer scoring points on user data protection policy, as well as permitted and prohibited uses of user data. However, no developer scores points on usage data access protocol. This may reflect that few, if any, companies actually share usage data externally, meaning companies may perceive that the need to develop protocols for sharing such data is limited. However, developers' data protection policies include many provisions that would allow them to share such usage data, and specific protocols for how and when they do so are not transparent.

**Developers lack transparency on the Feedback subdomain.** Developers score relatively poorly on Feedback indicators, scoring only 30% of the available points. While every developer but Amazon has a public mechanism for collecting feedback on its model, none provide information such as a feedback summary or details on government inquiries, such as requests for user data (which social media companies disclose). This is likely a function of how nascent the foundation model ecosystem is: companies have only been collecting feedback for a few years, and it took social media companies several years to respond to public calls for transparency around the feedback they receive from users and governments. Moving forward, more robust transparency reporting practices that provide the public with more information regarding these forms of feedback will likely be necessary.[60]

**Developers are fairly transparent on the Model Updates subdomain.** 5 of 10 developers provide clear information about their versioning protocol, change log, and deprecation policy. Inflection and Amazon, however, score zero points on these indicators, which may be due in part due to the face that Inflection-1 and Titan Text are at an earlier stage of release than some other flagship models. While there is a wide variation in the type, specificity, and quality of documentation provided related to Model Updates, as with other indicators, we assess these metrics generously and allocate points on the basis of transparency alone.

---

[59]We note that following September 15, 2023, Anthropic released information about its approach to responsible scaling: `https://www.anthropic.com/index/anthropics-responsible-scaling-policy`.

[60]For example, consider the EU's DSA Transparency Database, implemented on the basis of the Digital Services Act to provide transparency on content moderation decisions: `https://transparency.dsa.ec.europa.eu/`.

**Developers score well on the User Interface subdomain, though this may change due to deployments on mobile phones.** Developers scored highly on User Interface indicators (average score of 85%), with more than half of developers scoring points on both indicators, which assess if users are told they are interacting with an AI system and if users are provided appropriate disclaimers. Developers frequently disclose to users that they are interacting with a specific foundation model by including the name of the foundation model somewhere in the user interface, while they give usage disclaimers upon sign-up for the user interface via a link to the terms of service or usage policy. Unlike all other indicators, we generally had to make use of step 7 in the and directly interact with developers' models via a user interface to assess these indicators. However, Amazon did not have a publicly available user interface in advance of September 15, 2023, meaning that it could not receive these points. We initially assessed transparency of deployments on mobile devices in some cases, though we ultimately did not consider these deployments for scoring. With that said, we highlight that the same standard for transparency of user interfaces does not currently appear to be met by mobile deployments from OpenAI and Inflection. Overall, we believe in the importance of providing transparency through user interfaces as it can help foundation models avoid the formation of the "dark patterns" we have seen develop with other digital technologies (Mathur et al., 2019). For example, we highlight that Anthropic does not make clear that a user is interacting with an AI system, except for the textual description "Message Claude."

## 7.5   Results for open and closed developers

Foundation models are released by different developers using a variety of release strategies (Liang et al., 2022a; Solaiman, 2023). In particular, we deliberately chose several developers that are more *open* (e.g. release the weights of their model, perhaps along with the data used to build the model) and others that are more *closed* (e.g. only provide access via an API). The topic of release and the (reductive) dichotomy of open vs. closed has emerged as a primary topic of technical and policy research on foundation models (Solaiman et al., 2019; Sastry, 2021; Shevlane, 2022; Liang et al., 2022a; Liang & Reich, 2022; Solaiman, 2023; Widder et al., 2023; Seger et al., 2023). To clarify how transparency differs between the open developers we assess (i.e. Meta, Hugging Face, Stability AI) and the closed developers (i.e. OpenAI, Google, Anthropic, Cohere, AI21 Labs, Inflection, Amazon), we emphasize the distinction in Figure 13.

**Open developers score higher in aggregate and on every domain.** We establish a clear trend that the open developers score higher overall, with all three being among the four highest-scoring developers (see Figure 7). In particular, every open developer is nearly at least as transparent in terms of aggregate score as the highest-scoring closed developer (OpenAI): Meta and Hugging Face are at least 5 points higher, and Stability AI is within a point of OpenAI. Further, this trend is established more strongly through domain-level analysis, where open developers score higher on average than closed developers across all domains (i.e. upstream, model, downstream). The mean score of open developers on upstream indicators is 53% compared to 9% for closed developers, 51% for open developers on model indicators compared to 39% for closed developers, and 49% on downstream indicators compared to 43% for closed developers. To ensure these trends are robust to outliers, we highlight that the trends hold even when considering medians instead of means (upstream: 50% to 9%, model: 48% to 45%, downstream: 51% to 43%).

We emphasize that our findings confirm common hypotheses that open developers will in general be more transparent with respect to the upstream resources required to build their models (which also aligns with some making the data they use publicly available), but our findings dispute hypotheses that open developers will be less transparent on downstream matters due to their weaker control over downstream use. While we believe that closed developers providing APIs are better positioned to collect information on the downstream use of their models, in practice these developers do not disclose this information to provide greater public transparency.

**Open developers score higher on most subdomains.** Open developers score higher than closed developers on 15 of the 23 subdomains, which account for 68 of the 100 indicators. The mean score of closed developers is higher than that of open developers on indicators in the subdomains of Capabilities, Risks, Model Mitigations, Trustworthiness, Usage Policy, Model Behavior Policy, and Downstream Documentation. We highlight that these seven subdomains point to two broader themes: closed developers in some cases

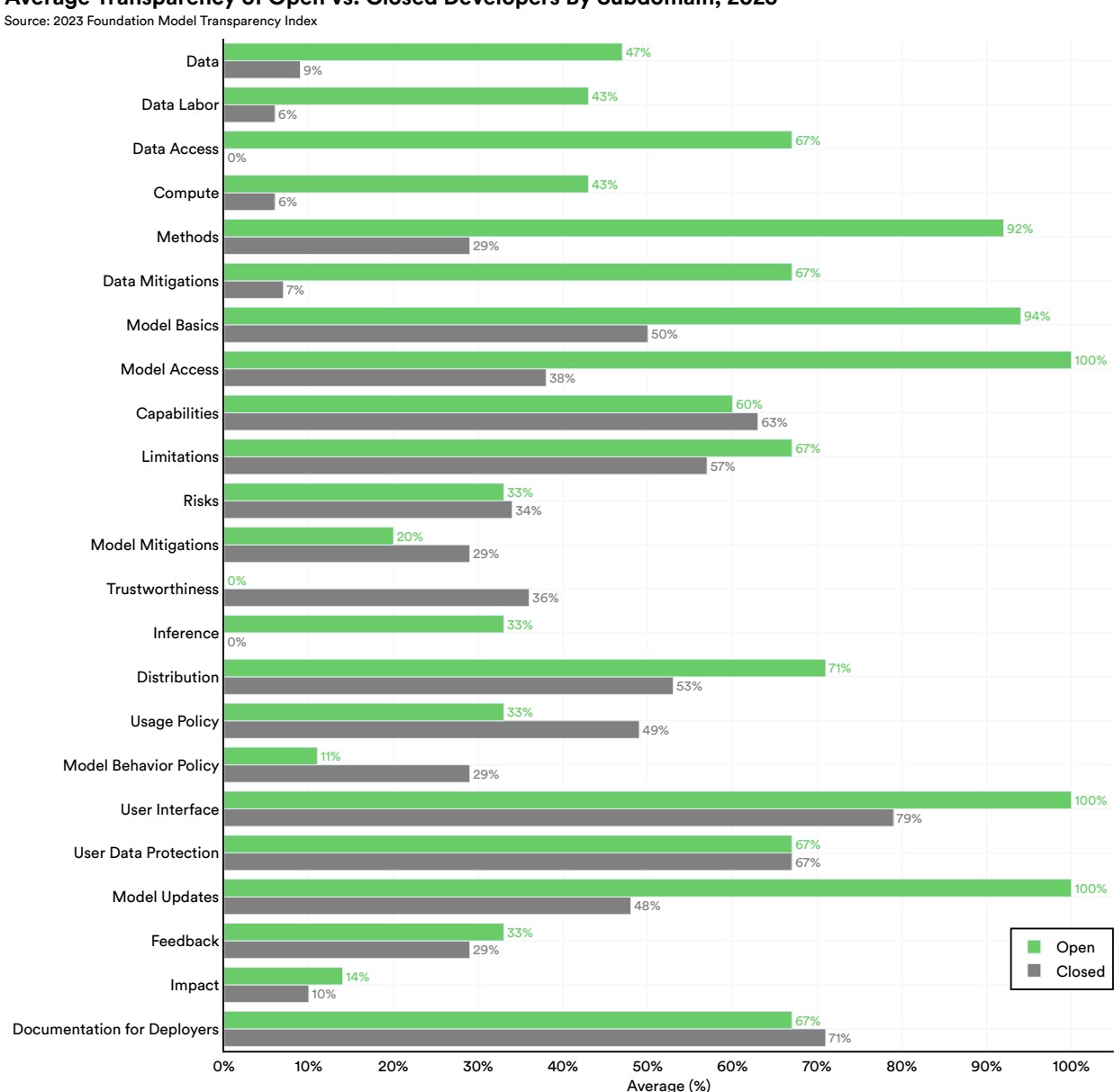

Figure 13: **Open vs. Closed by Subdomains.** The mean score for the 3 open developers (Meta, Hugging Face, Stability AI) and the 7 closed developers (OpenAI, Anthropic, Google, Cohere, AI21 Labs, Inflection, Amazon) across each of the 23 subdomains. Note: the number of indicators per subdomain varies widely.

may be higher-resourced or face stronger incentives to proactively address certain matters around responsible AI (e.g. Risks, Model Mitigations, Trustworthiness). In addition, closed developers often have a closer coupling between the foundation model we assessed and downstream services, meaning that certain user-related aspects of transparency are potentially of higher priority (namely the Usage Policy). For example, many closed developers provide products built on top of their flagship foundation model, providing users of their platforms and clients who license their proprietary foundation models with an opportunity to push for transparency.

The mean score of open developers is higher than closed developers on every upstream subdomain, with major score differentials especially for the Data, Compute, and Methods subdomains. Looking at the difference in average scores by release strategy, we see large disparities in favor of open models in each domain, with the largest gaps for Data Access (67% to 0%), Methods (92% to 29%), and Data Mitigations (67% to 7%). We also observe similar large differentials (40%+) for Model Basics, Model Access, and Model Updates. While less stark, we highlight the superior transparency on average for the Distribution subdomain as especially surprising given that closed developers maintain greater control over distribution by virtue of being closed.

**Indicator-level analysis further demonstrates the disparity between open and closed developers.** At the indicator level, the median open developer outscores the median closed developer on 28 indicators (18 upstream, 7 model, 3 downstream), while the median closed developer scores higher on just 6 indicators (0 upstream, 2 model, 4 downstream). The median open developer and the median closed developer both score points on 22 indicators and neither scores points on 44 indicators.

**The open developers we assessed provide greater transparency than their closed counterparts.** Overall, each level of analysis points in the same direction: open developers are reliably more transparent. In particular, we highlight that the release of assets (e.g. model weights, data, code) may be significantly underweighted in terms of its broader transparency effects. Our findings dispel the belief that closed developers are more likely to be transparent about downstream matters due to their greater control over deployment, while emphasizing that both open and closed developers continue to be extremely opaque in terms of the downstream impact of their foundation models. With this in mind, we caution that our assessment is necessarily based on the practices of some of the highest-resourced open and closed developers, so these trends should not be taken as sufficient evidence to claim that all open developers are more transparent than closed developers. And we believe there is ample opportunity for closed developers to address these gaps in transparency as we discuss in §8.1: RECOMMENDATIONS-DEVELOPERS.

### 7.6 Correlations between companies

**Measuring correlations.** The $100 \times 10$ scores introduces data-driven structure. In particular, it clarifies relationships that arise in practice between different regions of the index. Here, we consider the *correlations*, in scores, focusing on company-to-company similarity for simplicity. For example, if two companies receive similar aggregate scores, is this because they satisfy all the same indicators or do they score points on two very different sets of indicators?

In Figure 14, we plot the correlation between every pair of companies. To measure correlation, we report the simple matching coefficient (SMC) or the agreement rate. The SMC is the fraction of the 100 indicators for which both companies receive the same score (i.e. both receive a zero or both receive a 1). As a result, a SMC of 0 indicates there is no indicator such that both companies receive the same score and a SMC of 1 indicates that for all indicators both companies receive the same score. For this reason, the correlation matrix is symmetric and guaranteed to be 1 on the diagonal.

To systematically analyze the results, we consider three patterns in the correlation matrix: (i) individual cells with very small or very large values (i.e. highly similar or highly dissimilar company pairs), (ii) individual rows with consistently small, consistently large, or highly varied values (i.e. unusual companies), and (iii) structural patterns across the correlation matrix.

**Strongly correlated company practices.** In terms of the most correlated company pairs, we identify a few regions of the correlation matrix. First, we identify the three most correlated pairs: (Cohere, AI21

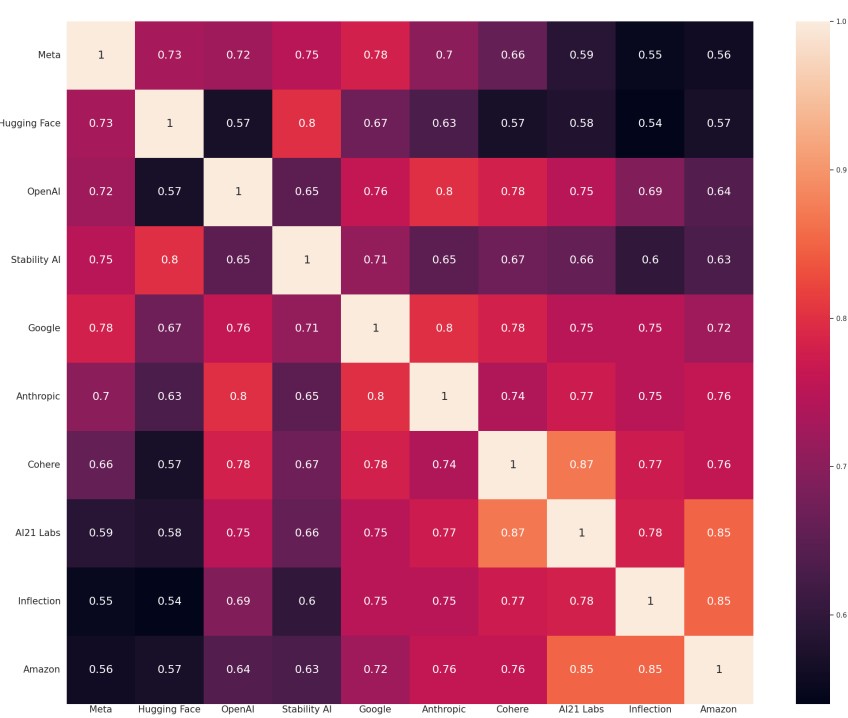

Figure 14: **Correlations between Companies.** The correlation between the scores for pairs of companies across all indicators. Correlation is measured using the simple matching coefficient (i.e. agreement rate), which is the fraction of all indicators for which both companies receive the same score (i.e. both receive the point or both do not receive the point).

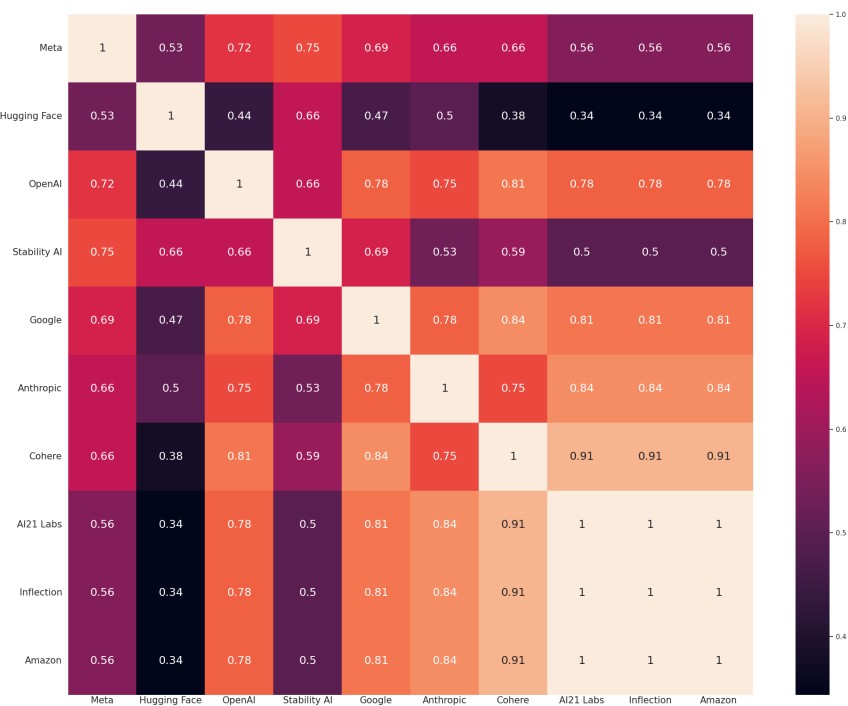

Figure 15: **Correlations between Companies (Upstream Indicators).** The correlation between the scores for pairs of companies across all indicators when only considering upstream indicators. Correlation is measured using the simple matching coefficient (i.e. agreement rate), which is the fraction of all indicators for which both companies receive the same score (i.e. both receive the point or both do not receive the point).

Labs; SMC = 0.87), (AI21 Labs, Amazon; SMC = 0.85), and (Inflection, Amazon; SMC = 0.85). These pairs are all among the four lowest-scoring companies, though we note the inclusion of Cohere is interesting given Cohere's overall score (34) is closer to the average (37) and the middle-scoring group of companies (i.e. including Google and Anthropic). In addition to these pairs, if we consider the other highly-correlated pairs (SMC ≥ 0.8), we identify: (Hugging Face, Stability AI; SMC = 0.80), (OpenAI, Anthropic; SMC = 0.80), and (Google, Anthropic; SMC = 0.80). In particular, we observe that the company correlations identify clear structure: Hugging Face and Stability AI are the only two developers to release both data and models openly, and the trio of OpenAI, Google, and Anthropic are the three members of the Frontier Model Forum that we assess.

**Weakly correlated company practices.** In contrast, we see that the least correlated pairs (SMC < 0.6) are pairs involving Meta and the three lowest-scoring developers as well as pairs involving Hugging Face and five of the seven closed developers (OpenAI, Cohere, AI21 Labs, Inflection, Amazon). These are all pairings between an open and a closed developer. More broadly, we highlight that Meta is the sole developer that is not correlated with SMC at least 0.80 with any other developer, with the most similar other developer being Google at 0.78 (see below for further analysis). This means Meta is rather unique in terms of the indicators where it scores points; it is the sole developer that is not strongly correlated with any other company, even including the two other open developers. Nevertheless, the least correlated pair of companies still agrees in over half the indicators (SMC = 0.54), which is not surprising given that all the companies are opaque (e.g. if all the companies all scored 0 on every indicator, they would necessarily be perfectly correlated with SMC = 1).

**Upstream correlations.** In Figure 15, we plot the correlation between every pair of companies when considering only indicators from the upstream domain. Since the four lowest-scoring companies overall

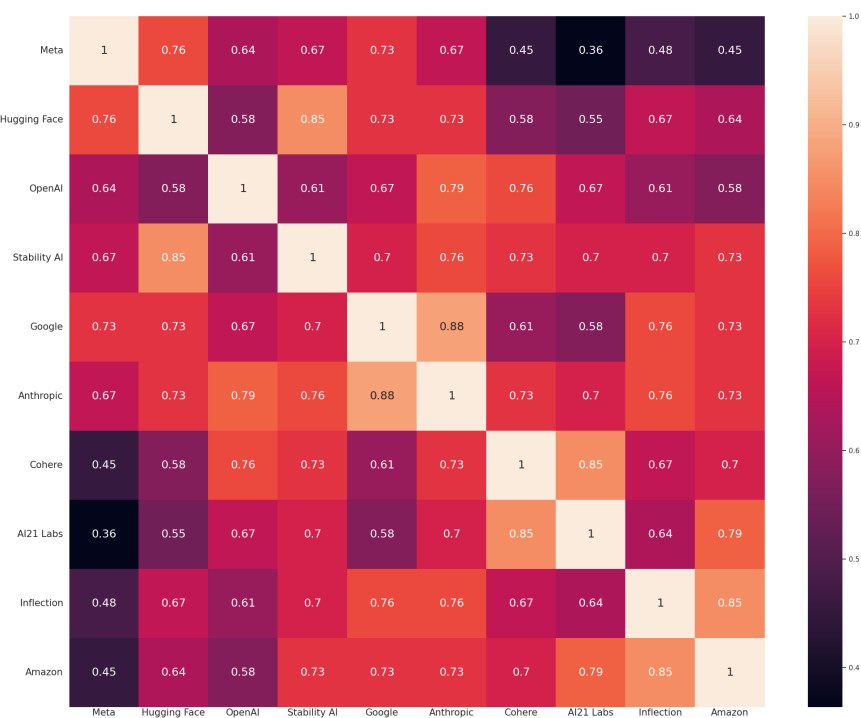

Figure 16: **Correlations between Companies (Model Indicators).** The correlation between the scores for pairs of companies across all indicators when only considering model indicators. Correlation is measured using the simple matching coefficient (i.e. agreement rate), which is the fraction of all indicators for which both companies receive the same score (i.e. both receive the point or both do not receive the point).

also score zero (or near-zero in the case of Cohere) points on the upstream indicators, they are necessarily extremely correlated. For the same reason, the extent to which the remaining six companies are correlated with the three lowest-scoring companies is precisely proportional to their own opacity on the upstream domain. Looking at the three companies that score in the middle overall (Google, Anthropic, Cohere), we see their indicator-level transparency is reasonably correlated. We also see a similar trend where OpenAI, Google, and Anthropic are correlated, though in this case OpenAI and Cohere are even more correlated with an SMC of 0.81. Interestingly, while the three open developers score much higher overall than any of the seven closed developers for the upstream domain, the correlations between them are somewhat different than in the other domains: there is a weaker correlation between Hugging Face and Stability AI, and Meta's correlation with OpenAI and Stability AI is stronger than its correlation with Hugging Face. Despite the fact that Meta and Hugging Face are the two highest-scoring companies on upstream, they are not especially correlated (SMC = 0.53) in that domain. These discrepancies coincide with the indicators where Hugging Face scores points and the other two open developers (Meta, Stability AI) do not, namely those relating to data sources and data labor. Given the large spread in scores across developers in the upstream domain, we see the related effect that the correlations can be quite variable with some at or near 1 and others well below 0.5 (minimum upstream SMC = 0.34).

**Model correlations.** In Figure 16, we plot the correlation between every pair of companies when considering only indicators from the model domain. In contrast to the upstream correlations, we see a much more varied picture. First, much like the overall correlations, we see strong correlations for (Cohere, AI21 Labs; SMC = 0.85) and (Inflection, Amazon; SMC = 0.85) but not necessarily for the other pairs between these four companies. Among the three Frontier Model Forum companies, we see a very strong correlation of 0.88 between Google and Anthropic, a fairly high correlation of 0.79 between OpenAI and Anthropic, but a considerably lower correlation for the third pair of OpenAI and Google at 0.67. These trends, where Anthropic is highly correlated with both, but OpenAI and Google are not necessarily correlated, mirror what

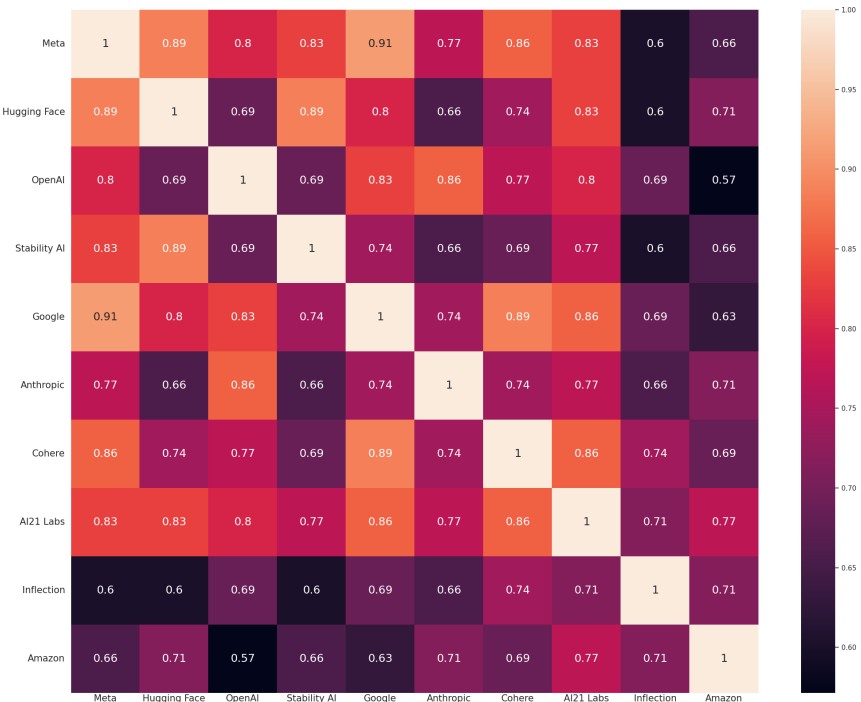

Figure 17: **Correlations between Companies (Downstream Indicators).** The correlation between the scores for pairs of companies across all indicators when considering only downstream indicators. Correlation is measured using the simple matching coefficient (i.e. agreement rate), which is the fraction of all indicators for which both companies receive the same score (i.e. both receive the point or both do not receive the point).

we observe for the overall correlations. Similar to what we observed for the overall correlations, Hugging Face and Stability AI are quite correlated as well with a correlation of 0.85, and Meta is not particularly correlated with any company (the highest is Hugging Face at 0.76).

**Downstream correlations.** In Figure 17, we plot the correlation between every pair of companies when considering only indicators from the downstream domain. The downstream correlations surface considerably different trends from the overall correlations or those for the other two domains. In particular, we first highlight that Meta is strongly correlated with Google in their scores on downstream indicators. Given that several of the downstream indicators related to broader corporate practices, the similarities between these companies may contribute to this result, though both companies are not strongly correlated with Amazon, the other Big Tech company we assess. Relatedly, we see fairly strong correlations between OpenAI and Anthropic, which again may relate to their fairly similar business practices mapping onto specific downstream indicators (e.g. indicators in the Model Behavior Policy subdomain). On the other hand, akin to the upstream subdomain, we see that Inflection is especially dissimilar from all of the open model developers (Meta, Hugging Face, Stability AI). And, unlike the other correlation matrices, OpenAI and Amazon are more dissimilar than usual. Overall, while we do not observe it as clearly in the other correlation analyses, here we see all three pairs of open developers are highly correlated: (Meta, Hugging Face; SMC = 0.89), (Meta, Stability AI; SMC = 0.83), (Hugging Face, Stability AI; SMC = 0.89). This may reflect that all open developers have shared transparency challenges on specific indicators within the downstream domain (e.g. monitoring mechanism and model behavior policy enforcement), perhaps stemming from the weaker control they have over downstream use. Overall, we find the complex structure and heterogeneity in the correlation for the downstream domain especially intriguing, given the aggregate scores for this domain are the most tightly clustered (see §7.4: DOWNSTREAM-RESULTS). That is to say, disaggregated indicator-level analysis is especially revealing for this domain compared to domain-level analysis.

# 8    Recommendations

The design of our indicators, execution of our assessment, and analysis of our results provide a rich supply of ideas for how to improve transparency. We center our attention on foundation model developers and deployers, along with policymakers. For each group of stakeholders, we provide concrete recommendations on the basis of this research. Additionally, we encourage researchers to scrutinize our overall approach in order to clarify how transparency for foundation models can be better understood in the future.

## 8.1    Recommendations for foundation model developers

By directly scoring foundation model developers, we provide explicit feedback on where developers are and are not transparent. In itself, this provides immediate guidance for these 10 companies in the context of their current flagship foundation models. Moreover, the Foundation Model Transparency Index provides valuable insights for these companies to consider related to their other models and future releases; it also has bearing on how foundation model developers that we did not assess can promote transparency. We provide 10 specific recommendations for foundation model developers.

1. **Increase transparency for existing foundation models.**

   - As our results show, the development and use of major foundation model developers' current flagship models is opaque. Developers should remedy this situation by releasing more information about the systems that are central to today's foundation model ecosystem. Increasing the transparency of future releases will not resolve this issue as there are hundreds of products, services, and other models that are built on top of today's flagship models.[61]

   - Developers should begin by focusing on low-hanging fruit, such as clarifying ambiguous language in their documentation, centralizing existing information sources, and sharing information about models that poses minimal concerns related to market competitiveness or legal risk. [62] Developers should also be clear about why they will not release certain information about their foundation models; developers should explicitly state the subdomains where they do not release information and explain why they do not do so.

2. **Increase transparency for future foundation model releases.**

   - Developers should substantially increase the transparency of future foundation model releases. Wherever possible, they should publicly disclose information related to the 100 indicators we outline as well as additional information they feel is important to share with the industry, the public, and governments. This might look like taking a transparency-first approach in which the developer prioritizes transparency throughout the model development process and includes transparency as an important performance metric for research teams.[63]

---

[61]Developers that signed on to the White House's first round of voluntary commitments (including Amazon, Anthropic, Google, Inflection, Meta, and OpenAI) have pledged only to improve transparency "for all new significant model public releases within scope," where the scope is defined as "generative models that are overall more powerful than the current industry frontier (e.g. models that are overall more powerful than any currently released models, including GPT-4, Claude 2, PaLM 2, Titan and, in the case of image generation, DALL-E 2)." Developers that signed on to the White House's second round of voluntary commitments (including Cohere and Stability AI) have pledged only to improve transparency for "generative models that are overall more powerful than the current most advanced model produced by the company making the commitment." See https://www.whitehouse.gov/wp-content/uploads/2023/07/Ensuring-Safe-Secure-and-Trustworthy-AI.pdf and https://www.whitehouse.gov/wp-content/uploads/2023/09/Voluntary-AI-Commitments-September-2023.pdf

[62]For example, Anthropic released significantly more information about Claude 2 than its previous flagship model, Claude, including in the form of a model card.

[63]One relevant analogy is to the development of open foundation models. Much as some developers begin the process of building a foundation model with the intention of making all model assets openly available, then subsequently decide if the risks of making a model asset openly available outweigh the potential benefits, developers could begin the development process with the assumption of maximum transparency and remove only some items along the way (Klyman, 2023).

- Profit-oriented developers commonly argue that certain forms of transparency can endanger their competitive advantage. Nevertheless, developers have a basic responsibility to weigh this concern against the the risks posed by their technology to society and the benefits of increasing societal understanding of this technology via transparency. These risks should determined by not only the developer but also the assessment of third party experts. Voluntary access for *independent*, third party audits (i.e. auditors not selected by the developer itself), can achieve a greater degree of transparency, and safeguard competition concerns with non-disclosure agreements. We would also argue audits are not always a good substitute for public transparency, and developers' arguments around competitive advantage should be carefully assessed for each indicator of transparency. These arguments are a common refrain to avoid meaningful community discussion about widespread practices that do not in actuality endanger competitive advantages.

3. **Follow industry best practices with respect to transparency.**

   - Our findings suggest that every developer could significantly improve transparency by drawing on different approaches to transparency from across the industry. At least one developer scores points on 82 of our 100 indicators: where developers are struggling to increase transparency in a specific issue area, they should look to developers that have already done so.

   - While the foundation model ecosystem is nascent, some developers have outlined best practices for responsible development that relate to transparency. For example, in their "Joint Recommendations for Language Model Development," OpenAI, Cohere, and AI21 Labs state that developers should "publish usage guidelines and terms of use ... document known weaknesses and vulnerabilities ... [and] model and use-case-specific safety best practices." (See Appendix D for additional examples of calls from developers for transparency.)

4. **Work with deployers to increase transparency.**

   - In cases where a developer is not the sole deployer of a foundation model, the developer should partner with deployers to increase transparency. For example, developers should attempt to require that deployers disclose usage statistics and provide usage disclaimers. Developers might do so through legal agreements that they sign with deployers that grant deployers the right to offer the foundation model. If a developer has little leverage over larger deployers it should consider partnering with similarly situated developers to increase their collective bargaining power. Without such efforts, it may be difficult for a developer to assess the downstream impact of its foundation models.

5. **Work with downstream developers to increase transparency.**

   - Foundation model developers should make it easy for downstream developers to be transparent in their release of fine-tuned models. In addition to increasing transparency for their own models, foundation model developers should release documentation to help downstream developers be more transparent and actively encourage them to do so.

6. **Work with regulators to increase transparency.**

   - While we believe that the public is entitled to information about each of the indicators of transparency that we examine, we recognize that it is unlikely that every foundation model developers will publicly release all of this information. In some cases, foundation model developers may argue the risks of disclosing such information are too great to justify public release. In many such cases, developers should still share this information with regulators such that governments have sufficient information to adequately scrutinize developers in the public interest.

7. **Use transparency to improve trust, safety and reliability.**

   - Sharing internal practices, documentation, and details about risks can lead to short term criticism and negative media coverage, but in the long term it can foster greater community trust than is possible with a more opaque approach. Investigative journalists will eventually expose practices that lead to systemic harms, and these harms are often exacerbated the longer they remain hidden, as illustrated by the Facebook Files Hagey & Horwitz (2021). Foundation models are technologies that could cause widespread harm, and the evidence suggests that safety and reliability will require dedicated and strong forms of transparency from foundation model developers.

8. **Dedicate resources to continue improving transparency over time.**

   - As technologies and risks rapidly evolve, the varieties of and baselines for meaningful transparency will also change. Well-resourced developers should dedicate personnel to adapting their documentation and releases to take account of this shifting landscape, rather than adhering to static benchmarks. Low-resourced developers should seek out funding in order to similarly improve transparency.

9. **Work to improve transparency in the foundation model ecosystem.**

   - There are many areas where transparency is sorely needed, ranging from the downstream impact of foundation model releases to the use of human labor in producing the data used to build foundation models. One cross-cutting issue is the fact that developers do not exist in a vacuum: the foundation models a developer releases depend on and significantly affect other parts of the ecosystem. Taking this into account, developers should increase transparency as a means of improving the health of the overall ecosystem.

   - Developers should use semantic versioning for their models (as is the norm in software engineering) such that there is no ambiguity as to the version of the model that is being distributed. Developers should also give as much notice as is practicable (e.g. 3 months notice) in advance of deprecating models in order to give any downstream dependencies adequate time to migrate to a new version.

   - Developers should release an artifact alongside their foundation models that includes information about models' upstream and downstream dependencies (Bommasani et al., 2023b). Information about the datasets, software frameworks, and applications the model depends upon, as well as products, services, and other models that depend upon the model, are essential for effective supply chain monitoring.

10. **Use the Foundation Model Transparency Index to increase transparency.**

    - The Foundation Model Transparency Index provides an extensive taxonomy of the key elements of transparency in the field. We encourage developers to score their non-flagship models on the index and see where they have room for improvement.

    - Each indicator contains significant detail that developers can utilize to increase transparency in specific issue areas. For quantitative metrics, indicators include information regarding the appropriate unit of measurement and level of precision. For qualitative metrics, indicators often provide de facto instructions for how to clearly share information about a specific subdomain with the public.

### 8.2 Recommendations for foundation model deployers

Foundation model developers are not the only actors with a responsibility to promote transparency: deployers of foundation models such as cloud services providers and companies that license foundation models from developers also have a significant role to play. Although deployers cannot unilaterally increase transparency as they are not the party responsible for building a foundation model, there are still some tools at their disposal for doing so and they should think seriously about the implications of relying on systems for which there is little publicly available information.

1. **Assess the risks of deploying a foundation model without adequate transparency.**

   - Deployers that make use of a developer's foundation model in their products and services should conduct pre-deployment risk assessments that include specific assessments of risks stemming from a lack of transparency. These risks may include increased legal liability for difficult-to-explain model behaviors, reduced trust from users due to the product's opacity, and lower product performance without adequate information about the data used to build the model.

2. **Require sufficient transparency in working with foundation model developers**

   - Foundation model deployers should work with developers to increase the level of transparency regarding their models. It is not only in a deployers' interest for developers to share information bilaterally, but also for developers to be transparent with the public about the risks and limitations of their models. Deployers themselves can help developers increase transparency by sharing usage statistics.

   - Deployers should go beyond information sharing requests to improve transparency. For example, deployers should aim to negotiate contracts with developers that require developers to publicly share information that is relevant to the developers' customers as well as the broader public, such as information regarding Model Updates, changes in Usage Policy, and Impact. In cases where deployers have little leverage over larger developers they should consider partnering with similarly situated deployers to increase their collective bargaining power.

3. **Do not put undue trust in opaque foundation models.**

   - Some deployers may take a foundation model from a reputable company at face value, assuming that all of the relevant information about that system is available to deployers and regulators. This could be a serious misjudgment: as our findings show, developers are overwhelmingly not transparent about the development and use of their foundation models. Assuming that a model complies with regulatory requirements regarding information sharing could come with substantial legal risk; for example, if new regulations primarily place information sharing requirements on deployers, they may face legal exposure related to their deployment of opaque foundation models. While developers are presumably more transparent in their relationships with deployers than in their public facing documentation, this is no guarantee that relevant information is shared across the 23 subdomains we identify.

### 8.3 Recommendations for policymakers

Policymakers across the United States, China, Canada, the European Union, the United Kingdom, India, Japan, the G7, and many other governments have already taken specific actions on foundation models and generative AI (see Appendix D). Evidence-driven policy that is grounded in a rich and sophisticated understanding of the current foundation model market is likely to achieve the best outcomes. As a result, our extensive characterization of transparency provides three core insights: (i) what aspects of transparency are present in status quo absent regulatory intervention, (ii) if mandated and enforced, what aspects of transparency would change relative to the status quo, and (iii) what substantive requirements beyond transparency would be most appropriate given the newfound transparency? We hope that lawmakers will draw on the information we aggregate in the Foundation Model Transparency Index to better inform policy initiatives. To be clear, our intent is to not to make a claim about whether specific governments should or should not regulate foundation models at this time, though some policy intervention is likely needed. Nor is our intent to recommend broad disclosure requirements, which could cause substantial harm if they are implemented without regard for differences in developers' business models and their level of financial resources, or without adequate government support for regulatory compliance. Our view is that a better understanding of the status quo will lead to smarter policy, which leads to the following recommendations.

1. **Transparency should be a top priority for AI legislation.**

   - Mechanisms to promote transparency should be among the suite of policy tools that lawmakers use to encourage responsible development of foundation models (Engler, 2023; Hacker et al., 2023). Unlike many other policy tools, transparency can be relatively low cost—where developers already possess the relevant information, sharing it does not require data collection. Another advantage of pro-transparency policies are that they can help solve collective action problems with respect to sharing information with the public. If one developer shares much more information about its foundation model with the public, then it could theoretically be penalized by investors or scrutinized by regulations for having more information about the model's risks and limitations publicly available than its competitors. As a result, a developer may be hesitant to be a first mover on transparency if its competitors are steadfast in maintaining opacity. By contrast, if that developer's peers must also share information about the risks and limitations of their foundation models, there is much less potential for transparency to represent a competitive disadvantage.

   - Transparency is a fundamental prerequisite for accountability, robust science, continuous innovation, and effective regulation. With additional information about companies' business practices, the impact of their foundation models, the resources used to build models, and the AI supply chain, governments would be much better positioned to enact comprehensive AI regulations.

   - Policymakers have a responsibility to ensure that the public has adequate information about extremely powerful AI systems that hundreds of millions of people use.

2. **Regulators should enforce existing regulation to promote transparency for foundation model developers.**

   - Governments already have substantial authority to require companies to share information about their business practices Ho (2012); Hess (2019); Irion (2022). For example, in recent years data protection authorities have increased their efforts to regulate the development and use of AI (Zanfir-Fortuna, 2023); they should consider using these authorities to solicit additional information from foundation model developers regarding the data they use to build foundation models and the labor that goes into producing that data. Similarly, sectoral regulators should consider scrutinizing the deployment of foundation models within their purview and require transparency where appropriate.

3. **Policymakers should be realistic about the limits of transparency.**

- Transparency is not an end in itself. While having more information about companies' business practices and the foundation model ecosystem will undoubtedly be helpful, the most significant benefits from transparency will stem from the ways in which it elicits changes in business practices and promotes responsible development and use of foundation models.

- Transparency is not a viable alternative to substantive change. Some interest groups and policymakers have nonetheless pushed for transparency requirements as a form of "light-touch" AI regulation. Rather than mandating that companies change their policies and practices, this approach would merely require some level of information sharing with the government. But transparency is only useful insofar as the information it yields is actionable. Increased transparency can help policymakers have sufficient information about the state of the industry that many governments seek to regulate.

- While transparency requirements may appear more feasible and even-handed than other policy interventions in the near term, policymakers should recognize that they are likely insufficient to reduce harm in many areas. Even if companies share more information about the impacts of their models on workers and the environment, that may not lead them to improve working conditions or reduce emissions. Policymakers should consider measures beyond transparency requirements in a wide variety of areas while balancing other important equities related to competition and algorithmic justice.

**4. Governments should craft a policy architecture that enables responsible development of open foundation models, which will in turn promote transparency.**

- Open foundation models are more transparent than closed foundation models, often by a significant margin. This means that policymakers with an interest in transparency should be hesitant to impose regulations on foundation model developers or deployers that make it considerably more difficult to build open foundation models. Measures that substantially increase the legal risk of developing open foundation models by holding foundation model developers liable for model outputs or by requiring comprehensive monitoring of downstream use may ultimately undermine transparency.

- Pro-competitive policies such as those that encourage a variety of different business models in the foundation model ecosystem can promote transparency. If there are only a few major technology companies that develop flagship foundation models, it will be easier for those companies to circumvent transparency rules by coordinating their activities. For instance, a handful of major closed developers could agree that a certain level of transparency is sufficient to satisfy their goals and to meet regulatory requirements, leading them to obfuscate their business practices in similar ways. If the foundation model ecosystem is dominated by a few incumbents, it will also be easier for those incumbents to jointly engage in regulatory capture as there will be no countervailing narrative from other developers in the ecosystem. By contrast, policies that result in a diverse array of open and closed foundation model developers could create a positive feedback loop for transparency. The higher level of transparency of open developers can help draw attention to the lack of information available about the resources required to build closed foundation models. Some closed developers in this environment may see it as in their interest to share more information about their models in order to engender more trust in their products and services, which can in turn push less transparent closed developers to alter their business practices.

# 9    Impact

The Foundation Model Transparency Index characterizes the transparency of foundation model developers at present. While this descriptive work already yields significant insights and value, our ambition for this work is to drive change. In short, our objective is to *improve* transparency in the foundation model ecosystem: we believe improved transparency will result in better science, more innovation, greater accountability, and ultimately give society greater collective confidence that this promising technology can truly advance the public interest. To achieve these lofty goals requires changing the conduct of powerful organizations. As with many similar efforts to drive change, we have conceptualized a specific theory of change and have considered specific limitations and risks of our work. Consistent with the work's spirit of transparency, we describe both matters plainly below.

## 9.1    Theory of change

Assessment of any kind naturally characterizes the status quo. However, our intent is for our assessment to drive change, especially given that our most fundamental finding is that there is insufficient transparency in the foundation model ecosystem. Bommasani (2023) argues that evaluation and assessment can drive change if there is sufficient uptake: we specifically articulate how assessment can motivate improvement through different forms of uptake.

**Assessment motivates improvement.**    By quantifying transparency and simultaneously scoring many developers, we hope that these organizations will improve their transparency scores over time. We aim to provide a characterization of the status quo that is broadly legible across the ecosystem by directly comparing organizations' transparency scores. Furthermore, specific areas where there is pervasive opacity, or where specific companies are less transparent than their counterparts, are prime targets for public pressure and scrutiny.

With respect to AI companies, we believe that certain teams and employees will play an outsized role in shaping transparency practices. Namely, responsible AI teams along with other teams that address ethics and safety are likely to shape many company-wide practices on transparency, including for their flagship foundation models. For this key group of individuals, we hope the Foundation Model Transparency Index provides a concrete list of indicators to proactively consider in making decisions. At present, we believe that some companies are not transparent about certain issue areas not because of specific countervailing concerns (e.g. profits, privacy, safety) but because they have not explicitly considered whether they should be transparent on this issue. In these cases, we believe the index provides a structured, well-argued resource that responsible AI teams can directly consider in making decisions around transparency. The index also provides an extensive account of why these specific indicators are valuable, which could help responsible AI teams advocate for greater transparency within their organizations. To be concrete, linking outward-facing transparency reporting with internal-facing company tracking could be a natural outcome where our index could bring about desired change while adding minimal overhead for these companies.

Indexes draw power from their subsequent iterations, allowing for improvements to be clearly measured and acknowledged over time. In the fast-moving foundation model ecosystem, subsequent versions could be motivated by (i) changes in the indicators, (ii) changes in the key companies to assess, (iii) changes in the flagship foundation models of those companies, and (iv) changes in the underlying materials for a specific company. As a result, we believe the maintenance and subsequent versions of the Foundation Model Transparency Index will be necessary for its sustained impact. We have not yet determined an exact cadence and strategy, though we will conduct future versions of the index.[64]

**Assessment guides standards and mandates.**    Fundamentally, the Foundation Model Transparency Index assesses companies on metrics of transparency that are selected and evaluated based on our judgments as experts in this domain. With this in mind, the indicators selected as well as the results could directly inform more formal processes. For instance, policymakers around the world are considering a spectrum of voluntary commitments, industry standards, and mandatory requirements for foundation model developers.

---

[64]See `https://crfm.stanford.edu/fmti` for the latest details.

Many current policy efforts, across the varying levels of voluntary and mandatory requirements, explicitly name transparency as a top-level priority and directly identify specific indicators and subdomains covered by our index (see Appendix D).

Our index is likely to be more comprehensive, more fine-grained, and more empirically grounded than most ongoing policy initiatives. As a result, our index provides direct value to policymakers. In selecting requirements, policymakers can use the index to explore the broader universe of potential transparency and disclosure requirements. In defining requirements, policymakers can use the index to explore specific definitions as well as edge cases that may complicate a requirement. And, in ultimately deciding which organizations to regulate and how to enforce regulations, policymakers can look at the current status quo to efficiently allocate resources.

As a brief example, consider the European Parliament's position for the EU AI Act,[65] which was adopted by the Parliament on June 14, 2023 by a vote of 499 in favour, 28 against and 93 abstentions.[66] Bommasani et al. (2023a) provide an initial analysis of potential compliance with the the Act as proposed in the context of foundation model developers. Given this legislative proposal, European lawmakers might recognize that topics related to upstream labor and downstream impact, which are covered in the Foundation Model Transparency Index, are not adequately addressed in the draft AI Act. Policymakers might also acknowledge that requirements to disclose a summary of any copyrighted training data are too vague and a more specific definition, such as the definition we provide in Appendix B, may be desirable to improve compliance. And, finally, policymakers might view the results of how open and closed developers fare in deciding which requirements are best targeted at which developers along the release spectrum. Overall, much as transparency is instrumental for key societal objectives like public trust, we believe the Foundation Model Transparency Index can be similarly instrumental for key societal processes like sound policy-making.

## 9.2    Limitations and risks

**Equating transparency and responsibility.**    Because we foreground transparency in our assessment of developers and their flagship models, it is likely that some will misinterpret the Foundation Model Transparency Index as a measure of the responsibility of companies. This is not the case for a number of reasons; most importantly, we award points on the basis of whether a developer is transparent about each indicator, not whether it has responsible business practices tied to that indicator. Concretely: if a developer discloses that it pays data laborers just one cent per hour, it would score points on the wages indicator under our methodology, while a developer that pays data laborers $20 an hour but does not make that information publicly available would score no points.

This means that one risk of our approach is that it could incentivize developers to be transparent in performative ways that merely increase the amount of information available about their flagship models but do not reflect an effort on the part of the developer to substantively improve its business practices. Nevertheless, we believe that additional information about each of these indicators is an invaluable first step towards understanding how developers build and use foundation models. This will in turn allow many other evaluations of responsible business practices, in which the level of transparency should be but one factor.

**Transparency-washing.**    There is no guarantee that improved transparency of foundation models will result in more responsible development. As critics of transparency-washing have persuasively argued (Zalnieriute, 2021), major technology companies have used transparency to create the illusion that they are responsible players with the public's best interest at heart. In this way, transparency can be a shield against further scrutiny, helping to convince the public that foundation models are safe and trustworthy when they may not be.

Similarly, companies may use transparency as a shield against comprehensive regulation. Companies could face substantial costs if they were required to increase pay for data laborers or forego certain risky use cases for their foundation models, leading some to argue that governments should simply require transparency in

---

[65]https://www.europarl.europa.eu/doceo/document/TA-9-2023-0236_EN.pdf
[66]https://www.europarl.europa.eu/news/en/press-room/20230609IPR96212/meps-ready-to-negotiate-first-ever-rules-for-safe-and-transparent-ai

these verticals. However, transparency alone will not change a business' fundamental incentives and, if used to water down regulation, can perpetuate harm. Notwithstanding this risk, transparency may be a more appropriate regulatory option for many of the indicators we consider given the early stage of the foundation model ecosystem and the risk that substantive requirements will disproportionately damage small and open developers.

**Gaming the index.**  Moving forward, developers might attempt to game the Foundation Model Transparency Index without actually improving transparency. They could do this by clarifying that they do not share information about certain practices and giving a justification for doing so. Developers might also exploit the fact that indicators are relatively generous, meaning that they could share minor additional information on indicators that are comparatively easy to satisfy without meaningfully improving transparency. Since scores could theoretically be gamed in this way, it is important to consider the Foundation Model Transparency Index in conjunction with other metrics of companies' business practices.

**Binary scoring.**  The fact that each indicator is binary limits the amount of information that each score can reflect when compared with more expressive scoring schemes. For the same indicator, it is often the case that several developers share much less information than others but they all score one point nonetheless as they cross the threshold for receiving points. Conversely, in certain instances developers disclose some information related to a particular indicator but it is insufficient to receive points, yet they are grouped alongside developers who disclose no information whatsoever about an indicator. We attempt to address this limitation by breaking complex indicators into discrete chunks, meaning that each indicator assesses one key dimension of transparency and can more easily be made binary.

**The models we assessed are predominantly language models.**  For the developers we assess, their associated flagship models are predominantly text-to-text language models (8 of the 10). Of the remaining two, only one includes images as an input (GPT-4) and only one outputs images (Stable Diffusion 2). None of the flagship models we considered include modalities beyond text and images, though these modalities may become more common in the coming years. With this in mind, in principle the indicators are chosen and defined in a largely modality-agnostic fashion to facilitate future assessment as the flagship models in the ecosystem diversify in terms of modalities.

**Most companies we assessed are headquartered in the U.S.**  Of the 10 developers we assess, 7 are headquartered in the United States. Although this reflects the disproportionate global reach of U.S. technology companies, there are salient foundation model developers in other parts of the world that we did not assess in this work. For instance, the index excludes foundation model developers in East Asia that we believe are sufficiently important to evaluate, but they often did not share enough information publicly to even attempt evaluation. We also did not consider Falcon-180B from the Technology Innovation Institute in Abu Dhabi,[67] as we had already finalized our evaluations when the model was released in September. We hope that researchers will use Foundation Model Transparency Index and our fully transparent methodology to assess the transparency of these developers as well as others around the world.

**Low bar for awarding points.**  We were generally quite generous in the scoring process. When we determined that a developer scored some version of a half-point, we usually rounded up. Since we assess transparency, we award developers points if they explicitly disclose that they do not share information about a particular indicator. We also read developers' documents with deference where possible, meaning that we often awarded points where there are grey areas. This means that developers' scores may actually be higher than their documentation warrants in certain cases as we had a low bar for awarding points on many indicators.

**Uneven costs.**  In designing the Index, we score all companies on the same set of indicators. In turn, when companies are compared on the basis of their scores, we recognize that the relative and absolute costs of achieving a particular score may differ greatly across companies. For example, companies with larger legal

---

[67]https://falconllm.tii.ae

teams or more employees dedicated to document specific activities are likely to fare better on the Index given many of the indicators (e.g. on company policies, on model documentation) are directly related to the work function of these personnel. Overall, insofar as the Index shapes a company's social status and perception, as well as impacts regulation in the AI industry, the Index may unevenly advantage and disadvantage companies given the underlying disparities in relevant resources like employee headcount. While in some ways this may reflect a fundamental aspect of transparency (i.e. companies require the employees to do the work required to produce and disclose information to be transparent), we express caution in overly focusing on the Index results to the detriment of specific companies.

## 10    Conclusion

Our research establishes the extent to which foundation model developers are transparent, set against the backdrop of decreasing transparency. Our findings show that the status quo is characterized by a widespread lack of transparency across developers, with significant unevenness in how individual developers fare and where they have room for improvement. We take this as a serious indictment of the overall ecosystem. Transparency is a broadly-necessary condition for other more substantive societal progress, and without improvement opaque foundation models are likely to contribute to harm. Foundation models are being developed, deployed, and adopted at a frenetic pace: for this technology to advance the public interest, real change must be made to rectify the fundamental lack of transparency in the ecosystem.

**Acknowledgements.**    We thank Alex Engler, Anna Lee Nabors, Anna-Sophie Harling, Arvind Narayanan, Ashwin Ramaswami, Aspen Hopkins, Aviv Ovadya, Benedict Dellot, Christie Lawrence, Connor Dunlop, Conor Griffin, Dan Ho, Dan Jurafsky, Deb Raji, Dilara Soylu, Divyansh Kaushik, Gerard de Graaf, Iason Gabriel, Irene Solaiman, John Hewitt, Joslyn Barnhart, Judy Shen, Madhu Srikumar, Marietje Schaake, Markus Anderljung, Mehran Sahami, Neel Guha, Peter Cihon, Peter Henderson, Rebecca Finlay, Rob Reich, Rohan Taori, Rumman Chowdhury, Russell Wald, Seliem El-Sayed, Seth Lazar, Stella Biderman, Steven Cao, Tatsu Hashimoto, Toby Shevlane, Vanessa Parli, Yann Dubois, Yo Shavit, and Zak Rogoff for discussions on the topics of foundation models, transparency, and/or indexes that informed the Foundation Model Transparency Index. We especially thank Loredana Fattorini for her extensive work on the visuals for this project, as well as Shana Lynch for her work in publicizing this effort.

**Foundation Model Developers.**    We thank the following individuals at their respective organizations for their engagement with our effort, including involvement in responding to our initial scores on behalf of their organizations. We emphasize that **this acknowledgement should not be understood as an endorsement of the Foundation Model Transparency Index by these individuals**, but simply that they were involved in our engagement with their organizations.

- AI21 Labs. Yoav Shoham

- Amazon. Bratin Saha, Vasi Philomin, Atul Deo, Swami Sivasubramanian, Peter Hallinan

- Anthropic. Jack Clark, Deep Ganguli, Thomas Liao

- Cohere. Aidan Gomez, Danielle Smalls, Seraphina Goldfarb-Tarrant, Nick Jakobi, Saurabh Baji

- Google. Slav Petrov, James Manyika, Kremena Goranova, Sarah Portik, Alexandra Belias

- Hugging Face. Clement Delangue, Meg Mitchell, Yacine Jernite

- Inflection. Mustafa Suleyman, Tim Hwang

- Meta. Joelle Pineau, Melanie Kambadur, Joe Spisak, Eric Smith, Louis Martin

- OpenAI. Miles Brundage, Lama Ahmad

- Stability AI. Emad Mostaque, Ben Brooks

**Funding.**    This work was supported in part by the 2022 Hoffman-Yee program at the Stanford Institute for Human-Centered Artificial Intelligence (HAI).[68] This work was supported in part by the AI2050 program at Schmidt Futures (Grant G-22-63429).

---

[68]https://hai.stanford.edu/2022-hoffman-yee-grant-recipients

**Conflict of Interest.** Given the nature of this work (e.g. potential to significantly impact particular companies and shape public opinion), we proactively bring attention to any potential conflicts of interest, deliberately taking a more expansive view of conflict of interest to be especially forthcoming.

- Betty Xiong is not, and has not, been affiliated with any of the companies evaluated in this effort or any other private sector entities.

- Daniel Zhang is not, and has not, been affiliated with any of the companies evaluated in this effort or any other private sector entities.

- Kevin Klyman is not, and has not, been affiliated with any of the companies evaluated in this effort or any other private sector entities.

- Nestor Maslej is not, and has not, been affiliated with any of the companies evaluated in this effort or any other private sector entities.

- Percy Liang was a post-doc at Google (September 2011–August 2012), a consultant at Microsoft (May 2018–May 2023), and a co-founder of Together AI (July 2022–present). He is not involved in any other companies.

- Rishi Bommasani is not, and has not, been affiliated with any of the companies evaluated in this effort. Rishi is an author of Jernite et al. (2022), as part of the BigScience initiative, that guided the data governance practices for developing BLOOM. As a result, he is also an author on the 350+ author BLOOM paper (Le Scao et al., 2022) that is often cited in the scoring of BLOOMZ.

- Sayash Kapoor worked at Meta until December 2020. He has not since worked for the company.

- Shayne Longpre has three connections to the assessed developers. He has worked as a Student Researcher at Google Brain in 2022, and is an on-going contributor to Cohere For AI, Cohere's non-profit volunteer research organization. Lastly, he was part of the BigScience initiative, where he contributed to BLOOM (Le Scao et al., 2022).

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

## A  Author contributions

This project was a team effort, built on countless contributions from everyone involved. Contributions span the conceptualization, design, execution, analysis, and presentation phases, along with the overarching vision, organization, coordination, and leadership. Below, we describe each team member's contribution.

**Betty Xiong**: Betty contributed to the design, execution, analysis, and presentation.
**Daniel Zhang**: Daniel contributed to the design, execution, and presentation.
**Kevin Klyman**: Kevin co-led the project and coordinated nearly all aspects of the initiative. Kevin contributed to the design, execution, analysis, and presentation.
**Nestor Maslej**: Nestor contributed to the design, execution, analysis, and presentation.
**Percy Liang**: Percy advised the project, shaping the initial vision. Percy contributed to the conceptualization, design, analysis, and presentation along with providing overarching guidance on all stages.
**Rishi Bommasani**: Rishi led the project, bringing together the team, providing the initial vision, and coordinating all aspects of the initiative. Rishi contributed to the conceptualization, design, execution, analysis, and presentation.
**Sayash Kapoor**: Sayash contributed to the design, execution, analysis, and presentation.
**Shayne Longpre**: Shayne contributed to the design, execution, analysis, and presentation.

## B  Indicators

1. Upstream → Data → **Data size**

   - Definition: For the data used in building the model, is the data size disclosed?

   - Notes: Data size should be reported in appropriate units (e.g. bytes, words, tokens, images, frames) and broken down by modality. Data size should be reported to a precision of one significant figure (e.g. 4 trillion tokens, 200 thousand images). No form of decomposition into data phases is required.

   - References: Bender & Friedman (2018a), Gebru et al. (2021)

2. Upstream → Data → **Data sources**

   - Definition: For all data used in building the model, are the data sources disclosed?

   - Notes: To receive this point, a meaningful decomposition of sources must be listed in an understandable way (e.g. named URLs/domains/databases/data providers). It does not suffice to say data is "sourced from the Internet" or comes from "licensed sources".

   - References: Gebru et al. (2021), Hutchinson et al. (2021)

3. Upstream → Data → **Data creators**

   - Definition: For all data used in building the model, is there some characterization of the people who created the data?

   - Notes: While information about data creators may not be easily discernible for some data scraped from the web, the general sources (URLs/domains) should be listed, and, for other data that is bought, licensed, or collected, a reasonable attempt at characterizing the underlying people who provided the data is required to receive this point. The relevant properties of people can vary depending on context: for example, relevant properties could include demographic information like fraction of Black individuals contributing to the dataset, geographic information like fraction of European individuals contributing to the dataset, language information like fraction of L1 English speakers, or occupational information like the fraction of professional artists.

- References: Gebru et al. (2021), Hutchinson et al. (2021)

4. Upstream → Data → **Data source selection**

   - Definition: Are the selection protocols for including and excluding data sources disclosed?

   - Notes: Selection protocols refer to procedures used to choose which datasets or subsets of datasets will be used to build a model. We will award this point even if the selection protocols are non-exhaustive.

   - References: Gebru et al. (2021), Hutchinson et al. (2021)

5. Upstream → Data → **Data curation**

- Definition: For all data sources, are the curation protocols for those data sources disclosed?

- Notes: Curation protocols refer to steps taken to further modify data sources, such as procedures to manage, annotate, and organize data. The aims of curation might include improving the quality, relevance, and representativeness of the data. We will award this point if the developer reports that it does not perform any further curation beyond the data sources.

- References: Gebru et al. (2021), Hutchinson et al. (2021)

6. Upstream → Data → **Data augmentation**

- Definition: Are any steps the developer takes to augment its data sources disclosed?

- Notes: Such steps might include augmenting data sources with synthetic data. We will award this point if the developer reports that it does not take any steps to augment its data.

- References: Gebru et al. (2021), Hutchinson et al. (2021)

7. Upstream → Data → **Harmful data filtration**

- Definition: If data is filtered to remove harmful content, is there a description of the associated filter?

- Notes: Such harmful content might relate to violence or child sexual abuse material. We will award this point if the developer reports that it does not perform any harmful data filtration.

- References: Dodge et al. (2021), Longpre et al. (2023)

8. Upstream → Data → **Copyrighted data**

- Definition: For all data used in building the model, is the associated copyright status disclosed?

- Notes: To receive this point, the copyright status (e.g. copyrighted, public domain) must relate to some decomposition of the data. We will award this point if there is some meaningful decomposition of the data, even if the decomposition is insufficient to receive the Data Creators point or if the disclosure is not comprehensive relative to legal copyright standards.

- References: Bandy & Vincent (2021), Cooper et al. (2023)

9. Upstream → Data → **Data license**

- Definition: For all data used in building the model, is the associated license status disclosed?

- Notes: To receive this point, the license status must relate to some decomposition of the data. We will award this point if there is some meaningful decomposition of the data, even if the decomposition is insufficient to receive the Data Creators point.

- References: Bandy & Vincent (2021), Cooper et al. (2023)

10. Upstream → Data → **Personal information in data**

- Definition: For all data used in building the model, is the inclusion or exclusion of personal information in that data disclosed?

- Notes: To receive this point, the disclosure of personal information must relate to some decomposition of the data. We will award this point if there is some meaningful decomposition of the data, even if the decomposition is insufficient to receive the Data Creators point. Additionally, we will award this point if the developer reports the inclusion of personal information, independent of if and how they mitigate related privacy concerns.

- References: West (2019), Brown et al. (2022)

11. Upstream → Data labor → **Use of human labor**

- Definition: Are the phases of the data pipeline where human labor is involved disclosed?

- Notes: Phases of the data pipeline that involve human labor include activities and tasks performed by people to collect, annotate, clean, or validate data. This indicator is inclusive of all data that is created by or on behalf of the developer. We will award this point if the developer gives a reasonable best-effort description of the use of human labor in their data pipeline.

- References: Kittur et al. (2013), Dzieza (2023)

12. Upstream → Data labor → **Employment of data laborers**

- Definition: Is the organization that directly employs the people involved in data labor disclosed for each phase of the data pipeline?

- Notes: Phases of the data pipeline that involve human labor include activities and tasks performed by people to collect, annotate, clean, or validate data. This indicator is inclusive of all data that is created by or on behalf of the developer. We will award this point if the developer provides the name of the organization that employs data laborers, even if other details about the employment relationship are not disclosed.

- References: Kittur et al. (2013), Dzieza (2023)

13. Upstream → Data labor → **Geographic distribution of data laborers**

- Definition: Is geographic information regarding the people involved in data labor disclosed for each phase of the data pipeline?

- Notes: This indicator is inclusive of all data that is created by or on behalf of the developer. We will award this point if the developer gives a reasonable best-effort description of the geographic distribution of labor at the country-level.

- References: Hao & Seetharaman (2023), Gray & Suri (2019a)

14. Upstream → Data labor → **Wages**

- Definition: Are the wages for people who perform data labor disclosed?

- Notes: This indicator is inclusive of data labor at all points of the model development process, such as training data annotation or red teaming data used to control the model. We will award this point if the developer reports that it does not compensate workers. For all data that is created by or on behalf of the developer,

- References: Kittur et al. (2013), Dzieza (2023)

15. Upstream → Data labor → **Instructions for creating data**

- Definition: Are the instructions given to people who perform data labor disclosed?

- Notes: This indicator is inclusive of all data that is created by or on behalf of the developer. We will award this point if the developer makes a reasonable best-effort attempt to disclose instructions given to people who create data used to build the model for the bulk of the data phases involving human labor.

- References: Sambasivan et al. (2021), Kittur et al. (2013)

16. Upstream → Data labor → **Labor protections**

- Definition: Are the labor protections for people who perform data labor disclosed?

- Notes: This indicator is inclusive of data labor at all points of the model development process, such as training data annotation or red teaming data used to control the model. It is also inclusive of all data that is created by or on behalf of the developer. As an example, labor protections might include protocols to reduce the harm to workers' mental health stemming from exposure to violent content when annotating training data. We will award this point if the developer reports that it does not protect workers or if it does not use data laborers and therefore has no labor protections.

- References: Crawford (2021), Gray & Suri (2019a)

17. Upstream → Data labor → **Third party partners**

   - Definition: Are the third parties who were or are involved in the development of the model disclosed?

   - Notes: This indicator is inclusive of partnerships that go beyond data labor as there may be third party partners at various stages in the model development process. We will award this point if the developer reports that it was the sole entity involved in the development of the model.

   - References: Crawford (2021), Gray & Suri (2019a)

18. Upstream → Data access → **Queryable external data access**

   - Definition: Are external entities provided with queryable access to the data used to build the model?

   - Notes: We will award this point for any reasonable mechanism for providing access: direct access to the data, an interface to query the data, a developer-mediated access program where developers can inspect requests, etc. Developers may receive this point even if there are rate-limits on the number of queries permitted to an external entity and restrictions on which external entities are given access, insofar as these limits and restrictions are transparent and ensure a reasonable amount of external access. We may accept justifications for prohibiting queries of specific parts of the data.

   - References: Gebru et al. (2021), Piktus et al. (2023)

19. Upstream → Data access → **Direct external data access**

   - Definition: Are external entities provided with direct access to the data used to build the model?

   - Notes: We will award this point if external entities can directly access the data without any form of gating from the developer. With that said, we may award this point if the developer provides justifications for prohibiting access to specific parts of the data or to unauthorized external entities.

   - References: Gebru et al. (2021), Piktus et al. (2023)

20. Upstream → Compute → **Compute usage**

   - Definition: Is the compute required for building the model disclosed?

   - Notes: Compute should be reported in appropriate units, which most often will be floating point operations (FLOPS). Compute should be reported to a precision of one significant figure (e.g. 5 x $10^{25}$ FLOPS). We will award this point even if there is no decomposition of the reported compute usage into compute phases, but it should be clear whether the reported compute usage is for a single model run or includes additional runs, or hyperparameter tuning, or training other models like reward models, or other steps in the model development process that necessitate compute expenditure.

   - References: Henderson et al. (2020), Strubell et al. (2019)

21. Upstream → Compute → **Development duration**

- Definition: Is the amount of time required to build the model disclosed?

- Notes: The continuous duration of time required to build the model should be reported in weeks, days, or hours to a precision of one significant figure (e.g. 3 weeks). No form of decomposition into phases of building the model is required for this indicator, but it should be clear what the duration refers to (e.g. training the model, training and subsequent evaluation and red teaming).

- References: Sevilla et al. (2022), Hoffmann et al. (2022b)

22. Upstream → Compute → **Compute hardware**

- Definition: For the primary hardware used to build the model, is the amount and type of hardware disclosed?

- Notes: In most cases, this indicator will be satisfied by information regarding the number and type of GPUs or TPUs used to train the model. The number of hardware units should be reported to a precision of one significant figure (e.g. 800 NVIDIA H100 GPUs). We will not award this point if (i) the training hardware generally used by the developer is disclosed, but the specific hardware for the given model is not, or (ii) the training hardware is disclosed, but the amount of hardware is not. We will award this point even if information about the interconnects between hardware units is not disclosed.

- References: Sevilla et al. (2022), Hoffmann et al. (2022b)

23. Upstream → Compute → **Hardware owner**

- Definition: For the primary hardware used in building the model, is the owner of the hardware disclosed?

- Notes: For example, the hardware owner may be the model developer in the case of a self-owned cluster, a cloud provider like Microsoft Azure, Google Cloud Platform, or Amazon Web Services, or a national supercomputer. In the event that hardware is owned by multiple sources or is highly decentralized, we will award this point if a developer makes a reasonable effort to describe the distribution of hardware owners.

- References: Sevilla et al. (2022), Hoffmann et al. (2022b)

24. Upstream → Compute → **Energy usage**

- Definition: Is the amount of energy expended in building the model disclosed?

- Notes: Energy usage should be reported in appropriate units, which most often will be megawatt-hours (mWh). Energy usage should be reported to a precision of one significant figure (e.g. 500 mWh). No form of decomposition into compute phases is required, but it should be clear whether the reported energy usage is for a single model run or includes additional runs, or hyperparameter tuning, or training other models like reward models, or other steps in the model development process that necessitate energy usage.

- References: Lacoste et al. (2019), Patterson et al. (2021)

25. Upstream → Compute → **Carbon emissions**

- Definition: Is the amount of carbon emitted (associated with the energy used) in building the model disclosed?

- Notes: Emissions should be reported in appropriate units, which most often will be tons of carbon dioxide emitted (tCO2). Emissions should be reported to a precision of one significant figure (e.g. 500 tCO2). No form of decomposition into compute phases is required, but it should be clear whether the reported emissions is for a single model run or includes additional runs, or hyperparameter tuning, or training other models like reward models, or other steps in the model development process that generate emissions.

- References: Lacoste et al. (2019), Patterson et al. (2021)

26. Upstream → Compute → **Broader environmental impact**

- Definition: Are any broader environmental impacts from building the model besides carbon emissions disclosed?

- Notes: While the most direct environmental impact of building a foundation model is the energy used and, therefore, the potential carbon emissions, there may be other environmental impacts. For example, these may include the use of other resources such as water for cooling data centers or metals for producing specialized hardware. We recognize that there does not exist an authoritative or consensus list of broader environmental factors. For this reason, we will award this point if there is a meaningful, though potentially incomplete, discussion of broader environmental impact.

- References: Luccioni & Hernández-García (2023), Strubell et al. (2019)

27. Upstream → Methods → **Model stages**

- Definition: Are all stages in the model development process disclosed?

- Notes: Stages refer to each identifiable step that constitutes a substantive change to the model during the model building process. We recognize that different developers may use different terminology for these stages, or conceptualize the stages differently. We will award this point if there is a clear and complete description of these stages.

- References: Mitchell et al. (2019), Chung et al. (2022)

28. Upstream → Methods → **Model objectives**

- Definition: For all stages that are described, is there a clear description of the associated learning objectives or a clear characterization of the nature of this update to the model?

- Notes: We recognize that different developers may use different terminology for these stages, or conceptualize the stages differently. We will award this point if there is a clear description of the update to the model related to each stage, whether that is the intent of the stage (e.g. making the model less harmful), a mechanistic characterization (e.g. minimizing a specific loss function), or an empirical assessment (e.g. evaluation results conducted before and after the stage).

- References: Mitchell et al. (2019), Chung et al. (2022)

29. Upstream → Methods → **Core frameworks**

   - Definition: Are the core frameworks used for model development disclosed?

   - Notes: Examples of core frameworks include Tensorflow, PyTorch, Jax, Hugging Face Transformers, Seqio, T5X, Keras, SciKit, and Triton. If there are significant internal frameworks, there should be some description of their function and/or a reasonably similar publicly-available analogue. We recognize that there does not exist an authoritative or consensus list of core frameworks. For this reason, we will award this point if there is a meaningful, though potentially incomplete, list of major frameworks for the first version of the index.

   - References: Mitchell et al. (2019), Chung et al. (2022)

30. Upstream → Methods → **Additional dependencies**

   - Definition: Are any dependencies required to build the model disclosed besides data, compute, and code?

   - Notes: For example, if the model depends on an external search engine, programmable APIs, or tools, this should be disclosed. We recognize that there is not widespread consensus regarding what constitutes key dependencies beyond the data, compute, and code. We will award this point only if developers give a reasonable best-effort description of any additional dependencies or make clear that no additional dependencies are required.

   - References: Lukas et al. (2023), Kim et al. (2023)

31. Upstream → Data Mitigations → **Mitigations for privacy**

   - Definition: Are any steps the developer takes to mitigate the presence of PII in the data disclosed?

   - Notes: Such steps might include identifying personal information in the training data, filtering specific datasets to remove personal information, and reducing the likelihood that models will output personal information. We will award this point if the developer reports that it does not take steps to mitigate the presence of PII in the data.

   - References: Kandpal et al. (2022), Cooper et al. (2023)

32. Upstream → Data Mitigations → **Mitigations for copyright**

   - Definition: Are any steps the developer takes to mitigate the presence of copyrighted information in the data disclosed?

   - Notes: Such steps might include identifying copyrighted data, filtering specific datasets to remove copyrighted data, and reducing the likelihood that models will output copyrighted information. We will award this point if the developer reports that it does take steps to mitigate the presence of copyrighted information in the data.

   - References: Bandy & Vincent (2021), Cooper et al. (2023)

33. Model → Model basics → **Input modality**

- Definition: Are the input modalities for the model disclosed?

- Notes: Input modalities refer to the types or formats of information that the model can accept as input. Examples of input modalities include text, image, audio, video, tables, graphs.

- References: Mitchell et al. (2019), Crisan et al. (2022)

34. Model → Model basics → **Output modality**

- Definition: Are the output modalities for the model disclosed?

- Notes: Output modalities refer to the types or formats of information that the model can accept as output. Examples of output modalities include text, image, audio, video, tables, graphs.

- References: Mitchell et al. (2019), Crisan et al. (2022)

35. Model → Model basics → **Model components**

- Definition: Are all components of the model disclosed?

- Notes: Model components refer to distinct and identifiable parts of the model. We recognize that different developers may use different terminology for model components, or conceptualize components differently. Examples include: (i) For a text-to-image model, components could refer to a text encoder and an image encoder, which may have been trained separately. (ii) For a retrieval-augmented model, components could refer to a separate retriever module.

- References: Mitchell et al. (2019), Crisan et al. (2022)

36. Model → Model basics → **Model size**

- Definition: For all components of the model, is the associated model size disclosed?

- Notes: This information should be reported in appropriate units, which generally is the number of model parameters, broken down by named component. Model size should be reported to a precision of one significant figure (e.g. 500 billion parameters for text encoder, 20 billion parameters for image encoder).

- References: Mitchell et al. (2019), Crisan et al. (2022)

37. Model → Model basics → **Model architecture**

- Definition: Is the model architecture disclosed?

- Notes: Model architecture is the overall structure and organization of a foundation model, which includes the way in which any disclosed components are integrated and how data moves through the model during training or inference. We recognize that different developers may use different terminology for model architecture, or conceptualize the architecture differently. We will award this point for any clear, though potentially incomplete, description of the model architecture.

- References: Mitchell et al. (2019), Crisan et al. (2022)

38. Model → Model basics → **Centralized model documentation**

- Definition: Is key information about the model included in a centralized artifact such as a model card?

- Notes: We recognize that different developers may share this information through different types of documentation, such as a system card or several clearly interrelated documents. We will award this point for the disclosure of any such centralized artifact that provides key information typically included in a model card, though the artifact may be longer-form than a standard model card (e.g. a technical report).

- References: Mitchell et al. (2019), Crisan et al. (2022)

39. Model → Model access → **External model access protocol**

- Definition: Is a protocol for granting external entities access to the model disclosed?

- Notes: A model access protocol refers to the steps, requirements, and considerations involved in granting authorized model access to external entities. We will award this point if the developer discloses key details of its protocol, including (i) where external entities can request access (e.g. via an access request form); (ii) explicit criteria for selecting external entities; and (iii) a transparent decision on whether access has been granted within a specified, reasonable period of time.

- References: Solaiman (2023), Shevlane (2022)

40. Model → Model access → **Blackbox external model access**

- Definition: Is black box model access provided to external entities?

- Notes: Black box model access refers to the ability to query the model with inputs and receive outputs, potentially without further access. Examples of external entities that might be granted access include researchers, third-party auditors, and regulators. We will award this point for any reasonable access level: direct access to the model weights, an interface to query the model, a developer-mediated access program where developers can inspect requests, etc. Developers may receive this point even if there are rate-limits on the number of queries permitted to an external entity and restrictions on the external entities that are permitted access, insofar as these limits and restrictions are transparent.

- References: Solaiman (2023), Shevlane (2022)

41. Model → Model access → **Full external model access**

   - Definition: Is full model access provided to external entities?

   - Notes: Full model access refers to the ability to access the model via the release of model weights. Developers may receive this point even if there are some restrictions on the external entities that are permitted access (e.g. geographic restrictions), insofar as these restrictions are transparent (e.g. via some high-level description of who has been granted access to the foundation model).

   - References: Solaiman (2023), Shevlane (2022)

42. Model → Capabilities → **Capabilities description**

   - Definition: Are the model's capabilities described?

   - Notes: Capabilities refer to the specific and distinctive functions that the model can perform. We recognize that different developers may use different terminology for capabilities, or conceptualize capabilities differently. We will award this point for any clear, but potentially incomplete, description of the multiple capabilities.

   - References: Srivastava et al. (2022), Liang et al. (2022c)

43. Model → Capabilities → **Capabilities demonstration**

   - Definition: Are the model's capabilities demonstrated?

   - Notes: Demonstrations refer to illustrative examples or other forms of showing the model's capabilities that are legible or understandable for the general public, without requiring specific technical expertise. We recognize that different developers may use different terminology for capabilities, or conceptualize capabilities differently. We will award this point for clear demonstrations of multiple capabilities.

   - References: Srivastava et al. (2022), Liang et al. (2022c)

44. Model → Capabilities → **Evaluation of capabilities**

   - Definition: Are the model's capabilities rigorously evaluated, with the results of these evaluations reported prior to or concurrent with the initial release of the model?

   - Notes: Rigorous evaluations refer to precise quantifications of the model's behavior in relation to its capabilities. We recognize that capabilities may not perfectly align with evaluations, and that different developers may associate capabilities with evaluations differently. We will award this point for clear evaluations of multiple capabilities. For example, this may include evaluations of world knowledge, reasoning, state tracking or other such proficiencies. Or it may include the measurement of average performance (e.g. accuracy, F1) on benchmarks for specific tasks (e.g. text summarization, image captioning). We note that evaluations on standard broad-coverage benchmarks are likely to suffice for this indicator, though they may not if the model's capabilities are presented as especially unusual such that standard evaluations will not suffice.

   - References: Srivastava et al. (2022), Liang et al. (2022c)

45. Model → Capabilities → **External reproducibility of capabilities evaluation**

- Definition: Are the evaluations of the model's capabilities reproducible by external entities?

- Notes: For an evaluation to be reproducible by an external entity, we mean that the associated data is either (i) publicly available or (ii) described sufficiently such that a reasonable facsimile can be constructed by an external entity. In addition, the evaluation protocol should be sufficiently described such that if the evaluation is reproduced, any discrepancies with the developer's results can be resolved. We recognize that there does not exist an authoritative or consensus standard for what is required for an evaluation to be deemed externally reproducible. Evaluations on standard benchmarks are assumed to be sufficiently reproducible for the purposes of this index. We will award this point for reproducibility of multiple disclosed evaluations. In the event that an evaluation is not reproducible, a justification by the model developer for why it is not possible for the evaluation to be made reproducible may be sufficient to score this point.

- References: Kapoor & Narayanan (2023), Liang et al. (2022c)

46. Model → Capabilities → **Third party capabilities evaluation**

- Definition: Are the model's capabilities evaluated by third parties?

- Notes: By third party, we mean entities that are significantly or fully independent of the developer. We will award this point if (i) a third party has conducted an evaluation of model capabilities, (ii) the results of this evaluation are publicly available, and (iii) these results are disclosed or referred to in the developer's materials.

- References: Raji et al. (2022b), Liang et al. (2022c)

47. Model → Limitations → **Limitations description**

- Definition: Are the model's limitations disclosed?

- Notes: Limitations refer to the specific and distinctive functions that the model cannot perform (e.g. the model cannot answer questions about current events as it only contains data up to a certain time cutoff, the model is not very capable when it comes to a specific application). We recognize that different developers may use different terminology for limitations, or conceptualize limitations differently. We will award this point for any clear, but potentially incomplete, description of multiple limitations.

- References: Raji et al. (2022a), Liang et al. (2022c)

48. Model → Limitations → **Limitations demonstration**

- Definition: Are the model's limitations demonstrated?

- Notes: Demonstrations refer to illustrative examples or other forms of showing the limitations that are legible or understandable for the general public, without requiring specific technical expertise. We recognize that different developers may use different terminology for limitations, or conceptualize the limitations differently. We will award this point for clear demonstrations of multiple limitations.

- References: Raji et al. (2022a), Liang et al. (2022c)

49. Model → Limitations → **Third party evaluation of limitations**

   - Definition: Can the model's limitations be evaluated by third parties?

   - Notes: By third parties, we mean entities that are significantly or fully independent of the model developers. In contrast to the third party evaluation indicators for capabilities and risks, we will award this point if third party evaluations are possible even if no third party has yet conducted them. Such evaluations are possible if, for example, the model is deployed via an API (or with open weights) and there are no restrictions on evaluating limitations (e.g. in the usage policy).

   - References: Raji et al. (2022b), Liang et al. (2022c)

50. Model → Risks → **Risks description**

   - Definition: Are the model's risks disclosed?

   - Notes: Risks refer to possible negative consequences or undesirable outcomes that can arise from the model's deployment and usage. This indicator requires disclosure of risks that may arise in the event of both (i) intentional (though possibly careless) use, such as bias or hallucinations and (ii) malicious use, such as fraud or disinformation. We recognize that different developers may use different terminology for risks, or conceptualize risks differently. We will award this point for any clear, but potentially incomplete, description of multiple risks.

   - References: Solaiman et al. (2023), Weidinger et al. (2021)

51. Model → Risks → **Risks demonstration**

   - Definition: Are the model's risks demonstrated?

   - Notes: Demonstrations refer to illustrative examples or other forms of showing the risks that are legible or understandable for the general public, without requiring specific technical expertise. This indicator requires demonstration of risks that may arise in the event of both (i) intentional (though possibly careless) use, such as biases or hallucinations and (ii) malicious use, such as fraud or disinformation. We recognize that different developers may use different terminology for risks, or conceptualize risks differently. We will award this point for clear demonstrations of multiple risks.

   - References: Solaiman et al. (2023), Weidinger et al. (2021)

52. Model → Risks → **Unintentional harm evaluation**

   - Definition: Are the model's risks related to unintentional harm rigorously evaluated, with the results of these evaluations reported prior to or concurrent with the initial release of the model?

   - Notes: Rigorous evaluations refer to precise quantifications of the model's behavior in relation to such risks. Unintentional harms include bias, toxicity, and issues relating to fairness. We recognize that unintended harms may not perfectly align with risk evaluations, and that different developers may associate risks with evaluations differently. We will award this point for clear evaluations of multiple such risks. We note that evaluations on standard broad-coverage benchmarks are likely to suffice for this indicator, though they may not if the model's risks related to unintentional harm are presented as especially unusual or severe.

   - References: Solaiman et al. (2023), Weidinger et al. (2021)

53. Model → Risks → **External reproducibility of unintentional harm evaluation**

- Definition: Are the evaluations of the model's risks related to unintentional harm reproducible by external entities?

- Notes: For an evaluation to be reproducible by an external entity, we mean that the associated data is either (i) publicly available or (ii) described sufficiently such that a reasonable facsimile can be constructed by the external entity. In addition, the evaluation protocol should be sufficiently described such that if the evaluation is reproduced, any discrepancies with the developer's results can be resolved. We recognize that there does not exist an authoritative or consensus standard for what is required for an evaluation to be deemed externally reproducible. Evaluations on standard benchmarks are assumed to be sufficiently reproducible for the purposes of this index. We will award this point for reproducibility of multiple disclosed evaluations. In the event that an evaluation is not reproducible, a justification by the developer for why it is not possible for the evaluation to be made reproducible may suffice.

- References: Kapoor & Narayanan (2023), Weidinger et al. (2021)

54. Model → Risks → **Intentional harm evaluation**

- Definition: Are the model's risks related to intentional harm rigorously evaluated, with the results of these evaluations reported prior to or concurrent with the initial release of the model?.

- Notes: Rigorous evaluations refer to precise quantifications of the model's behavior in relation to such risks. Intentional harms include fraud, disinformation, scams, cybersecurity attacks, designing weapons or pathogens, and uses of the model for illegal purposes. We recognize that unintentional harms may not perfectly align with risk evaluations, and that different developers may associate risks with evaluations differently. We will award this point for clear evaluations of multiple such risks. We note that evaluations on standard broad-coverage benchmarks are likely to suffice for this indicator, though they may not if the model's risks related to unintentional harm are presented as especially unusual or severe.

- References: Solaiman et al. (2023), Weidinger et al. (2021)

55. Model → Risks → **External reproducibility of intentional harm evaluation**

- Definition: Are the evaluations of the model's risks related to intentional harm reproducible by external entities?

- Notes: For an evaluation to be reproducible by an external entity, we mean that the associated data is either (i) publicly available or (ii) described sufficiently such that a reasonable facsimile can be constructed by the external entity. In addition, the evaluation protocol should be sufficiently described such that if the evaluation is reproduced, any discrepancies with the developer's results can be resolved. We recognize that there does not exist an authoritative or consensus standard for what is required for an evaluation to be deemed externally reproducible. Evaluations on standard benchmarks are assumed to be sufficiently reproducible for the purposes of this index. We will award this point for reproducibility of multiple disclosed evaluations. In the event that an evaluation is not reproducible, a justification by the model developer for why it is not possible for the evaluation to be made reproducible may suffice.

- References: Kapoor & Narayanan (2023), Weidinger et al. (2021)

56. Model → Risks → **Third party risks evaluation**

- Definition: Are the model's risks evaluated by third parties?

- Notes: By third party, we mean entities that are significantly or fully independent of the developer. A third party risk evaluation might involve the developer allowing a third party to choose a methodology for evaluating risks that differs from that of the developer. We will award this point if (i) a third party has conducted an evaluation of model risks, (ii) the results of this evaluation are publicly available, and (iii) these results are disclosed or referred to in the developer's materials. If the results are not made public (but are disclosed to have been conducted) and/or the results are not discoverable in the developer's materials, we will not award this point. We may accept a justification from either the third party or the developer for why part of the evaluation is not disclosed in relation to risks.

- References: Raji et al. (2022b), Weidinger et al. (2021)

57. Model → Model Mitigations → **Mitigations description**

- Definition: Are the model mitigations disclosed?

- Notes: By model mitigations, we refer to interventions implemented by the developer at the level of the model to reduce the likelihood and/or the severity of the model's risks. We recognize that different developers may use different terminology for mitigations, or conceptualize mitigations differently. We will award this point for any clear, but potentially incomplete, description of multiple mitigations associated with the model's risks. Alternatively, we will award this point if the developer reports that it does not mitigate risk.

- References: Solaiman et al. (2023), Weidinger et al. (2021)

58. Model → Model Mitigations → **Mitigations demonstration**

- Definition: Are the model mitigations demonstrated?

- Notes: Demonstrations refer to illustrative examples or other forms of showing the mitigations that are legible or understandable for the general public, without requiring specific technical expertise. We recognize that different developers may use different terminology for mitigations, or conceptualize mitigations differently. We will award this point for clear demonstrations of multiple mitigations. We will also award this point if the developer reports that it does not mitigate the risks associated with the model.

- References: Solaiman et al. (2023), Weidinger et al. (2021)

59. Model → Model Mitigations → **Mitigations evaluation**

- Definition: Are the model mitigations rigorously evaluated, with the results of these evaluations reported?

- Notes: Rigorous evaluations refer to precise quantifications of the model's behavior in relation to the mitigations associated with its risks. We will award this point for clear evaluations of multiple mitigations.

- References: Huang et al. (2023), Weidinger et al. (2021)

60. Model → Model Mitigations → **External reproducibility of mitigations evaluation**

    - Definition: Are the model mitigation evaluations reproducible by external entities?

    - Notes: For an evaluation to be reproducible by an external entity, we mean that the associated data is either (i) publicly available or (ii) described sufficiently such that a reasonable facsimile can be constructed by the external entity. In addition, the evaluation protocol should be sufficiently described such that if the evaluation is reproduced, any discrepancies with the developer's results can be resolved. In the case of mitigations evaluations, this will usually involve details about a comparison to some baseline, which may be a different, unmitigated version of the model. We recognize that there does not exist an authoritative or consensus standard for what is required for an evaluation to be deemed externally reproducible. We will award this point for reproducibility of multiple disclosed evaluations. In the event that an evaluation is not reproducible, a justification by the model developer for why it is not possible for the evaluation to be made reproducible may suffice.

    - References: Kapoor & Narayanan (2023), Weidinger et al. (2021)

61. Model → Model Mitigations → **Third party mitigations evaluation**

    - Definition: Can the model mitigations be evaluated by third parties?

    - Notes: By third party, we mean entities that are significantly or fully independent of the model developers. This indicator assesses whether it is possible for third parties to assess mitigations, which is not restricted to the methods the developer uses to assess mitigations. In contrast to the third party evaluation indicators for capabilities and risks, we will award this point if third party evaluations are possible even if no third party has yet conducted them.

    - References: Raji et al. (2022b), Weidinger et al. (2021)

62. Model → Trustworthiness → **Trustworthiness evaluation**

    - Definition: Is the trustworthiness of the model rigorously evaluated, with the results of these evaluations disclosed?

    - Notes: Rigorous evaluations refer to precise quantifications of the model's behavior in relation to its trustworthiness. For example, this may include evaluations of the model's robustness or reliability, its uncertainty, calibration, or causality, or its interpretability or explainability. We recognize that trustworthiness may not perfectly align with evaluations, and that different developers may associate trustworthiness with evaluations differently. We will award this point for a clear evaluation of the trustworthiness of the model.

    - References: Brundage et al. (2020), Wang et al. (2023a)

63. Model → Trustworthiness → **External reproducibility of trustworthiness evaluation**

- Definition: Are the trustworthiness evaluations reproducible by external entities?

- Notes: For an evaluation to be reproducible by an external entity, we mean that the associated data is either (i) publicly available or (ii) described sufficiently such that a reasonable facsimile can be constructed by the external entity. In addition, the evaluation protocol should be sufficiently described such that if the evaluation is reproduced, any discrepancies with the developer's results can be resolved. We recognize that there does not exist an authoritative or consensus standard for what is required for an evaluation to be deemed externally reproducible. Evaluations on standard benchmarks are assumed to be sufficiently reproducible for the purposes of this index. We will award this point for reproducibility of at least one evaluation. In the event that an evaluation is not reproducible, we may accept a justification by the model developer for why it is not possible for the evaluation to be made reproducible.

- References: Kapoor & Narayanan (2023), Shneiderman (2020)

64. Model → Inference → **Inference duration evaluation**

- Definition: Is the time required for model inference disclosed for a clearly-specified task on a clearly-specified set of hardware?

- Notes: The duration should be reported in seconds to a precision of one significant figure (e.g. 0.002 seconds). We recognize that no established standard exists for the standardized reporting of inference evaluation. Therefore, we permit the developer to specify the task and hardware setup, as long as both are disclosed. The hardware in this evaluation need not be the hardware the developer uses for inference if it in fact does any inference itself. For example, the specific task might be generating 100,000 tokens as 5,000 sequences of length 20 and the fixed set of hardware might be 8 NVIDIA A100s. The hardware in this evaluation need not be the hardware the developer uses for inference if it in fact does any inference itself.

- References: Reddi et al. (2020), Narayanan et al. (2023)

65. Model → Inference → **Inference compute evaluation**

- Definition: Is the compute usage for model inference disclosed for a clearly-specified task on a clearly-specified set of hardware?

- Notes: Compute usage for inference should be reported in FLOPS to a precision of one significant figure (e.g. $5 \times 10^{25}$ FLOPS). We recognize that no established standard exists for the standardized reporting of inference evaluation. Therefore, we permit the developer to specify the task and hardware setup, as long as both are clear. For example, the specific task might be generating 100k tokens as 5k sequences of length 20 and the fixed set of hardware might be 8 NVIDIA A100s. The hardware in this evaluation need not be the hardware the developer uses for inference if it in fact does any inference itself.

- References: Reddi et al. (2020), Narayanan et al. (2023)

66. Downstream → Distribution → **Release decision-making**

   - Definition: Is the developer's protocol for deciding whether or not to release a model disclosed?

   - Notes: We recognize that the release of a foundation model falls along a spectrum, with many forms of partial release, and that different developers may conceptualize release differently. We will award this point for any clear protocol that discusses the decision-making process, including if the protocol is more general to the developer rather than the specific foundation model under consideration.

   - References: Solaiman (2023), Liang et al. (2022a)

67. Downstream → Distribution → **Release process**

   - Definition: Is a description of the process of how the model was released disclosed?

   - Notes: A description of the release process might include information about who received access to the model at what stage of the release of the model. For example, a developer might conduct a staged release where it releases the model to a select group at first and subsequently makes the model more widely available. We recognize that the release of a foundation model falls along a spectrum, with many different forms of release, and that different developers may conceptualize release differently. We will award this point for any detailed discussion of the release process, including if the discussion is more general to the developer rather than the specific foundation model under consideration.

   - References: Solaiman (2023), Liang et al. (2022a)

68. Downstream → Distribution → **Distribution channels**

   - Definition: Are all distribution channels disclosed?

   - Notes: By distribution channel, we mean any pathway by which the model is made accessible to entities beyond the developer. We recognize that distribution channels may arise without the knowledge of the model developer. For example, the weights of a model may be released through one distribution channel and then be distributed through other channels. We will award this point if the developer discloses all of the distribution channels of which it is aware.

   - References: Cobbe et al. (2023), Widder & Wong (2023)

69. Downstream → Distribution → **Products and services**

   - Definition: Does the developer disclose whether any products and services offered by the developer are dependent on the model?

   - Notes: We recognize that a developer may provide many products and services that depend on a foundation model or internal derivatives of the model. We will award this point for a reasonable best-effort description of any ways the developer makes internal use of the model in its products or services.

   - References: Cobbe et al. (2023), Cen et al. (2023)

70. Downstream → Distribution → **Detection of machine-generated content**

   - Definition: Are any mechanisms for detecting content generated by this model disclosed?

   - Notes: Such a mechanism might include storing a copy of all outputs generated by the model to compare against, implementing a watermark when generating content using the model, or training a detector post-hoc to identify such content. We will award this point if any such mechanism is disclosed or if the developer reports that it has no such mechanism.

   - References: Kirchenbauer et al. (2023), Kuditipudi et al. (2023)

71. Downstream → Distribution → **Model License**

   - Definition: Is a license for the model disclosed?

   - Notes: In the event that licenses are written more generally, it should be clear which assets they apply to. We recognize that different developers may adopt different business models and therefor have different types of model licenses. Examples of model licenses include responsible AI licenses, open-source licenses, and licenses that allow for commercial use.

   - References: Pistilli et al. (2023), Chen et al. (2023a)

72. Downstream → Distribution → **Terms of service**

   - Definition: Are terms of service disclosed for each distribution channel?

   - Notes: We will award this point if there are terms-of-service that appear to apply to the bulk of the model's distribution channels.

   - References: Rakova et al. (2022), Liu et al. (2021)

73. Downstream → Usage policy → **Permitted and prohibited users**

    - Definition: Is a description of who can and cannot use the model disclosed?

    - Notes: Such restrictions may relate to countries (e.g. US-only), organizations (e.g. no competitors), industries (e.g. no weapons industry users) or other relevant factors. These restrictions on users are often contained in multiple policies; we group them here for simplicity. We will awarded this point for a clear description of permitted, restricted, and prohibited users of the model.

    - References: Cohere (2022), Meta (2023)

74. Downstream → Usage policy → **Permitted, restricted, and prohibited uses**

    - Definition: Are permitted, restricted, and prohibited uses of the model disclosed?

    - Notes: We will award this point if at least two of the following three categories are disclosed: (i) permitted uses, (ii) restricted uses, and (iii) prohibited uses. By restricted uses, we mean uses that require a higher level of scrutiny (such as permission from or a separate contract with the developer) to be permitted. These uses are generally included in an acceptable use policy, model license, or usage policy.

    - References: Cohere (2022), Meta (2023)

75. Downstream → Usage policy → **Usage policy enforcement**

    - Definition: Is the enforcement protocol for the usage policy disclosed?

    - Notes: By enforcement protocol, we refer to (i) mechanisms for identifying permitted and prohibited users, (ii) mechanisms for identifying permitted/restricted/prohibited uses, (iii) steps the developer takes to enforce its policies related to such uses, and (iv) the developer's procedures for carrying out these steps. We will award this point for a reasonable best-effort attempt to provide the bulk of this information, though one line indicating the developer reserves the right to terminate accounts is insufficient. Alternatively, we will award this point if the developer reports that it does not enforce its usage policy.

    - References: Cohere (2022), Meta (2023)

76. Downstream → Usage policy → **Justification for enforcement action**

    - Definition: Do users receive a justification when they are subject to an enforcement action for violating the usage policy?

    - Notes: For example, does the developer disclose a protocol for telling users which part of the usage policy they violated, when they did so, and what specifically was violative? Enforcement actions refer to measures to limit a user's ability to use the model, such as banning a user or restricting their ability to purchase tokens. We will award this point if the developer discloses that it gives justification for enforcement actions or, alternatively, if it discloses that it does not provide justification for enforcement actions or that it does not enforce its usage policy.

    - References: Cohere (2022), Meta (2023)

77. Downstream → Usage policy → **Usage policy violation appeals mechanism**

- Definition: Is a mechanism for appealing potential usage policy violations disclosed?

- Notes: We will award this point if the developer provides a usage policy violation appeals mechanism, regardless of whether it is provided via a user interface or distribution channel.

- References: Cohere (2022), Meta (2023)

78. Downstream → Model behavior policy → **Permitted, restricted, and prohibited model behaviors**

- Definition: Are model behaviors that are permitted, restricted, and prohibited disclosed?

- Notes: We refer to a policy that includes this information as a model behavior policy, or a developer's policy on what the foundation model can and cannot do (e.g. such a policy may prohibit a model from generating child sexual abuse material). We recognize that different developers may adopt different business models and that some business models may make enforcement of a model behavior policy more or less feasible. We will award this point if at least two of the three categories (i.e. permitted, restricted, and prohibited model behaviors) are disclosed. Alternatively, we will award this point if the developer reports that it does not impose any restrictions on its model's behavior.

- References: Reuter & Schulze (2023), Qi et al. (2023)

79. Downstream → Model behavior policy → **Model behavior policy enforcement**

- Definition: Is the enforcement protocol for the model behavior policy disclosed?

- Notes: By enforcement protocol, we refer to mechanisms for identifying whether model behavior is permitted or prohibited and actions that may arise in the event the model behavior policy is violated. For example, the developer may make updates to the model in response to issues with the model's adherence to the model behavior policy. We will award this point if there is a clear description of the enforcement protocol, or if the developer reports that it does not enforce its model behavior policy or that it has no such restrictions on the model's behavior.

- References: Brundage et al. (2020), Qi et al. (2023)

80. Downstream → Model behavior policy → **Interoperability of usage and model behavior policies**

- Definition: Is the way that the usage policy and the model behavior policy interoperate disclosed?

- Notes: For example, if a user attempts to use the model for a prohibited use such as spam, how does the model behavior policy apply if at all? We will also award this point if the developer reports that it does not impose any restrictions on its model's behavior in the event of usage policy violation.

- References: Reuter & Schulze (2023), Qi et al. (2023)

81. Downstream → User Interface → **User interaction with AI system**

   - Definition: For distribution channels with user-facing interfaces, are users notified (i) that they are interacting with an AI system, (ii) of the specific foundation model they are interacting with, and (iii) that outputs are machine-generated?

   - Notes: A user-facing interface refers to the means by which the user interacts with the foundation model, including how the user can observe outputs from the foundation model and other notifications. We will award this point if, for all distribution channels with user-facing interfaces, the user is provided adequate transparency as to the foundation model being distributed and the potential presence of any model outputs.

   - References: Wang et al. (2023b), Nakao et al. (2022)

82. Downstream → User Interface → **Usage disclaimers**

   - Definition: For distribution channels with user-facing interfaces, are users provided with disclaimers involving model use?

   - Notes: A user-facing interface refers to the means by which the user interacts with the foundation model, including how the user can observe outputs from the foundation model and other notifications. Usage disclaimers could include information about what constitutes a usage policy violations or how users should interpret model outputs. We will award this point if, for all distribution channels with user-facing interfaces, the user is provided with usage disclaimers.

   - References: Wang et al. (2023b), Nakao et al. (2022)

83. Downstream → User data protection → **User data protection policy**

   - Definition: Are the protocols for how the developer stores, accesses, and shares user data disclosed?

   - Notes: We will also award this point if the developer reports that it has no user data protection policy.

   - References: Nissenbaum (2024), King (2020)

84. Downstream → User data protection → **Permitted and prohibited use of user data**

   - Definition: Are permitted and prohibited uses of user data disclosed?

   - Notes: Developers use user data for a range of purposes such as building future models, updating existing models, and evaluating both existing and future models. We will award this point if a developer discloses its policy on the use of user data from interactions associated with this model, including both permitted and prohibited uses. This may span different distribution channels if multiple channels supply user data to the developer. Alternatively, we will award this point if the developer reports it does not impose any limits on its use of user data.

   - References: Nissenbaum (2024), King (2020)

85. Downstream → User data protection → **Usage data access protocol**

    - Definition: Is a protocol for granting external entities access to usage data disclosed?

    - Notes: Usage data refers to the data created through user interaction with the model, such as user inputs to the model and associated metadata such as the duration of the interaction. A usage data access protocol refers to the steps, requirements, and considerations involved in granting external entities access to usage data; this goes beyond stating the conditions under which related personal information may be shared with external entities. We will award this point for a clear description of the usage data access protocol or if the developer reports it does not share usage data with external entities.

    - References: Lapowsky (2018), King (2020)

86. Downstream → Model Updates → **Versioning protocol**

    - Definition: Is there a disclosed version and versioning protocol for the model?

    - Notes: By versioning, we mean that each instance of the model is uniquely identified and that the model is guaranteed to not change when referring to a fixed version number; alternatively, the version clearly indicating a specific instance of the model may be able to change by noting that it is the "latest" or an "unstable" version. We recognize that different developers may adopt different versioning practices that may differ from standard semantic versioning practices used elsewhere in software engineering.

    - References: Chen et al. (2023b), Lam et al. (2020)

87. Downstream → Model Updates → **Change log**

    - Definition: Is there a disclosed change log for the model?

    - Notes: By change log, we mean a description associated with each change to the model (which should be indicated by a change in version number). We recognize that different developers may adopt different practices for change logs that may differ from practices used elsewhere in software engineering. We will award this point if the change log provides a clear description of changes that is legible to a technical audience.

    - References: Chen et al. (2023b), Li et al. (2016)

88. Downstream → Model Updates → **Deprecation policy**

    - Definition: Is there a disclosed deprecation policy for the developer?

    - Notes: By deprecation policy, we refer to a description of what it means for a model to be deprecated and how users should respond to the deprecation (e.g. instructions to migrate to a newer version). We will award this point for a clear disclosure of a deprecation policy or if there is no risk of deprication (e.g. if the developer openly releases model weights).

    - References: Chen et al. (2023b), Haryono et al. (2020)

89. Downstream → Feedback → **Feedback mechanism**

   - Definition: Is a feedback mechanism disclosed?

   - Notes: By feedback mechanism, we refer to a means for external entities to report feedback or issues that arise in relation to the foundation model. Such entities may include but are not necessarily limited to users. We will award this point if the developer discloses a feedback mechanism that has been implemented.

   - References: Bommasani et al. (2023b), Raji et al. (2022b)

90. Downstream → Feedback → **Feedback summary**

   - Definition: Is a report or summary disclosed regarding the feedback the developer received or, alternatively, the way the developer responded to that feedback?

   - Notes: We recognize that there does not exist an authoritative or consensus standard for what is required in a feedback report. For this reason, we will award this point if there is a meaningful, though potentially vague or incomplete, summary of feedback received.

   - References: Chen et al. (2021), Piorkowski et al. (2022)

91. Downstream → Feedback → **Government inquiries**

   - Definition: Is a summary of government inquiries related to the model received by the developer disclosed?

   - Notes: Such government inquiries might include requests for user data, requests that certain content be banned, or requests for information about a developer's business practices. We recognize that there does not exist an authoritative or consensus standard for what is required for such a summary of government inquiries. For this reason, we will award this point if (i) there is a meaningful, though potentially vague or incomplete, summary of government inquiries, or (ii) a summary of government inquiries related to user data.

   - References: Chou, Bommasani et al. (2023b)

92. Downstream → Impact → **Monitoring mechanism**

   - Definition: For each distribution channel, is a monitoring mechanism for tracking model use disclosed?

   - Notes: By monitoring mechanism, we refer to a specific protocol for tracking model use that goes beyond an acknowledgement that usage data is collected. We will also award this point for a reasonable best-effort attempt to describe monitoring mechanisms, or if a developer discloses that a distribution channel is not monitored.

   - References: Springer & Whittaker (2018), Bommasani et al. (2023b)

93. Downstream → Impact → **Downstream applications**

- Definition: Across all forms of downstream use, is the number of applications dependent on the foundation model disclosed?

- Notes: We recognize that there does not exist an authoritative or consensus standard for what qualifies as an application. We will award this point if there is a meaningful estimate of the number of downstream applications, along with some description of what it means for an application to be dependent on the model.

- References: Vipra & Korinek (2023), Bommasani et al. (2023b)

94. Downstream → Impact → **Affected market sectors**

- Definition: Across all downstream applications, is the fraction of applications corresponding to each market sector disclosed?

- Notes: By market sector, we refer to an identifiable part of the economy. While established standards exist for describing market sectors, we recognize that developers may provide vague or informal characterizations of market impact. We will award this point if there is a meaningful, though potentially vague or incomplete, summary of affected market sectors.

- References: Vipra & Korinek (2023), Bommasani et al. (2023b)

95. Downstream → Impact → **Affected individuals**

- Definition: Across all forms of downstream use, is the number of individuals affected by the foundation model disclosed?

- Notes: By affected individuals, we principally mean the number of potential users of applications. We recognize that there does not exist an authoritative or consensus standard for what qualifies as an affected individual. We will award this point if there is a meaningful estimate of the number of affected individuals along with a clear description of what it means for an individual to be affected by the model.

- References: Vipra & Korinek (2023), Bommasani et al. (2023b)

96. Downstream → Impact → **Usage reports**

- Definition: Is a usage report that gives usage statistics describing the impact of the model on users disclosed?

- Notes: We recognize that there does not exist an authoritative or consensus standard for what is required in a usage report. Usage statistics might include, for example, a description of the major categories of harm that has been caused by use of the model. We will award this point if there is a meaningful, though potentially vague or incomplete, summary of usage statistics.

- References: Brown (2023), Bommasani et al. (2023b)

97. Downstream → Impact → **Geographic statistics**

- Definition: Across all forms of downstream use, are statistics of model usage across geographies disclosed?

- Notes: We will award this point if there is a meaningful, though potentially incomplete or vague, disclosure of geographic usage statistics at the country-level.

- References: Brown (2023), Bommasani et al. (2023b)

98. Downstream → Impact → **Redress mechanism**

- Definition: Is any mechanism to provide redress to users for harm disclosed?

- Notes: We will also award this point if the developer reports it does not have any such redress mechanism.

- References: Vipra & Myers West (2023), Bommasani et al. (2023b)

99. Downstream → Documentation for Deployers → **Centralized documentation for downstream use**

- Definition: Is documentation for downstream use centralized in a centralized artifact?

- Notes: Centralized documentation for downstream use refers to an artifact, or closely-linked artifacts, that consolidate relevant information for making use of or repurposing the model. Examples of these kinds of artifacts include a website with dedicated documentation information, a github repository with dedicated documentation information, and an ecosystem card. We recognize that different developers may take different approaches to centralizing information. We will award this point if there is a clearly-identified artifact(s) that contains the majority of substantive information (e.g. capabilities, limitations, risks, evaluations, distribution channels, model license, usage policies, model behavior policies, feedback and redress mechanisms, dependencies).

- References: Gebru et al. (2021), Mitchell et al. (2019)

100. Downstream → Documentation for Deployers → **Documentation for responsible downstream use**

- Definition: Is documentation for responsible downstream use disclosed?

- Notes: Such documentation might include details on how to adjust API settings to promote responsible use, descriptions of how to implement mitigations, or guidelines for responsible use. We will also award this point if the developer states that it does not provide any such documentation. For example, the developer might state that the model is offered as is and downstream developers are accountable for using the model responsibly.

- References: Bommasani et al. (2023b), Brown (2023)

## C   Search protocol

In this section, we outline the search process we used to look for evidence that a foundation model developer satisfies our requirements for a given indicator.

### C.1   General search process

#### C.1.1   Keyword Definitions

Each item under review has associated search keywords in our GitHub repository.[69]

#### C.1.2   Model-Item Pair Searches

For every model-item pair, we conduct a search using the defined keywords within the centralized resources associated with the respective models listed below.

#### C.1.3   Search Methodology

We employ the following format for every model-item-keyword tuple while using Google search, and read through the first 10 search results.

```
site:[Refer to developer's website list below] [Refer to model name list below
    ] [Enter keyword]
```

For example, for GPT-4's energy efficiency item, the searches would be:

```
site:openai.com gpt-4 energy
site:openai.com gpt-4 efficien
```

#### C.1.4   Justification

We note the source (e.g., website, company blog post, paper) for each piece of evidence that helped confirm an item is present, alongside the justification. We link to an archive.org URL that contains the justification (instead of linking to developers' pages directly), to maintain records.

#### C.1.5   Avoid Search Personalization

To minimize the influence of personalized search results, we perform all searches in a private or incognito browser tab.

#### C.1.6   Determination Criteria

If we find one piece of evidence that fully justifies 1 point - or, in rarer cases, 0 points - for an item, we don't perform other searches.

#### C.1.7   Distribution Channels

In certain limited cases where the above steps fail to generate any information for indicators related to distribution channels, we interact with the developer's intended distribution channel (if disclosed), such as its API or its preferred deployment partner's API, or the documentation related to this API. We search for the required information via this distribution channel to the extent possible. We also use proxies, such as model playgrounds, if enterprise access is otherwise required.

---

[69]https://www.github.com/stanford-crfm/fmti

### C.2  Developer website

- AI21 Labs (Jurassic-2): `ai21.com`

- Amazon (Titan Text): `aws.amazon.com/bedrock/titan/`

- Anthropic (Claude): `anthropic.com`

- Cohere (Command): `cohere.com`

- Google (PaLM 2): `ai.google`

- Hugging Face (BLOOMZ): `bigscience.huggingface.co`

- Inflection (Inflection-1): `inflection.ai`

- Meta (Llama 2): `ai.meta.com`

- OpenAI (GPT-4): `openai.com`

- StabilityAI (Stable Diffusion 2): `stability.ai`

### C.3  Centralized resources for all models

#### C.3.1  AI21 Labs (Jurassic-2)

- `https://docs.ai21.com/docs/jurassic-2-models`

- `https://docs.ai21.com/docs/responsible-use`

- `https://uploads-ssl.webflow.com/60fd4503684b466578c0d307/61138924626a6981ee09caf6_jurassic_tech_paper.pdf`

- `https://www.ai21.com/blog/introducing-j2`

- `https://docs.ai21.com/docs/responsible-use#usage-guidelines`

- `https://studio.ai21.com/terms-of-use`

- `https://studio.ai21.com/privacy-policy`

- `https://docs.ai21.com/changelog`

#### C.3.2  Amazon (Titan Text)

- `https://aws.amazon.com/bedrock/titan/`

- `https://docs.aws.amazon.com/pdfs/bedrock/latest/APIReference/bedrock-api.pdf#API_ListFoundationModels`

- `https://aws.amazon.com/aup/`

#### C.3.3  Anthropic (Claude 2)

- `https://legal.anthropic.com/#aup`

- `https://vault.pactsafe.io/s/9f502c93-cb5c-4571-b205-1e479da61794/legal.html#aup`

- `https://console.anthropic.com/docs/api/supported-regions`

- `https://legal.anthropic.com/#terms`

- `https://legal.anthropic.com/#privacy`

- `https://docs.anthropic.com/claude/docs`

- `https://www.anthropic.com/index/claude-2`

- `https://www.anthropic.com/earlyaccess`

- `https://www-files.anthropic.com/production/images/Model-Card-Claude-2.pdf`

- `https://www.anthropic.com/index/frontier-threats-red-teaming-for-ai-safety`

### C.3.4 Cohere (Command)

- https://docs.cohere.com/docs/
- https://cohere.com/security
- https://dashboard.cohere.ai/playground/generate
- https://cohere.com/terms-of-use
- https://cloud.google.com/blog/products/ai-machine-learning/accelerating-language-model-training-with-cohere-and-google-cloud-tpus
- https://cohere.com/data-usage-policy
- https://cohere.com/privacy
- https://cohere-inc.secureframetrust.com/

### C.3.5 Google (PaLM 2)

- https://ai.google/static/documents/palm2techreport.pdf
- https://developers.generativeai.google/models/language
- https://policies.google.com/terms/generative-ai/use-policy
- https://developers.generativeai.google/guide/safety_guidance
- https://developers.generativeai.google/products/palm
- https://developers.generativeai.google/available_regions
- https://developers.generativeai.google/terms#content_license_and_data_use

### C.3.6 Hugging Face (BLOOMZ)

- https://arxiv.org/abs/2211.01786
- https://huggingface.co/docs/transformers/model_doc/bloom
- https://huggingface.co/bigscience/bloom
- https://arxiv.org/abs/2303.03915
- https://arxiv.org/abs/2211.05100
- https://proceedings.neurips.cc/paper_files/paper/2022/file/ce9e92e3de2372a4b93353eb7f3dc0bd-Paper-Datasets_and_Benchmarks.pdf

### C.3.7 Inflection (Inflection-1)

- https://inflection.ai/assets/Inflection-1.pdf
- https://inflection.ai/inflection-1
- https://inflection.ai/assets/MMLU-Examples.pdf
- https://heypi.com/policy#privacy
- https://inflection.ai/safety

### C.3.8 Meta (Llama 2)

- https://arxiv.org/pdf/2307.09288.pdf
- https://github.com/facebookresearch/llama/blob/main/MODEL_CARD.md
- https://ai.meta.com/static-resource/responsible-use-guide/

### C.3.9 OpenAI (GPT-4)

- `https://openai.com/research/gpt-4`
- `https://openai.com/policies/usage-policies`
- `https://openai.com/form/chat-model-feedback`
- `https://platform.openai.com/docs`
- `https://openai.com/customer-stories`
- `https://status.openai.com/`
- `https://openai.com/policies/terms-of-use`
- `https://cdn.openai.com/policies/employee-data-privacy-notice.pdf`
- `https://cdn.openai.com/papers/gpt-4-system-card.pdf`
- `https://arxiv.org/pdf/2303.08774.pdf`
- `https://openai.com/research/triton`
- `https://openai.com/pricing`
- `https://platform.openai.com/docs/deprecations`
- `https://openai.com/waitlist/gpt-4-api`
- `https://openai.com/our-structure`
- `https://openai.com/api-data-privacy`

### C.3.10 StabilityAI (Stable Diffusion 2)

- `https://huggingface.co/stabilityai/stable-diffusion-2`
- `https://openreview.net/forum?id=M3Y74vmsMcY`
- `https://huggingface.co/terms-of-service`
- `https://huggingface.co/stabilityai/stable-diffusion-2/blob/main/LICENSE-MODEL`
- `https://platform.stability.ai/legal/terms-of-service`
- `https://stability.ai/use-policy`

## D Calls for transparency

In recent years, transparency has been a rallying cry for activists, a boon to researchers, and a tangible first step for governments interested in regulating foundation models. Here we outline some of the salient calls for transparency to illustrate the different stakeholders with an interest in a more transparent foundation model ecosystem.

**Calls for transparency from governments.**

A wide variety of governments have made transparency in the development of foundation models a top priority in their wider agenda for AI regulation. In the U.S., the White House has secured voluntary commitments from 16 companies that include a commitment "to publicly reporting their AI systems' capabilities, limitations, and areas of appropriate and inappropriate use" in the form of "transparency reports."[70] The AI Risk Management Framework from the U.S. National Institute for Standards and Technology outlines the U.S. federal government's current approach to transparency for foundation models and other AI systems.[71] The AI Risk Management Framework states "Trustworthy AI depends upon accountability. Accountability presupposes transparency. Transparency reflects the extent to which information about an AI system and its outputs is available to individuals interacting with such a system ... Meaningful transparency provides access to appropriate levels of information based on the stage of the AI lifecycle and tailored to the role or knowledge of AI actors or individuals interacting with or using the AI system."

The SAFE framework for regulating AI proposed by Senate Majority Leader Schumer aims to ensure that "AI is developed and deployed in a responsible and transparent manner" and to "support US-led innovation in AI technologies—including innovation in security, transparency and accountability."[72] Transparency is also one of the five pillars of the bipartisan framework for a U.S. AI Act proposed by Senators Hawley and Blumenthal; their framework specifically suggests "requiring transparency from the companies developing and deploying A.I. systems" as it relates to training data, limitations, accuracy, safety, and user interaction with an AI system.[73] A variety of other draft legislation in the U.S. would require a higher level of transparency for foundation model developers, such as the Algorithmic Accountability Act[74] at the federal level and California's Safety in Artificial Intelligence Act.[75]

In the EU, transparency and information sharing have become a central focus of the draft EU AI Act. For instance, Article 52 of the Act imposes "transparency obligations" for some types of AI systems. The European Parliament's draft of the AI Act included specific obligations for foundation model developers: "foundation models should have information obligations and prepare all necessary technical documentation for potential downstream providers to be able to comply with their obligations under this Regulation. Generative foundation models should ensure transparency about the fact the content is generated by an AI system, not by humans."[76] Developers of high-risk AI systems may also be required to provide additional transparency about their systems such that deployers have adequate information about risks and how to mitigate them.

China has gone a step further, with the central government adopting regulations that impose transparency requirements on foundation model deployers. China's "Interim Measures for the Management of Generative Artificial Intelligence Services" state that organizations deploying foundation models, including via an API, must "employ effective measures to increase transparency in generative AI services."[77] The law further

---

[70]See https://www.whitehouse.gov/briefing-room/statements-releases/2023/07/21/fact-sheet-biden-harris-admin istration-secures-voluntary-commitments-from-leading-artificial-intelligence-companies-to-manage-the-risks-p osed-by-ai/ and https://www.whitehouse.gov/wp-content/uploads/2023/07/Ensuring-Safe-Secure-and-Trustworthy-AI. pdf and https://www.whitehouse.gov/briefing-room/statements-releases/2023/09/12/fact-sheet-biden-harris-admin istration-secures-voluntary-commitments-from-eight-additional-artificial-intelligence-companies-to-manage-the -risks-posed-by-ai/ and https://www.whitehouse.gov/wp-content/uploads/2023/09/Voluntary-AI-Commitments-Septemb er-2023.pdf

[71]https://nvlpubs.nist.gov/nistpubs/ai/NIST.AI.100-1.pdf

[72]https://www.democrats.senate.gov/imo/media/doc/schumer_ai_framework.pdf

[73]https://www.blumenthal.senate.gov/imo/media/doc/09072023bipartisanaiframework.pdf

[74]See https://www.congress.gov/bill/118th-congress/house-bill/5628/all-info?s=2&r=1 and https://docs.google. com/document/d/1A1bJ1mkIfE3eZuSbDmz3HGVtOvQDegHl53q3ArO7m44/

[75]https://leginfo.legislature.ca.gov/faces/billTextClient.xhtml?bill_id=202320240SB294

[76]https://www.europarl.europa.eu/doceo/document/TA-9-2023-0236_EN.pdf

[77]http://www.cac.gov.cn/2023-07/13/c_1690898327029107.htm

specifies that "providers shall formulate clear, specific, and feasible tagging rules" for data and that "providers shall establish and complete mechanisms for making complaints and reports, setting up easy complaint and reporting portals, disclosing the process for handling them and the time limits for giving responses."

Many other governments have also highlighted the importance of transparency in the development and use of foundation models. Canada has released a "Voluntary Code of Conduct on the Responsible Development and Management of Advanced Generative AI Systems," which has been signed by Cohere, the Montreal Institute for Learning Algorithms, and the Vector Institute among other organizations.[78] Canada's Voluntary Code of Conduct states that signatories commit to achieve transparency such that "sufficient information is published to allow consumers to make informed decisions and for experts to evaluate whether risks have been adequately addressed." It further specifies that "developers of advanced generative systems available for public use" are required to "Publish information on capabilities and limitations of the system... Develop and implement a reliable and freely available method to detect content generated by the system, with a near-term focus on audio-visual content (e.g., watermarking). ... Publish a description of the types of training data used to develop the system, as well as measures taken to identify and mitigate risks." Japan is reportedly in the process of adopting its own code of conduct, which may go beyond voluntary commitments.[79]

India's report on "Impact, Opportunity, and Challenges of Generative AI," coauthored by India's Ministry of Electronics and Information Technology, states that transparency should be a central feature of India's regulatory framework for ensuring responsible use of generative AI.[80] The United Arab Emirates' generative AI guide, published by the Office of the Minister for Artificial Intelligence, Digital Economy, and Remote Work Applications, highlights the importance of transparency for generative AI in terms of data protection: "Transparency is crucial to data privacy because it enables individuals to know how their data is collected, processed, and used by organizations. By being transparent, organizations can provide clear and concise information about their data privacy practices, policies, and procedures."[81] Data protection authorities around the world are "de facto regulating generative AI" by using their existing authorities, including those related to information sharing; for example, data protection authorities in Brazil, Japan, and South Korea launched investigations into OpenAI's ChatGPT in 2023.[82]

Some governments have highlighted the fact that existing transparency requirements already apply to foundation model developers and ought to be enforced as such. The UK Competition and Markets Authority notes that transparency requirements are already in place under consumer protection law, and that foundation model developers must comply with the transparency provisions of the UK Consumer Rights Act.[83] The U.S. Federal Trade Commission has stated that "we take note–and can take action–if companies aren't upfront about what consumers are buying, who made it, how it was made, or what rights people have in their own creations. ... When offering a generative AI product, [companies] may need to tell customers whether and the extent to which the training data includes copyrighted or otherwise protected material."[84]

It is also worth noting that many governments have emphasized the importance of transparency in the development and use of AI systems outside of the context of foundation models. The national AI strategies of Colombia,[85], Egypt,[86] Indonesia,[87] and India[88] highlight the importance of transparency as do the national AI strategies of other countries.[89]

---

[78]https://ised-isde.canada.ca/site/ised/en/voluntary-code-conduct-responsible-development-and-management-advanced-generative-ai-systems

[79]https://english.kyodonews.net/news/2023/10/3b83adf1e28d-japans-ai-draft-guidelines-ask-for-measures-to-address-overreliance.html

[80]https://indiaai.s3.ap-south-1.amazonaws.com/docs/generative-ai-report.pdf

[81]https://ai.gov.ae/wp-content/uploads/2023/04/406.-Generative-AI-Guide_ver1-EN.pdf

[82]https://fpf.org/blog/how-data-protection-authorities-are-de-facto-regulating-generative-ai/

[83]https://www.gov.uk/government/publications/ai-foundation-models-initial-report

[84]https://www.ftc.gov/business-guidance/blog/2023/08/cant-lose-what-you-never-had-claims-about-digital-ownership-creation-age-generative-ai

[85]https://colaboracion.dnp.gov.co/CDT/Conpes/Económicos/3975.pdf

[86]https://mcit.gov.eg/Upcont/Documents/Publications_672021000_Egypt-National-AI-Strategy-English.pdf

[87]https://ai-innovation.id/images/gallery/ebook/stranas-ka.pdf

[88]https://www.niti.gov.in/sites/default/files/2019-01/NationalStrategy-for-AI-Discussion-Paper.pdf

[89]https://oecd.ai/en/dashboards/overview

**Calls for transparency from international organizations.**   The UN High Commissioner for Human Rights, Volker Türk, has argued that existing rules for businesses squarely apply to foundation model developers. In a speech in July 2023, Türk stated that generative AI "companies must live up to their responsibilities to respect human rights in line with the Guiding Principles on Business and Human Rights."[90] In addition to requiring human rights due diligence, the UN Guiding Principles on Business and Human Rights explicitly refer to transparency as it relates to a company's obligation to (i) transparently communicate the human rights impact of its products and (ii) be transparent in administering grievance processes.[91]

Türk further argued that without adequate guarantees of transparency, generative AI and other types of AI systems should be banned or suspended. He said "regulations need to require assessment of the human rights risks and impacts of AI systems before, during, and after their use. Transparency guarantees, independent oversight, and access to effective remedies are needed, particularly when the State itself is using AI technologies. AI technologies that cannot be operated in compliance with international human rights law must be banned or suspended until such adequate safeguards are in place."

UN Secretary-General António Guterres has foregrounded transparency as well. The UN's digital agenda, summarized in Guterres' Global Digital Compact, makes three key proposals related to transparency: (i) the international community should "make transparency, fairness and accountability the core of AI governance," (ii) governments should "consider the adoption of a declaration on data rights that enshrines transparency," and (iii) researchers and companies should be responsible for transparently communicating the risks of AI systems.[92]

The G7 Hiroshima AI Process, which was launched in May 2023 and focuses on generative AI, makes "promotion of transparency" one of its core aims.[93] A September 2023 joint statement on the Hiroshima AI Process by G7 Digital and Technology Ministers committed the G7 to "develop guiding principles for organizations developing, deploying, and using advanced AI systems, in particular foundation models and generative AI," and stated that one such guiding principle could be "publicly report models' capabilities, limitations and domains of appropriate and inappropriate use, ensuring sufficient transparency."[94]

More broadly, international organizations have long noted that transparency is essential for responsible development of AI systems. The OECD AI Principles, adopted in 2019, include transparency as one of five principles for trustworthy AI. The principle on "transparency and explainability" reads: "AI Actors should commit to transparency and responsible disclosure regarding AI systems. To this end, they should provide meaningful information, appropriate to the context, and consistent with the state of art: (i) to foster a general understanding of AI systems; (ii) to make stakeholders aware of their interactions with AI systems, including in the workplace; (iii) to enable those affected by an AI system to understand the outcome; and, (iv.) to enable those adversely affected by an AI system to challenge its outcome based on plain and easy-to-understand information on the factors, and the logic that served as the basis for the prediction, recommendation or decision."[95] The G20 AI Principles, also adopted in 2019, include this OECD principle on transparency verbatim. [96] A number of other countries have committed to the OECD AI Principles, including Argentina, Brazil, Egypt, and Singapore.[97]

---

[90] https://www.ohchr.org/en/statements/2023/07/artificial-intelligence-must-be-grounded-human-rights-says-high-commissioner

[91] For instance, the UN Guiding Principles on Business and Human Rights state, "The responsibility to respect human rights requires that business enterprises have in place policies and processes through which they can both know and show that they respect human rights in practice. Showing involves communication, providing a measure of transparency and accountability to individuals or groups who may be impacted and to other relevant stakeholders, including investors." See https://www.ohchr.org/sites/default/files/documents/publications/guidingprinciplesbusinesshr_en.pdf

[92] https://indonesia.un.org/sites/default/files/2023-07/our-common-agenda-policy-brief-gobal-digi-compact-en.pdf

[93] https://www.whitehouse.gov/briefing-room/statements-releases/2023/05/20/g7-hiroshima-leaders-communique/

[94] https://www.politico.eu/wp-content/uploads/2023/09/07/3e39b82d-464d-403a-b6cb-dc0e1bdec642-230906_Ministerial-clean-Draft-Hiroshima-Ministers-Statement68.pdf

[95] https://legalinstruments.oecd.org/en/instruments/OECD-LEGAL-0449

[96] https://wp.oecd.ai/app/uploads/2021/06/G20-AI-Principles.pdf

[97] https://oecd.ai/en/ai-principles

**Calls for transparency from foundation model developers.**   Foundation model developers have also called for greater transparency and touted the benefits of transparency in their own business practices. For example, in June 2022 AI21 Labs, Cohere, and OpenAI published "Joint Recommendation for Language Model Deployment" that advocated for increased transparency (Cohere, 2022). Their recommendations stated that developers should "Publish usage guidelines and terms of use of LLMs ... Document known weaknesses and vulnerabilities, such as bias or ability to produce insecure code ... Documentation should also include model and use-case-specific safety best practices."

Individual developers have highlighted the importance of transparency as well. Anthropic ties the importance of transparency to interpretability in its paper on Constitutional AI and in describing the company's "Core Views on AI Safety" (Bai et al., 2022).[98] Inflection prioritizes transparency in its decision-making about the choices it makes with regard to safety. Inflection's Safety Policy states "Safety at its heart is a question of values. Companies choose what risks to prioritize, and how to address them. We believe the best principle is to be deliberate about these choices, and transparent with our users about the specific values we build into our AIs. We may prioritize values that you disagree with. That's OK. We think that there is room for many perspectives ... We commit to sharing publicly what positions we aim to take in our AIs."[99]

OpenAI has argued that transparency can help companies work together to mitigate safety concerns regarding foundation models.[100] Askell et al. (2019) argue "information that companies provide about their intentions and actions—how transparent they are—can play an important role in whether other companies will cooperate with them." OpenAI also requires transparency from its suppliers: OpenAI's Supplier Code of Conduct states that "OpenAI expects all Suppliers to adhere to the highest standards of integrity, transparency, honesty, and ethical conduct in all their business dealings."[101]

Cohere states that transparency is important for its responsible development of large language models, noting that it has "invested in technical and non-technical measures to mitigate potential harm and make our development processes transparent."[102] Cohere's Usage Guidelines prohibit users from using Cohere's platform for applications with "no transparency," meaning those that "do not disclose that the content is generated through automated means."[103]

Stability AI has called for transparency in connection with its advocacy for open foundation models. In a May 2023 report submitted to the U.S. Senate Judiciary Subcommittee on Privacy, Technology, and the Law, Stability AI wrote "Models like Stable Diffusion and StableLM demonstrate our commitment to AI technology that is transparent, accessible, and human-centric: ... We develop open models for transparency. Researchers can 'look under the hood' to verify performance, identify potential risks, and help develop safeguards. Organizations across the public and private sector can customize these models for their own needs without exposing sensitive data or ceding control of their AI capabilities."[104] The report further argues "These principles can help to advance important policy objectives. Transparent models promote safety and security. ... open models enable the transparent identification, assessment, and management of risks consistent with the National Institute of Standards and Technology AI Risk Management Framework."

Hugging Face has also called for transparency as part of its push for open foundation models. In written testimony before the U.S. House Committee on Science, Space, and Technology, Hugging Face CEO Clement Delangue stated "Rigorous documentation practices for AI systems, with transparent reporting that follows well-defined protocols, serves three main goals: incentivizing responsible development; ensuring researchers and developers consider values and priorities that may otherwise be overlooked; and creating a paper trail for review. ... transparency from entities about how and where they deploy AI systems to understand what

---

[98]As the blog post summarizing the paper states, "Constitutional AI is also helpful for transparency: we can easily specify, inspect, and understand the principles the AI system is following." See `https://www.anthropic.com/index/claudes-constitution` and `https://www.anthropic.com/index/core-views-on-ai-safety`

[99]`https://inflection.ai/safety`

[100]`https://openai.com/research/cooperation-on-safety`

[101]`https://openai.com/policies/supplier-code`

[102]`https://cohere.com/responsibility`

[103]`https://docs.cohere.com/docs/usage-guidelines`

[104]`https://stability.ai/blog/stability-ai-letter-us-senate-ai-oversight`

evaluations are most urgently needed."[105] Hugging Face has, along with various partners, released a number of artifacts that advance transparency such as tools for exploring datasets (Piktus et al., 2023).

In articulating Meta's position with respect to Llama 2, Touvron et al. (2023) state that "It is important to understand what is in the pretraining data both to increase transparency and to shed light on root causes of potential downstream issues, such as potential biases. ... open releases promote transparency and allow more people to access AI tools, democratizing the technology and decentralizing AI expertise." Meta's Responsible Use Guide for Llama 2 encourages downstream developers to "build transparency and reporting mechanisms in user interactions ... consider ways to provide transparency to end users regarding potential risks and limitations of the system prior to or at the time of user interaction." [106]

Amazon makes clear that transparency is important with respect to the way in which it communicates its policies to users. Amazon Web Services' Data Privacy Center states that "Our contracts are written in plain, straightforward language to be transparent and help you understand the data privacy protections that we offer. We also provide ongoing data transparency reporting."[107]

Google highlights transparency in its AI principles, writing "For datasets and models, the consistent outcome is to create and publish detailed documentation of datasets and models in the form of structured transparency artifacts known as data and model cards (see the following section for details), which function like nutrition labels, providing information such as the provenance of the data (if a data card) and model performance when tested for fairness (if a model card)."[108] Google's AI principles also detail the "Transparency Artifacts" that Google researchers have built, such as Healthsheets and a Data Cards Playbook.

Microsoft has also produced such artifacts, namely in the form of "Transparency Notes," which "are intended to help you understand how our AI technology works, the choices system owners can make that influence system performance and behavior, and the importance of thinking about the whole system, including the technology, the people, and the environment."[109]

A large number of developers and deployers that we do not assess have also expressed the importance of transparency (Jobin et al., 2019; Fjeld et al., 2020; WEF, 2023). Notable among them is EleutherAI, a non-profit research group that is a leading developer of open foundation models (Skowron & Biderman, 2023). Phang et al. (2022) write that "EleutherAI's approach to research goes beyond transparency: by doing research entirely in public, anyone in the world can observe and contribute at every stage," adding that such public-facing research fosters a highly collaborative, diverse, and innovative research community.

**Calls for transparency from researchers, civil society, and labor.** While governments and companies have consistently underscored the value of transparency, less powerful actors have banded together to push public and private entities to meaningfully improve transparency along with the business practices that transparency uncovers.

Researchers have driven much of the improvement in transparency for foundation model developers, with innovations like model cards, datasheets, and data statements leading to substantial gains (Mitchell et al., 2018; Gebru et al., 2018; Bender & Friedman, 2018a). Some have sought to solidify these improvements in transparency by strengthening the field of algorithmic auditing (Costanza-Chock et al., 2022). Mozilla's Open Source Audit Tooling project calls for better infrastructure to evaluate and audit AI systems (Raji, 2022). Another proposal to bolster the auditing ecosystem is for governments to conduct third-party audits of AI systems under their existing authority to protect consumers and data subjects (Miller, 2021).

Recently, coalitions of researchers led by organizations like LAION have come together to call for greater transparency in the foundation model ecosystem (LAION, 2023). In recent congressional hearings, expert testimony has expressed "The Need for Transparency in Artificial Intelligence" (Gregory, 2023). Belli & Gaspar (2023) detail the central importance of transparent foundation models from the perspective of

---

[105]https://republicans-science.house.gov/_cache/files/5/5/551f066b-4483-4efd-b960-b36bc02d4b66/B82DBAFFA56F31
799E058FB2755C2348.2023-06-22-mr.-delangue-testimony.pdf
[106]https://ai.meta.com/static-resource/responsible-use-guide/
[107]https://aws.amazon.com/compliance/data-privacy/Privacy_at_AWS_
[108]https://ai.google/static/documents/ai-principles-2022-progress-update.pdf
[109]https://learn.microsoft.com/en-us/legal/cognitive-services/language-service/transparency-note

experts across Asia, Africa, and Latin America. Other researchers still have argued that transparency, while necessary, is far from sufficient to regulate AI (Hartzog, 2023).

Data workers employed as contractors by foundation model developers have also mobilized for increased transparency (Gray & Suri, 2019b).[110] For example, in July 2023 members of the African Content Moderators Union filed a petition with Kenya's parliament requesting an investigation into OpenAI, Meta, Google, and other multinational technology companies that employ content moderators in Kenya.[111] The petition states that OpenAI used a vendor, Sama, to hire the petitioners as contractors who "trained the ChatGPT algorithm," and alleges that "the contracts did not sufficiently describe the nature of the job ... we were not properly informed of the nature of the work we would be undertaking." The petition further alleges that although this data labor included "reading and viewing material that depicted sexual and graphic violence and categorizing it accordingly so that ChatGPT's artificial intelligence could learn it for the purposes of its future interactions with people ... throughout the contract of training ChatGPT we were not afforded psychosocial support."[112]

The Partnership on AI has advocated for transparency with respect to the employment of data enrichment workers, writing "While shifting how the broader field approaches data enrichment is not a trivial task, increasing transparency regarding current practices and developing more practical guidance can move the field towards improved conditions for data enrichment workers. Greater transparency can help emphasize the central role of data enrichment workers, create the basis for a rich public dialogue of how to improve conditions for workers, and increase confidence in AI models themselves."[113]

Civil society groups with a range of different focus areas agree that transparency is a pressing priority for policymakers and foundation model developers. For instance, 123 civil society organizations, including AccessNow, Algorithm Watch, and the European Center for Not-for-Profit Law, released a statement advocating for the prioritization of more serious transparency requirements in the EU AI Act.[114] The statement advocates the inclusion of a "mandatory impact assessments are a crucial measure to ensure foresight and accountability for potential AI-related harms," and that "information on all uses of AI systems by public authorities, regardless of the systems' risk level, should be made public in the EU database." Additionally, they call for "an obligation for providers and/or users to include information regarding the environmental impact of AI systems," which is not a provision in the EU AI Act. Freedom House has also warned that "AI has allowed governments to refine their online censorship" and threatens to exacerbate the decline in global internet freedom. AI has allowed governments to enhance and refine their online censorship, and foundation models may exacerbate this trend.[115] Freedom House points to transparency requirements as a mechanism to identify and combat evolving and subtle censorship pressures.

In October 2023, the U.S. Federal Trade Commission convened a workshop on the "Creative Economy and Generative AI," where creators from across different industries demanded increased transparency. In the words of one participant, "The creative economy only works when the basic tenants of consent, credit, compensation, and transparency are followed. ... Without transparency, we can't even know the extent of

---

[110]Some policymakers have focused on the importance of transparency with respect to data labor. For example, in a letter to the CEOs of major foundation model developers, eight members of the U.S. Congress wrote "Tech companies also must be more transparent about the role data workers play in their AI, so that consumers can make informed choices about the products they use. Unfortunately, many companies have sidestepped these duties, and that must change. ... Please share any plans your company has to be more transparent about the role its data workers play and their working conditions." See https://www.markey.senate.gov/imo/media/doc/letter_to_artificial_intelligence_companies_on_data_worker_labor_conditions_-_091323pdf1.pdf

[111]The African Content Moderators Union has also sued Meta, alleging that it unlawfully fired workers for their union organizing. See https://techcrunch.com/2023/08/23/meta-and-moderators-agree-to-mediation/

[112]https://x.com/mercymutemi/status/1678984336996028416?s=46

[113]In addition to conducting a case study in partnership with Google DeepMind exploring how to increase transparency regarding data labor, the Partnership on AI has separately published a white paper recommending that developers increase transparency in wages and pay structure for data enrichment workers. See https://partnershiponai.org/wp-content/uploads/2022/11/case-study_deepmind.pdf and http://partnershiponai.org/wp-content/uploads/2021/08/PAI-Responsible-Sourcing-of-Data-Enrichment-Services.pdf

[114]https://www.fairtrials.org/app/uploads/2022/05/Civil-society-reacts-to-EP-AI-Act-draft-report_FINAL.pdf

[115]https://freedomhouse.org/report/freedom-net/2023/repressive-power-artificial-intelligence

how much of these companies have taken. They took our work and data to train for-profit technologies that then directly compete against us in our own markets using generative media that is meant to mimic us."[116]

Despite its limits, transparency is a necessary and broadly popular first step towards accountability for harm caused by AI systems (Kaminski, 2020; Bates et al., 2023). In the context of the rapid rollout of extremely powerful AI systems such as foundation models, transparency is all the more urgent. Companies developing and deploying foundation models should heed the call.

---

[116]https://www.ftc.gov/system/files/ftc_gov/pdf/creative-economy-and-generative-ai-transcript-october-4-2023.pdf

