# OpenReview forum: "The 2023 Foundation Model Transparency Index"
_TMLR — Accepted by TMLR_

### Review · Reviewer_C82T · 2024-10-22

**Summary Of Contributions:**

## Summary:
The paper introduces the Foundation Model Transparency Index, which aims to codify the transparency of foundation models by defining 100 fine-grained indicators cover aspects from the upstream resources used in model development, details about the models themselves, to downstream usage and distribution channels. The author evaluates 10 major foundation model developers, and presents 10 key findings about the current state of transparency in foundation models, emphasizing the lack of disclosure on the downstream impact and how users can seek redress for harm.

## Contributions:
- The introduction of a comprehensive transparency index that uses 100 detailed indicators.
- Evaluation of 10 flagship models from leading developers.
- Presentation of top-level findings on the transparency of foundation model developers.
## Strengths:
- The development of a standardized index to assess foundation model transparency is a significant contribution to the field.
- The detailed indicators across various dimensions (upstream resources, model characteristics, downstream use) provide a well-rounded assessment.
- The report reveals important gaps in transparency, especially regarding the downstream impact of these models, which is crucial for governance and public accountability.

**Audience:**

Yes

**Claims And Evidence:**

Yes

**Requested Changes:**

- Page 1 and page 17, footnote exceed the margin.
- Figures 3, 4, and 5 also exceed the margins; adjusting the table width and reducing space between rows could resolve this.
- Inconsistent formatting of subsections: Sections 5.1 and 5.2 begin on new pages, while subsections elsewhere in the paper do not. - Standardizing this throughout the paper would improve consistency.
- There’s odd spacing on page 32.
- Text following Figures 10, 11, and 12 appears to be misformatted, possibly due to unprocessed LaTeX commands. This needs correction to ensure clarity.
- Section 3, which briefly introduces the index, feels underdeveloped, given that the core contribution of the paper is the transparency index itself. I will suggest to merge Section 3 with Section 4.

**Strengths And Weaknesses:**

- Lack of Commercial Consideration: The paper recommends increasing transparency across multiple facets of foundation model development and use, but it does not sufficiently address the commercial implications. Foundation models are highly competitive, with companies investing billions of dollars in developing these technologies. Expecting them to disclose transparency details, especially related to core competitive aspects like data and training processes, might not be practical. The recommendation for transparency levels comparable to research institutions may not align with the commercial reality of the foundation model market, where proprietary technologies offer a significant competitive edge.
- Inclusion of More Models: The report was completed in September 2023, but since it’s now October 2024, many new LLMs have been developed. While it’s understandable that journal papers take time, I still recommend including more recent models to keep the analysis up to date. Specifically, the report focuses on models from 10 major developers, but this selection may introduce bias, particularly as it seems based on market dominance. Including other models, such as OLMo, LLM360, and models from Chinese developers like Qwen and DeepSeek, would provide a more diverse perspective. Relying solely on developers from established companies could skew the findings, as these companies often prioritize commercial secrecy over transparency. Including models from academic institutions or models explicitly aimed at higher transparency would give a more balanced view.
- Structural and Formatting Issues (see Requested Changes)

---

> ### Author Response · Authors · 2024-12-21
> **Author response to Reviewer C82T**
>
> We thank the reviewer for their thorough response and appreciate that they believe this work is "a a significant contribution to the field", that the indicators provide a "well-rounded assessment" and that our report "reveals important gaps in transparency ..., which is crucial for governance and public accountability". To respond to the specific points raised:
>
>
> > Lack of Commercial Consideration: The paper recommends increasing transparency across multiple facets of foundation model development and use, but it does not sufficiently address the commercial implications. Foundation models are highly competitive, with companies investing billions of dollars in developing these technologies. Expecting them to disclose transparency details, especially related to core competitive aspects like data and training processes, might not be practical. The recommendation for transparency levels comparable to research institutions may not align with the commercial reality of the foundation model market, where proprietary technologies offer a significant competitive edge.
>
>
> We agree with the reviewer: many reasons, including matters of commercial interest, trade secret, and competitive advantage, will countervail transparency. We especially highlight that our report makes the heterogeneity in transparency practices very clear: in many cases, one company discloses information where other competitors do not. For example, the max score for any company is 54 (Meta) and the average is a 37 yet 82 indicators are satisfied by at least one company. So, even though all the companies are deeply invested in their commercial outcomes, they clearly make very different decisions on what to disclose publicly.
>
> Build on this, our intent with the Index is not to say that every company should score a 100 today. While there are clear benefits to such public transparency that we articulate, we think of our work as clarifying exactly what is transparent today. We see this as a vital primitive because we believe that, equipped with a precise understanding of the status, we can better enable the public, government, and other stakeholders to engage in determining how we should trade-off between commercial benefits from opacity and public benefits from transparency.
>
> > Inclusion of More Models: The report was completed in September 2023, but since it’s now October 2024, many new LLMs have been developed. While it’s understandable that journal papers take time, I still recommend including more recent models to keep the analysis up to date. Specifically, the report focuses on models from 10 major developers, but this selection may introduce bias, particularly as it seems based on market dominance. Including other models, such as OLMo, LLM360, and models from Chinese developers like Qwen and DeepSeek, would provide a more diverse perspective. Relying solely on developers from established companies could skew the findings, as these companies often prioritize commercial secrecy over transparency. Including models from academic institutions or models explicitly aimed at higher transparency would give a more balanced view.
>
>
> We agree that evaluating new companies and models is essential, but we emphasize the Index is an annual report (this paper is for the 2023 Index). Namely, we highlight that the 2024 Index was published in May 2024, though we do not cite or point to it currently because of the clear risk of deanonymization in the peer review process, which we are not sure how to navigate. So, in short we emphasize that 2024 Index was conducted, published in May 2024, and covered 14 developers {8 that overlap with 2023; 2 of the original 10 did not engaged} that submitted structured transparency reports for the first time that were then made public. And we are currently in the progress of conducting the 2025 Index.

---

### Review · Reviewer_ZEgq · 2024-11-02

**Summary Of Contributions:**

This paper describes the authors' overall work around the Foundation Model Transparency Index (referred to as "the index" in this review) as of 2023. Reflecting the global trends of extensive application of foundation models, mostly led by development of large language models and image generation models, the authors designed the index to assess whether the developers of the foundation models disclosed appropriate information about/around the model.

The index consists of 100 indicators, grouped by 3 domains and more specific subdomains. The domains are upstream: how the models are constructed, model: what the models are, and downstream: how the models can be applied, capturing a variety of perspectives of transparency. The index measures the transparency of model developers, not the model itself. Each indicator is a binary classifier having a definition, detailed explanation, and several references to support its importance.

The authors measured the index of 10 flagship model developers by assessing publicly available information of the models and receiving rebuttals from the developers. Authors then analysed the obtained indicators and provided many insights on them, including overall tendency of each developer, tendency of each domain, comparison between open/closed model developers, and correlation between each specific developer.

Given the assessment above, the authors finally proposed several recommendations for model developers, deployers, and policymakers, to improve the transparency around the foundation model.

**Audience:**

Yes

**Broader Impact Concerns:**

This paper contains information that will directly affect against every participant of the foundation model market, especially model developers, deployers, and policymakers, but I don't think we need ethical care on the content itself.

**Claims And Evidence:**

Yes

**Requested Changes:**

Requests for changes are almost stated in the Weaknesses subsection in the previous response. I think there is no essential flaw on the paper, but the paper contains somewhat less information than I expected.

In addition, the paper contains many presentation errors, especially misformatting citations and figures, requiring overall verification of its appropriateness.

**Strengths And Weaknesses:**

## Strengths

This is a massive and well-organized work. The information that indicators refer to is dominated by that which is deemed important to users of the underlying model, and it is a natural desire that this information should be provided by model developers. The assessment was carried out with the involvement of actual model developers, which is an important achievement in ensuring the effectiveness of the index. By comparing discrepancies between model developers, it is also important that the authors point out that they could probably easily improve some transparency.

## Weaknesses

Whether each model developer can improve a particular indicator depends largely on the model developer's stance. Even if another model developer meets a particular indicator, it is not possible to directly determine whether it is an easy improvement item for another developer. In particular, as shown in the paper, the indicator response trends of open developers and closed developers are very different, which is likely to reflect fundamental differences in positions, and it is thought that it will not be easy to bridge this gap. Of course, it is clear that formulating an index has the effect of motivating model developers to disclose appropriate information. As a more in-depth analysis, it would be appropriate to have a detailed discussion for each model developer about why a particular indicator is as it is at present and whether there is a possibility of improving it in the future. However, this discussion would require deeper insight into each developer than is currently written in the paper. For this reason, it is thought that there is no problem in simply indicating it as a direction for future research. As a note, the rebuttals of each developer are likely to be useful in analyzing this information. Since the authors have such information, in principle, these analyses would be possible. However, it is thought that this paper intentionally avoids detailed analysis on it.

The recommendations in the paper do not include any suggestions for end users of foundation models or the market itself. It is clear that developers and policymakers have a direct influence on transparency, but their decisions are essentially influenced by public opinion formed by market participants. In my personal experience, foundation model users often only pay attention to its downstream performance. In contrast, they often do not have much interest in whether transparency is guaranteed. It is easy to think that this lack of interest will be a factor that hinders the spread of transparency, and it is necessary to encourage market participants to correctly understand the significance of the index. Also, each recommendation in the paper may be perceived as somewhat subjective, though they looks appropriate in my personal opinion. It would be better to mention which recommendation is effective in improving which indicator.

Finally, although this is mentioned briefly in the limitations, the paper does not provide detailed development procedures for the index and protocols for future improvements. Considering the long-term operation of the index, these are more strategically important than the individual indicators currently adopted, and it would be better to focus on their development.

---

> ### Author Response · Authors · 2024-12-21
> **Author response to Reviewer ZEgq**
>
> We thank the reviewer for their thorough response and appreciate that they believe this work is "a massive and well-organized work". To respond to the specific points raised:
>
> > Whether each model developer can improve a particular indicator depends largely on the model developer's stance. Even if another model developer meets a particular indicator, it is not possible to directly determine whether it is an easy improvement item for another developer. In particular, as shown in the paper, the indicator response trends of open developers and closed developers are very different, which is likely to reflect fundamental differences in positions, and it is thought that it will not be easy to bridge this gap. Of course, it is clear that formulating an index has the effect of motivating model developers to disclose appropriate information. As a more in-depth analysis, it would be appropriate to have a detailed discussion for each model developer about why a particular indicator is as it is at present and whether there is a possibility of improving it in the future. However, this discussion would require deeper insight into each developer than is currently written in the paper. For this reason, it is thought that there is no problem in simply indicating it as a direction for future research.
>
>
> We fully agree with the reviewer that just because company A discloses information sufficient to award an indicator, this does not imply it will be easy for company B (an obvious example is when we award company A an indicator for releasing model weights where company B's business model hinges on retaining control to model weights). With this in mind, we do believe showing precedent of another major company disclosing the information is very useful for making the argument for greater transparency, even if other factors may countervail it (e.g. different business models). Two more general points to raise are: (i) as we conduct subsequent Index iterations, we will empirically reveal what is more malleable vs. stubborn to change and (ii) through conducting this research, we will better characterize the developer-internal cost models for transparency and where there are barriers due to organization, technical or other factors or where other values are at odds with transparency for specific indicators.
>
>
> > The recommendations in the paper do not include any suggestions for end users of foundation models or the market itself. It is clear that developers and policymakers have a direct influence on transparency, but their decisions are essentially influenced by public opinion formed by market participants. In my personal experience, foundation model users often only pay attention to its downstream performance. In contrast, they often do not have much interest in whether transparency is guaranteed. It is easy to think that this lack of interest will be a factor that hinders the spread of transparency, and it is necessary to encourage market participants to correctly understand the significance of the index.
>
>
> We fully agree, though our belief at this time is the Index is most useful for engaged stakeholders (i.e. model developers, academics working in this area, policymakers working on AI policy). While we do believe the broader array of users and consumers of foundation models should be better informed, we did not yet feel the Index (referring to the 2023 version that is the subject of this paper/review) was yet ready to be maximally legible to this much broader array of stakeholders (who, instead, may focus on ChatbotArena/HELM/OpenLLM Leaderboard/MMLU/GPQA or other model-level evaluation scores to make decisions). Moving forward, we agree that the market should be cognizant of FMTI but we are still working on the strategy to best do this.
>
>
> > Finally, although this is mentioned briefly in the limitations, the paper does not provide detailed development procedures for the index and protocols for future improvements. Considering the long-term operation of the index, these are more strategically important than the individual indicators currently adopted, and it would be better to focus on their development.
>
> This is an excellent point that reflects the early status of the Index at the time of writing, where we had only begun to consider the long-term operation. Since then we have successfully published the 2024 Index and are currently working on the 2025 Index. We would be happy to discuss this more extensively, though this discussion in our view would make more sense on the live FMTI website as opposed to the 2023 FMTI paper, but we would be happy to include some version of it there if that seems appropriate.

---

> > ### Author Response · Authors · 2024-12-21
> > **Note on developer rebuttals**
> >
> > To address a point made in passing in the review about the company rebuttals to the initial FMTI scores:
> >
> > > As a note, the rebuttals of each developer are likely to be useful in analyzing this information. Since the authors have such information, in principle, these analyses would be possible. However, it is thought that this paper intentionally avoids detailed analysis on it.
> >
> >
> > To clarify, the initial rebuttals sent to us and the subsequent communication does, indeed, prove useful for understanding this. We do describe some of this information in an aggregate/anonymized way, but given we gave the companies prior notice that we would not disclose this intermediary communication further, we deliberately chose to not do much on this. In part, this reflects a strategic belief on our part that is it is very valuable to built trust with companies for the first iteration of this process, even if this means not performing the maximal analysis possible. In our view, this has been very beneficial as can be seen in the 2024 Index which has the companies directly prepare structured transparency reports that are made public for the first time.

---

### Review · Reviewer_mY2L · 2024-11-18

**Summary Of Contributions:**

This paper introduces the "Foundation Model Transparency Index" - an index of 100 binary indicators to score foundation model developers with regards to their transparency around the process of developing and deploying their foundation models. The indicators are grouped into  three main pillars - upstream, model-level and downstream, which describe the model life cycle from its inception (i.e. data, hardware, compute) over the output artefacts (i.e. model) towards their usage.

The paper scores ten major foundation model developers on the index and presents and analyses the findings, suggesting that there is significant headroom for all developers to improve their transparency, with upstream transparency (regarding data usage and creation) affected the most.

The paper continues with different analyses, comparing similarities and contrasting differences between the developers and concludes with a set of comprehensive recommendations for foundation model developers, deployers and policy makers.

**Audience:**

Yes

**Broader Impact Concerns:**

As per my second weaknesses point, I think the paper should acknowledge that indices such as the one proposed here could lead to regulatory capture which stifles competition and actually has the inverse effect of what's intended, by eliminating non-for-profit ad small-scale enterprises. Overall, I don't think this would require a dedicated Broader Impact Statement.

**Claims And Evidence:**

Yes

**Requested Changes:**

As per the weaknesses outlined above:

- In Section 8, please argue more precisely which findings result in which recommendations
- In Section 9, please add a discussion on how the index itself could lead to regulatory capture and favour large companies, alongside with an outlook on future work e.g., whether it would make sense to hold different types of developers to different standards (e.g. a fully open enterprise that has limited means of hosting their developed models but instead relies on publishing weights will likely be unable to track its usage) and/or whether to pair it with a scoring system that awards point if the information provided is sufficient to understand the process evaluated by an indicator.
- Please include AI2 as a case-study of a non-for-profit publisher with their fully open Olmo/Dolma suite or a convincing argument as to why not.


Minor remarks:

- page 10 2nd bullet missing a citation
- page 11 "vital in fostering improvement over time" requires citation.
- page 28 second line in chapter 7: typo
- detailed figures (pp 32-35, 38) are missing
- page 37: citation is broken
- page 23: how are points 1 and 2 measured?

**Strengths And Weaknesses:**

In my opinion, the development of the index and the corresponding empirical study that scores developers is a laudable effort. The process seems sound and was carried out fairly, the chosen indicators are driven by pragmatism, the results and analyses are comprehensive. My appreciation of the article may seem brief in contrast to the rest of the review, but I do want to underscore that in my opinion this is sound work that is _worthy_ of publication in TMLR.

Some of the weaknesses that I identified:

With all the effort, the findings are not too surprising and are largely driven by the developers' goals and broader business strategy. Similar is the case with the pairwise developer comparisons. The recommendations, particularly those for developers, could be seen as common-sense and while they broadly follow from the empirical study and analysis, I think they should be reframed, in order to more precisely argue which findings lead to which recommendation.

Two more abstract points: Firstly, as the paper rightfully mentions, the index could be gamed quite easily by just publishing minimal information that satisfies the indicators (e.g., documenting that some information pertaining to an indicator is not disclosed would technically satisfy the indicator). I wonder if it would make sense to pair the index with a scoring system that awards point if the information provided is actually sufficient, for a more robust take.

Secondly, implicitly or explicitly, the design of the index favours large companies, by equally weighting certain things that big companies have the resources to do (as opposed to non-for-profits or startups) with arguably more fundamental issues regarding transparency. To exemplify, a large company will have the legal "fire-power" to draft comprehensive usage policies, TOS etc, allowing them to score easily in this dimensions, while concealing information about the equally-weighted data sources, data licenses and annotation wages. This manifests itself in the overall evaluation scores, where Meta ranks first (perhaps ironically, since Meta was used as the very example of bad transparency practices) and OpenAI ranks third. This could further contribute to the "regulatory capture" mentioned on page 56. While it is positive that the paper is concerned with this phenomenon, I think it should acknowledge that The Foundation Index could contribute to it (e.g. in Section 9.2).

To further emphasise the previous point, the restriction of choosing developers as companies (big tech or startup) seems to imply that developing foundation models is driven by industry. While I understand that the evaluation has been carried out before the publication of AI2's Olmo/Dolma models, it would be interesting to see, how an "fully open by design" foundation model published by a non-for-profit organisation, would score on the index, and whether any of the decisions made by AI2 could be used as an example to improve transparency.

---

> ### Author Response · Authors · 2024-12-21
> **Author Response to Reviewer mY2L**
>
> We thank the reviewer for their thorough response and appreciate that they believe this work is "worthy of publication in TMLR". To respond to the specific points raised:
>
> > With all the effort, the findings are not too surprising and are largely driven by the developers' goals and broader business strategy. Similar is the case with the pairwise developer comparisons. The recommendations, particularly those for developers, could be seen as common-sense and while they broadly follow from the empirical study and analysis, I think they should be reframed, in order to more precisely argue which findings lead to which recommendation.
>
>
> We fully agree that the overall finding of opacity is not surprising, but we do believe it is important to empirically quantify this. More broadly, we believe many of the lower-level findings were at least information that we did not know (e.g. disparities across developers on what they disclose), though some elements were anticipatable (e.g. open developers being more transparent about how they build models than their closed counterparts). As to the relationship between our findings and recommendations, we would be happy to provide this mapping.
>
> > Firstly, as the paper rightfully mentions, the index could be gamed quite easily by just publishing minimal information that satisfies the indicators (e.g., documenting that some information pertaining to an indicator is not disclosed would technically satisfy the indicator). I wonder if it would make sense to pair the index with a scoring system that awards point if the information provided is actually sufficient, for a more robust take
>
>
> To be clear, only for very few indicators is it the case that disclosing "we do not disclose this information" would qualify for a point (e.g. we award a point to companies that do not disclose wages if they disclose they don't conduct involve human labor for data creation). In contrast, for many of the indicators there is some specific criterion that must be met for the information to be sufficiently detailed to award a point (e.g. compute must be disclosed in FLOPs up to one significant figure + the order of magnitude, e.g. 3 x 10^25 FLOPs).
>
>
> > Secondly, implicitly or explicitly, the design of the index favours large companies, by equally weighting certain things that big companies have the resources to do (as opposed to non-for-profits or startups) with arguably more fundamental issues regarding transparency. To exemplify, a large company will have the legal "fire-power" to draft comprehensive usage policies, TOS etc, allowing them to score easily in this dimensions, while concealing information about the equally-weighted data sources, data licenses and annotation wages. This manifests itself in the overall evaluation scores, where Meta ranks first (perhaps ironically, since Meta was used as the very example of bad transparency practices) and OpenAI ranks third. This could further contribute to the "regulatory capture" mentioned on page 56. While it is positive that the paper is concerned with this phenomenon, I think it should acknowledge that The Foundation Index could contribute to it (e.g. in Section 9.2).
>
>
> We do agree that many aspects of the Index implicitly could be seen as benefitting higher-resourced entities that have technical/legal/policy personnel and resources to expend on documenting information. And we agree there can be a substitution here in terms of the overall score (i.e. prioritizing indicators that can be achieved through more legal/policy work as opposed to technical work). We would be happy to acknowledge this more directly alongside related discussion of transparency washing.
>
> **Requested changes**
>
> Overall, we are happy to address the two requested changes in Sections 8 and 9 as well as address all typos. As to scoring AI2, we do not plan to this at this time since: (i) the 2023 Index only scores entities for their 2023 practices, so scoring AI2 for 2024/2025 practices would be confusing at the least and (ii) we deliberately chose to not score non-commercial entities as we describe in our selection criteria in Section 5.1 for multiple reasons, including regulatory alignment with regulations that only regulate companies developing foundation models.

---

> > ### Comment · Reviewer_mY2L · 2025-01-13
> > **OK**
> >
> > I can see your rationale why you don't want to include AI2, but I still maintain that this might paint a distorted picture and your conclusions will favour certain companies unfairly.
> >
> > As for my other points, I am okay with the suggested edits.

---

### Decision · Action_Editor_e1RW · 2025-02-04

**Recommendation:** Accept with minor revision

**Comment:**

All reviewers agree that establishing guidelines for transparent foundation model development is essential, and this paper represents a significant contribution.
The Foundation Model Transparency Index is acknowledged as an important milestone in this effort.

While some concerns were raised, the authors' proposed updates are considered sufficient.
Overall, the reviewers find the study methodologically sound, well-considered in its selection of indicators, and comprehensive in its analyses.

The following is my request to the authors upon the acceptance of the paper:
- Incorporate all suggested changes to reflect the feedback provided.
- Properly format the paper to align with TMLR guidelines. Specifically:
  - Ensure that URLs in footnotes do not exceed the page width.
  - Adjust figures (such as Figures 10 and 11) so that they fit within the TMLR format. Additionally, remove any unintended formatting options that appear in the text (e.g., `margin=1.7cm`).
- De-anonymize the paper as required for the final submission.

**Audience:**

The transparency of LLMs is a topic of significant interest within the current ML community.
This study aligns well with the scope of TMLR.

**Claims And Evidence:**

The paper introduces the "Foundation Model Transparency Index" designed to evaluate how openly foundation model developers share information about their models.
The index comprises 100 binary indicators organized into three categories:

* Upstream: How models are constructed, including details about data, hardware, and compute resources.
* Model-level: Information about the models themselves.
* Downstream: How models are applied.

The authors applied this index to assess the transparency practices of ten leading foundation model developers.
Their analysis indicates that while all developers have room for improvement, transparency—especially regarding the upstream processes like data usage and creation—is notably missing.

In addition to comparing the transparency levels among different developers, the paper offers recommendations for model developers, deployers, and policymakers, aiming to enhance transparency of LLMs.